# Saturation genome editing maps the functional spectrum of pathogenic *VHL* alleles

Megan Buckley [1], Chloé Terwagne [1], Athina Ganner[2,12], Laura Cubitt [1,12], Reid Brewer[3,12], Dong-Kyu Kim[3,12], Christina M. Kajba[1], Nicole Forrester[1], Phoebe Dace[1], Joachim De Jonghe [1], Scott T. C. Shepherd[4,5,6], Chelsea Sawyer[7], Mairead McEwen[3,8], Sven Diederichs[9,10], Elke Neumann-Haefelin[2], Samra Turajlic [4,5,6], Evgueni A. Ivakine [3,8,11] & Gregory M. Findlay [1]✉

To maximize the impact of precision medicine approaches, it is critical to identify genetic variants underlying disease and to accurately quantify their functional effects. A gene exemplifying the challenge of variant interpretation is the von Hippel–Lindau tumor suppressor (*VHL*). *VHL* encodes an E3 ubiquitin ligase that regulates the cellular response to hypoxia. Germline pathogenic variants in *VHL* predispose patients to tumors including clear cell renal cell carcinoma (ccRCC) and pheochromocytoma, and somatic *VHL* mutations are frequently observed in sporadic renal cancer. Here we optimize and apply saturation genome editing to assay nearly all possible single-nucleotide variants (SNVs) across *VHL*'s coding sequence. To delineate mechanisms, we quantify mRNA dosage effects and compare functional effects in isogenic cell lines. Function scores for 2,268 *VHL* SNVs identify a core set of pathogenic alleles driving ccRCC with perfect accuracy, inform differential risk across tumor types and reveal new mechanisms by which variants impact function. These results have immediate utility for classifying *VHL* variants encountered clinically and illustrate how precise functional measurements can resolve pleiotropic and dosage-dependent genotype–phenotype relationships across complete genes.

Delineating rare genetic variants underlying disease phenotypes remains a major challenge in human genetics. For the majority of cancer-associated genes, more variants of uncertain significance (VUS) have been reported than variants whose phenotypic effects are known[1–4]. In the context of both germline testing and tumor profiling, VUS represent a missed opportunity to improve patient care through precision medicine approaches.

Most variants observed are too rare to enable statistically robust genotype–phenotype associations. Computational models of variant effects have improved due to greater availability of training data and the use of machine learning[5–10]. However, such models are not accurate enough to dictate clinical decisions without additional evidence[11]. Mechanistic knowledge of variants in tumor suppressor genes can inform which individuals will benefit from preventative measures and guide therapeutic selection[12–14]. There is, however, a scarcity of functional data available for linking variants to phenotypes[15].

The von Hippel–Lindau (VHL) tumor suppressor is a 213-amino acid protein encoded on chromosome 3p that functions as an E3

**Fig. 1 | A highly optimized SGE protocol to assay *VHL* variants. a**, CRISPR knockout screening data from the Cancer Dependency Map (DepMap)[26] reveal *VHL* loss widely leads to reduced growth in cell lines lacking *VHL* mutations ($n = 29$ kidney-derived and $n = 1,048$ other lines; boxplot: center line, median; box limits, upper and lower quartiles; whiskers, 1.5× interquartile range; all points shown). **b**, CRISPR-induced editing of *VHL* was performed in HAP1 cells (day 0), and outcomes were quantified by NGS. Distributions of InDel scores, calculated as the $\log_2$ ratio of day 13 frequency to day 6 frequency, show frameshifting, and in-frame InDels are strongly depleted in parental HAP1 (left) compared to HAP1–HIF1A–knockout (–KO) (right) (median InDel score −3.20 versus −0.20, respectively; Wilcoxon rank-sum two-sided $P = 5.6 \times 10^{-18}$). **c**, The strategy to perform SGE across the complete coding sequence of *VHL* is shown, with ClinVar variant counts for all 'pathogenic' and 'likely pathogenic' variants (red) and VUS (orange) displayed from gnomAD[39]. SGE regions were designed to tile exons 1–3, as well as a region of intron 1. A total of 480 SNVs in ClinVar are in SGE regions, of which 269 are VUS. (Introns not to scale.) **d**, For each SGE region, a library of oligos containing all possible SNVs was synthesized and cloned into a vector with homology arms to facilitate genomic integration via CRISPR-induced HDR. Variants present in cells were quantified over time via amplicon sequencing, and function scores were calculated to reflect variants' effects on fitness. **e**,**f**, Function scores for synonymous, nonsense and canonical splice site SNVs are shown for a single SGE region (exon 2) assayed in normal media (**e**) or media supplemented with 2.5 µM DAB (**f**).

ubiquitin ligase in complex with cullin-2, elongins C and B (ELOC and ELOB) and ring-box 1 (ref. 16). In normoxic conditions, VHL ubiquitinates the α-subunit of hypoxia-inducible factor (HIF), targeting HIF for proteasomal degradation. In hypoxic conditions, HIF is protected from VHL-mediated degradation and signals to promote glycolysis and angiogenesis. Loss of VHL function due to mutation can lead to constitutive HIF activity and tumor development[17].

Somatic *VHL* mutations are frequently observed in renal cell carcinomas (RCCs), most commonly clear cell RCC (ccRCC). During ccRCC evolution, chromosome 3p deletion typically precedes a loss-of-function (LoF) mutation to the remaining *VHL* allele, resulting in increased HIF activity[18]. *VHL* mutations have been observed in other types of RCC and extrarenal cancers, but their functional significance is less certain.

Pathogenic germline variants in *VHL* predispose patients to different neoplasias in an autosomal dominant manner, a condition known as VHL disease[19]. Affected patients have varying susceptibilities to different tumors, including ccRCC, pheochromocytoma and hemangioblastoma. The risk of each tumor depends largely on the specific germline variant. Classically, type 1 VHL disease variants lead to complete LoF and include whole-gene deletions, nonsense variants and frameshifting insertions and deletions (InDels). Type 1 variants predispose patients

to ccRCC, hemangioblastoma and other neoplasms in a HIF-dependent manner. Type 2 variants, in contrast, are associated with high pheochromocytoma risk and are most often missense variants. Attempts have been made to subclassify type 2 VHL disease further—type 2C disease is marked by pheochromocytomas only, type 2A disease includes hemangioblastomas and other benign tumors and type 2B disease further includes ccRCC[20].

These clinical classifications have helped explain patterns of tumors in families, yet a complete molecular accounting of how different mutations confer distinct pathologies has remained elusive. In a curated database of *VHL* mutations[21], many variants have been associated with both type 1 and type 2 disease. Other variants have been implicated in recessive diseases, such as congenital polycythemia or germline VHL deficiency[22,23]. Although protein-truncating variants typically cause type 1 disease, in rare instances, patients with nonsense variants have presented with type 2 disease[24]. Such discrepancies highlight the challenge of developing individualized surveillance and therapy plans for patients without functional evidence of variants' effects.

In addition to variants known to cause VHL disease, there are over 800 VUS in *VHL* reported in ClinVar[3]. It is unknown what fraction of these variants cause disease, and likewise how many variants yet to be reported may prove pathogenic. While genetic evidence may be

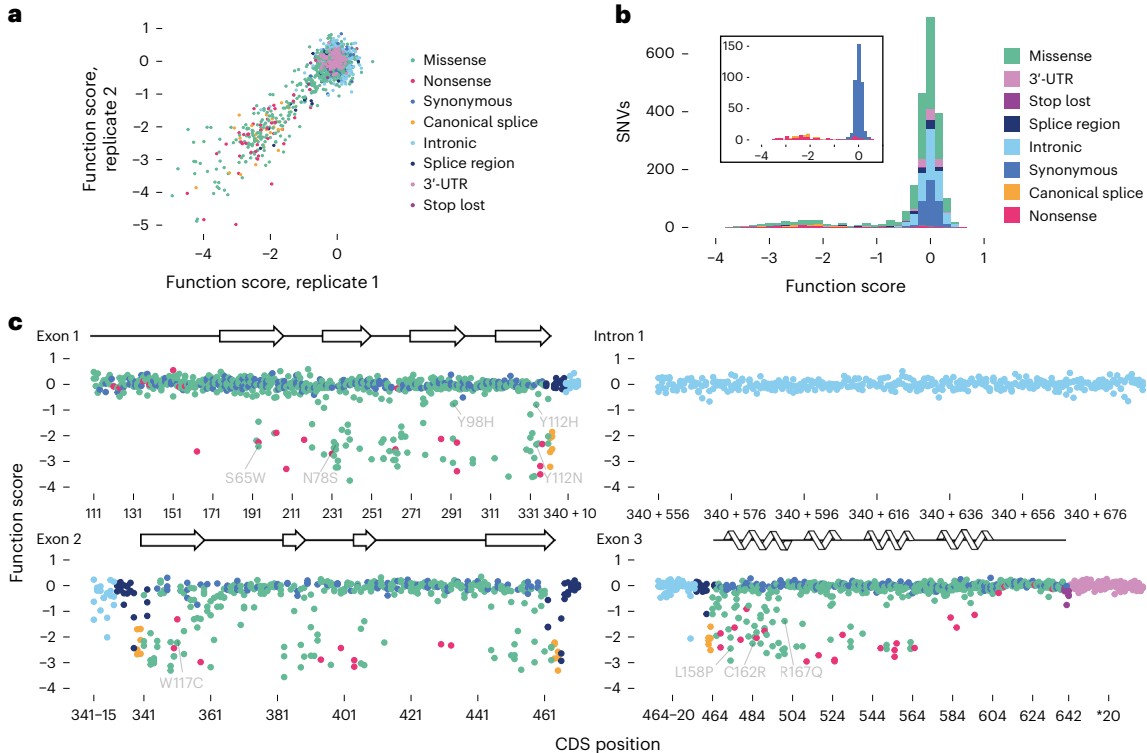

**Fig. 2 | A complete map of SNV effects across *VHL*. a**, Function scores across transfection replicates are plotted for *n* = 2,200 SNVs (Pearson's *R* = 0.90). **b**, Replicate scores were averaged and normalized to obtain a final function score for each variant. The inset shows only synonymous, nonsense and canonical splice site SNVs. **c**, Function scores are plotted by coding sequence (CDS)

position for each coding and intronic region assayed, with β-sheets and α-helices of VHL's secondary structure[36] shown above. Scores of select well-studied VHL disease variants are indicated by amino acid substitution, with SGE data for additional variants of known phenotype in Supplementary Table 2.

converging on a near-complete set of pathogenic *VHL* alleles[21], substantially more variants are predicted by computational models to be deleterious than have been linked to disease[25]. It is unclear whether such variants have yet to be encountered due to their rarity, whether they are incompatible with life or whether they are truly benign.

For patients harboring *VHL* variants whose phenotypic effects are unknown, well-calibrated functional data may prove useful in aiding diagnosis and management. The recently demonstrated efficacy of HIF2α inhibitors for preventing ccRCC progression[14] suggests quantifying variants' precise effects on HIF regulation may prove valuable for guiding therapeutic selection. More broadly, *VHL* serves as a powerful model to assess the extent to which functional data can recapitulate genotype–phenotype relationships in humans, owing to the fact that it is frequently mutated in ccRCC[2] and the extensive knowledge regarding phenotypic effects of germline variants[21].

Here we systematically measure the functional consequences of *VHL* variants across the complete gene by using saturation genome editing (SGE). In total, we scored 2,268 single-nucleotide variants (SNVs) for HIF-dependent effects on cellular fitness, defining LoF variants underlying ccRCC development with 100% accuracy. Our assay captures clinically meaningful differences in the degree of functional impairment among pathogenic alleles and reveals new mechanisms explaining genotype–phenotype associations, suggesting a role in improving diagnostic and therapeutic precision.

## Results

### An optimized SGE assay to precisely score *VHL* variants

To develop a high-throughput assay for *VHL* variants, we assessed the effect of *VHL* loss across cell lines. Except for kidney-derived lines, CRISPR-induced knockout of *VHL* almost uniformly reduces cell fitness (Fig. 1a)[26,27]. To investigate *VHL*'s essentiality in the haploid human line

HAP1, InDels in exon 2 were generated with CRISPR and sequenced at multiple timepoints. Robust depletion of InDels over time confirmed the essentiality of *VHL* for normal HAP1 proliferation (Fig. 1b). The strong selection against InDels was eliminated by prior knockout of *HIF1A* (Fig. 1b), indicating *VHL* loss confers a HIF-dependent growth defect.

SGE is a method by which all possible SNVs in a genomic region are assayed in multiplex using CRISPR–Cas9 editing[28]. When SGE is performed in HAP1, a single variant is engineered per haploid cell, allowing variants' effects on growth to be quantified by next-generation sequencing (NGS). Seven SGE libraries were made to tile the coding sequence of *VHL*, as well as exon-proximal regions of introns and a region deep within intron 1 covering the 5′ end of a reported pseudoexon (Fig. 1c)[29,30]. Each library consisted of all possible SNVs in a region cloned into vectors with homology arms to facilitate genomic integration (Fig. 1d).

To measure more subtle growth effects, SGE experiments were performed using a highly optimized protocol modified from published work[31] to feature improved transfection efficiency, a longer time course and addition of 10-deacetyl-baccatin-III (DAB) to maintain haploidy[32] (Fig. 1e,f, Extended Data Figs. 1 and 2 and Supplementary Note). Amplicon sequencing of genomic DNA (gDNA) collected on days 6 and 20 was used to calculate a 'function score' for each SNV reflective of cellular fitness (Methods). SNVs with significantly reduced function scores (that is, 'depleted' SNVs) were defined for each SGE region by applying a false discovery rate (FDR) of 0.01. After stringent quality filtering, function scores for *n* = 2,268 SNVs were obtained, comprising 85.4% of all possible SNVs in SGE regions (Supplementary Table 1).

### Mapping LoF variants

The majority of variants designed but not scored map to the GC-rich 5′ region of exon 1, where the rate of editing was lowest (Extended Data Fig. 3a). In contrast to other coding regions, the 5′ region of exon 1

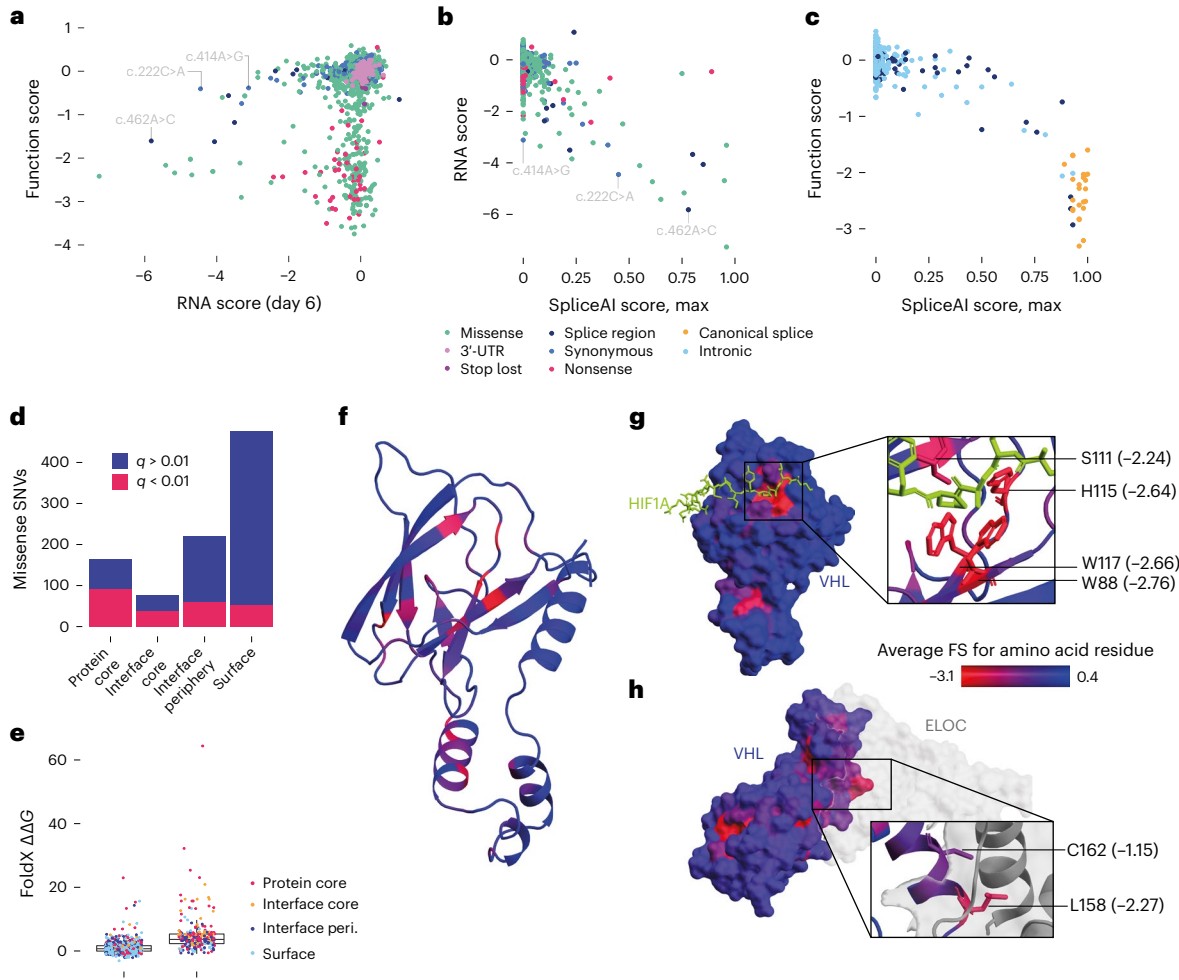

**Fig. 3 | Function scores capture fitness effects secondary to splicing alterations and impairment of protein function. a**, Targeted RNA-sequencing of *VHL* mRNA from edited cells was performed to calculate RNA scores for *n* = 1,626 exonic SNVs. The relationship between RNA scores and function scores reveals most LoF variants are expressed at normal levels in mRNA, and only low RNA scores reliably predict LoF at the cellular level (Extended Data Fig. 4c). Two synonymous variants shown independently to disrupt splicing are indicated[23,57], as well as c.462A>C, the splice region variant with the lowest RNA score. **b,c**, The maximum SpliceAI score for each SNV is plotted against RNA scores for exonic SNVs (Pearson's *R* = −0.70; **b**) and function scores for intronic SNVs (*R* = −0.90; **c**).

**d**, The proportion of missense variants scoring as depleted by SGE (*q* < 0.01) is displayed for each residue exposure label. **e**, FoldX Δ*G* predictions were higher for *n* = 242 depleted missense SNVs (median = 3.63) than for *n* = 697 nondepleted SNVs (median 0.70) (boxplot: center line, median; box limits, upper and lower quartiles; whiskers, 1.5× interquartile range). **f–h**, The average score of missense variants at each amino acid position is shown in color on the VHL structure (PDB: 1LM8)[58]. Residues highly intolerant to missense variation include S111, H115, W117 and W88 at the HIF1A recognition site (**g**), as well as L158 and C162 at the ELOC interface (**h**).

lacked significantly depleted SNVs, including nonsense variants (Extended Data Fig. 3b,c). This is consistent with a previously characterized alternative translation initiation site at p.M54 producing a fully functional VHL isoform[33], and corroborates the lack of pathogenic variants proximal to p.M54 in ClinVar (Fig. 1c).

Moving forward, we restricted analyses to *n* = 2,200 SNVs scored with high reproducibility by excluding SNVs assayed in the 5′-most SGE region (Fig. 2a,b). Among *n* = 115 remaining SNVs upstream of p.M54, none were significantly depleted. Likewise, no SNVs assayed in the 3′-untranslated region (UTR) or deep within intron 1 scored significantly (Fig. 2b,c). In contrast, between p.M54 and p.R200, all but one nonsense variant was depleted (*n* = 43, median score = −2.4), as were all canonical splice site SNVs (*n* = 24, median score = −2.3; Fig. 2b,c). Most missense variants scored neutrally, although 22.4% were depleted.

To better resolve mechanisms of functional impairment, we derived *n* = 1,626 'RNA scores' for coding variants by performing targeted RNA-sequencing on day 6 and day 20 samples (Extended Data Fig. 4

and Supplementary Table 1). RNA scores reflect SNVs' effects on full-length *VHL* mRNA levels.

Comparison of RNA scores to function scores reveals only large reductions in mRNA confidently predict LoF (Fig. 3a). Indeed, 16 of 17 SNVs with RNA scores below −3.0 were significantly depleted, reflecting a minimum mRNA dosage required for normal growth (Extended Data Fig. 4c). While RNA scores across timepoints were highly correlated, variants depleted in mRNA on day 6 tended to be less depleted on day 20 (Extended Data Fig. 5a–e), suggesting selection for cells expressing sufficient *VHL* mRNA. Indeed, only 6 of 17 variants with day 6 RNA scores below −3.0 had day 20 RNA scores below −3.0.

Splicing predictions from SpliceAI[10] were strongly correlated with RNA scores (Fig. 3b and Extended Data Fig. 5f). In total, 16 of 17 SNVs with RNA scores less than −3.0 had SpliceAI scores greater than 0.08 (median of 0.61), whereas 83% of variants with RNA scores greater than −3.0 had SpliceAI scores of 0.00. The only variant with a low RNA score and a SpliceAI score of 0.00, c.414A>G, is a pathogenic variant known to promote exon 2 skipping[34]. While we cannot measure RNA

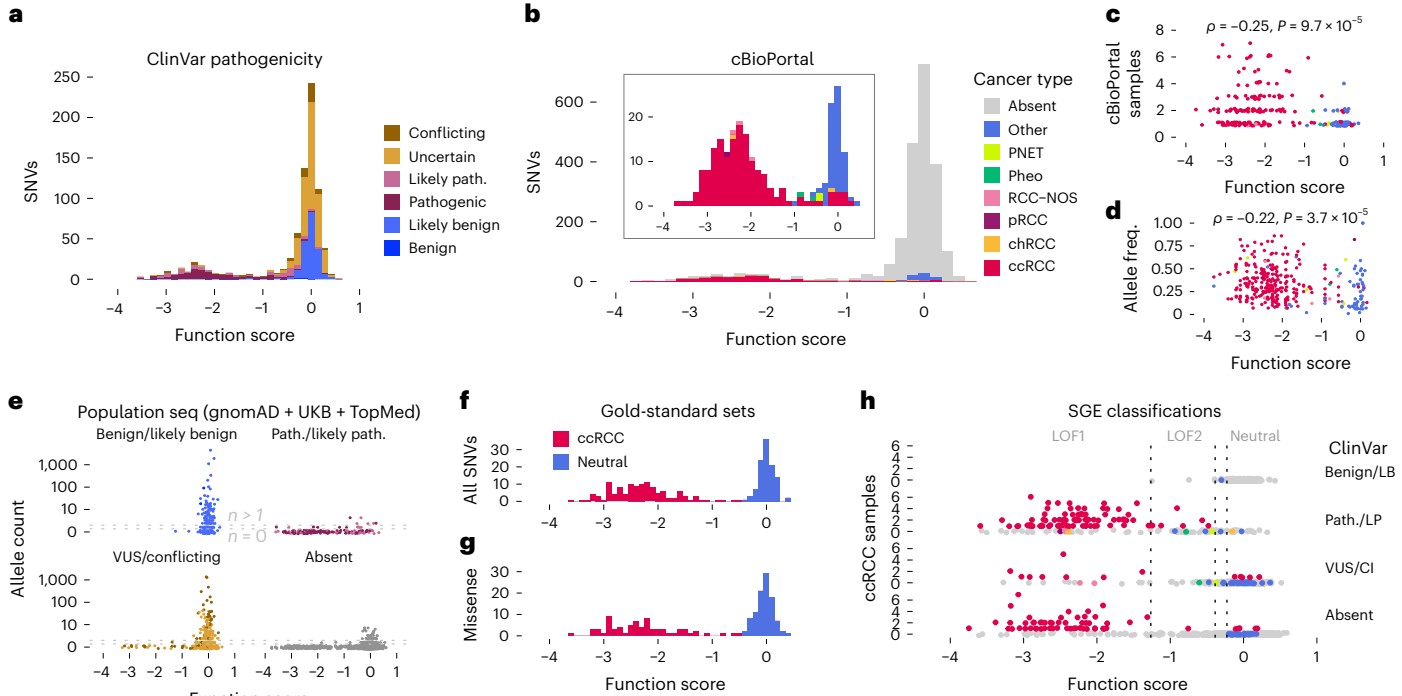

**Fig. 4 | Highly accurate identification of pathogenic *VHL* alleles in humans using SGE. a,b**, The distribution of function scores for *VHL* SNVs reported in ClinVar[3] (**a**) and observed in cBioPortal[1,2] cancer samples (**b**) is shown. **c,d**, For SNVs observed in tumors, lower function scores correlate with more independent observations in cBioPortal (**c**; $n = 233$ SNVs, Spearman's $\rho = -0.25$, two-sided $P = 9.7 \times 10^{-5}$; $\rho = -0.58$, $P = 2.7 \times 10^{-22}$ when analysis restricted to $n = 172$ SNVs seen in ccRCC) and higher allele frequencies within samples (**d**; $n = 334$ samples, $\rho = -0.22$, two-sided $P = 3.7 \times 10^{-5}$). **e**, A combined population allele count for each SNV assayed was determined by summing independent observations from gnomAD[39], UKB[38] and TOPMed[40]. Of $n = 233$ variants observed more than once in population sequencing, 96.6% were not significantly depleted. **f,g**, Gold-standard sets of variants were defined using orthologous data from ClinVar, cBioPortal and population sequencing. In **f**, all variants with at least two cBioPortal ccRCC observations or one ccRCC observation and a ClinVar

'pathogenic' or 'likely pathogenic' annotation ($n = 120$ ccRCC-associated SNVs) are plotted with variants deemed 'benign' or 'likely benign' in ClinVar and seen at least once in population sequencing ($n = 108$ neutral SNVs). **g**, Due to the lack of missense variants classified as 'benign' or 'likely benign' in ClinVar, we defined a neutral set to include those present in at least two population controls and not deemed 'pathogenic' or 'likely pathogenic' in ClinVar ($n = 99$ neutral missense SNVs versus $n = 73$ ccRCC missense SNVs). **h**, Function scores are plotted against cBioPortal ccRCC observations for SNVs reported in ClinVar to be 'benign' or 'likely benign' (LB), 'pathogenic' or 'likely pathogenic' (LP) or VUS (here including SNVs with conflicting interpretations), and for SNVs absent from ClinVar. SNVs are colored by cancer type (as in **b**). Lines correspond to thresholds for distinguishing LOF1 (less than −1.26), LOF2 (less than −0.39) and neutral (greater than −0.23) classes.

scores for intronic SNVs, SpliceAI scores also correlate highly with function scores for intronic variants ($R = -0.90$; Fig. 3c), implicating splice disruption as the mechanism driving functional effects. Of note, nonsense variants as a class were only minimally depleted in mRNA (median RNA score = −0.19; Extended Data Fig. 4a), indicating minimal nonsense-mediated decay.

We next explored the features of missense variants impacting function. While 55.8% of mutations in the protein core and 49.4% of mutations in the interface core were depleted, only 26.7% of peripheral interface mutations and 11.1% of surface mutations were depleted (Fig. 3d). FoldX-predicted[35] $\Delta\Delta G$ values were higher for variants with low function scores (median $\Delta\Delta G = 3.63$ for depleted SNVs versus median $\Delta\Delta G = 0.70$ for other missense SNVs; Fig. 3e). Indeed, 76.9% of depleted missense variants had $\Delta\Delta G$ predictions greater than 2.0, compared to 20.6% of missense variants not depleted by SGE. Restricting the analysis to significantly depleted missense variants, $\Delta\Delta G$ predictions correlate inversely with function scores ($\rho = -0.41$), indicating the degree of destabilization is predictive of the level of functional impairment.

Overlaying function scores to the VHL structure[36] reveals LoF missense variants tend to occur in β-sheets and α-helices (Fig. 3f). Residues highly intolerant to missense variation include those forming the substrate recognition site as well as specific contacts with ELOC (Fig. 3g,h). Collectively, these findings indicate that most LoF variants

exert effects by altering function at the protein level, with only large decreases in *VHL* mRNA sufficient to cause functional impairment.

For validation, SNVs spanning a wide range of function scores were introduced independently to HAP1 cells. Western blots confirmed that 9 of 9 variants with significantly low function scores led to increased HIF1A expression (Extended Data Fig. 6). Two synonymous variants with low RNA scores, c.222C>A and c.462A>C, resulted in reduced VHL protein expression, but only c.462A>C led to clear HIF1A upregulation, consistent with its lower RNA score and function score. Overall, these data confirm that *VHL* variants depleted in SGE impair HIF1A regulation in HAP1 cells.

### Function scores distinguish *VHL* variants driving disease

To assess whether function scores predict variant pathogenicity, we performed several analyses. First, including all *VHL* variant annotations in ClinVar meeting assertion criteria, function scores distinguish 'pathogenic' variants from 'benign' and 'likely benign' variants with 95.2% sensitivity and 97.9% specificity (Fig. 4a and Extended Data Fig. 7a,c,d). Notably, strong predictive performance is observed specifically for missense and splice region variants (Extended Data Fig. 7e,f). These broad analyses include variants with diverse phenotypes described in ClinVar.

Next, tumor sequencing data from cBioPortal[1,2] were used to examine the functional effects of *VHL* mutations across human cancers. Over 93% of SNVs seen in at least one RCC sample of any type scored

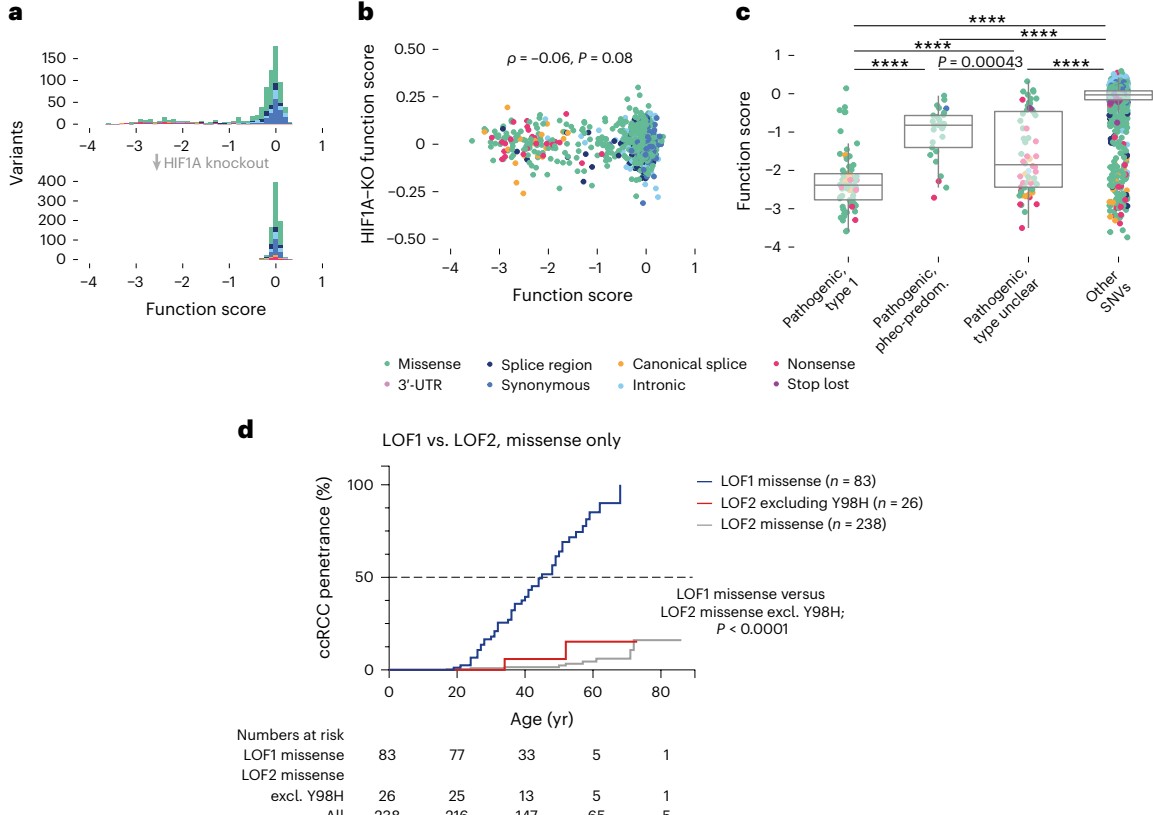

**Fig. 5 | A gradient of functional effects underlies phenotypic differences of *VHL* variants. a**, SNVs (*n* = 797) in exons 2 and 3 were assayed in HAP1–HIF1A–KO cells. Compared to previous data (top), variants were well-tolerated, independent of consequence. **b**, Function scores across isogenic lines showed no significant correlation (Spearman's *ρ* = −0.06, *P* = 0.08). **c**, 'Pathogenic' and 'likely pathogenic' SNVs from ClinVar were grouped based on annotations in VHLdb[21]. SNVs associated only with type 1 VHL disease or ccRCC were deemed 'type 1' (*n* = 74 SNVs), whereas SNVs associated only with type 2 disease or predominantly pheochromocytoma were deemed 'pheo-predominant' (*n* = 29 SNVs, excluding SNVs associated with type 2B disease). The remaining pathogenic SNVs lacked unambiguous phenotypic data in VHLdb (*n* = 64 SNVs, 'type unclear'). The boxplot shows function scores for SNVs in these categories, as well as for *n* = 2,033 other SNVs assayed, including variants either not deemed pathogenic in ClinVar or absent (one-way ANOVA, adjusted *P* = 0.00043 between 'pheo-predominant' and 'type unclear'; ****P* < 1.0 × 10⁻¹⁰ for all other comparisons; boxplot: center line, median; box limits, upper and lower quartiles; whiskers, 1.5× interquartile range; all points shown). **d**, Patients in the Freiburg VHL Registry with missense variants assayed by SGE (*n* = 321) were grouped based on the function class of their germline variant. Due to the high prevalence of p.Y98H in this cohort, an additional LOF2 group excluding p.Y98H was analyzed, as well. A Kaplan–Meier estimator was used to assess age-related ccRCC penetrance (log-rank test for significance).

as depleted (Fig. 4b and Extended Data Fig. 7b). This was in contrast to *VHL* variants in other tumor types, which typically scored neutrally, with the exception of variants in pheochromocytomas and pancreatic neuroendocrine tumors (PNETs)—extrarenal tumors also linked to germline *VHL* mutations[37]. SNVs observed in more patients and at higher allele frequencies had significantly lower function scores, confirming variants depleted in SGE to be drivers of oncogenesis (Fig. 4c,d).

In contrast to SNVs identified in RCC, nearly all *VHL* SNVs present in population sequencing databases scored neutrally (Fig. 4e). For example, among *n* = 119 SNVs seen at least five times in total across the UK Biobank (UKB)[38], gnomAD[39] and TOPMed[40] databases, no SNV scored below −0.40 (mean = −0.03, s.d. = 0.14). Likewise, the lowest function score for any SNV seen at least twice was −0.77. The narrow distribution of scores around zero indicates that the vast majority of variants seen more than once in population sequencing are unlikely to cause VHL disease.

Considering the inherent uncertainty in both germline and somatic variant classification, we defined a 'gold-standard' set of *n* = 120 ccRCC-associated variants supported by at least two independent lines of evidence. SGE function scores perfectly separate these ccRCC-associated SNVs from *n* = 108 SNVs deemed 'benign' or 'likely benign' in ClinVar and encountered in population sequencing (Fig. 4f). ClinVar lacks 'benign' and 'likely benign' missense variants in

*VHL*. However, function scores also cleanly separate *n* = 73 missense SNVs in the gold-standard ccRCC set from *n* = 99 missense SNVs seen in population sequencing at least twice and not deemed 'pathogenic' or 'likely pathogenic' in ClinVar (Fig. 4g).

Together, these analyses indicate that function scores can be used as strong evidence to support variant classification. Toward this end, we defined four function classes reflective of variants' scores in relation to gold-standard distributions (Methods). In summary, 'LOF1' and 'LOF2' variants both have significantly low function scores, although only 'LOF1' variants scored comparably to gold-standard ccRCC variants. 'Neutral' variants scored similarly to gold-standard benign variants, whereas 'intermediate' variants were scored ambiguously by SGE.

Of *n* = 430 SNVs reported as VUS or with conflicting interpretations in ClinVar, 39 (9.1%) scored as LOF1 or LOF2 (Fig. 4h and Extended Data Fig. 7g). A comparable fraction of SNVs absent from ClinVar scored as such (9.9% of *n* = 1,406 SNVs), many of which have been observed in ccRCC already. LOF1 variants were far more likely to be observed in ccRCC samples than LOF2 variants (*n* = 225 LOF1 SNVs seen in 1.30 ccRCC samples on average versus *n* = 102 LOF2 SNVs seen in 0.11 samples on average; Extended Data Fig. 7h), indicating the degree of functional impairment is strongly linked to the likelihood that a ccRCC will develop once a mutation arises somatically.

Applying function classes to variants in cBioPortal reveals that 28 of 83 SNVs not currently deemed 'oncogenic' likely promote cancer development (Extended Data Fig. 7i). Most of these 28 were identified in ccRCC. Conversely, of $n = 150$ SNVs currently deemed 'oncogenic', 10.0% scored as neutral by SGE. Such SNVs were most often identified in tumors not associated with *VHL* mutations, which is consistent with these variants not driving disease. In light of these findings, SGE data have clear potential for improving the interpretation of variants seen in cancer.

Finally, we compared function scores for missense variants to outputs from computational predictors, including CADD[25], REVEL[5], EVE[7], boostDM[8] and VARITY[9]. Overall, SGE scores performed substantially better at predicting pathogenic missense variants in ClinVar as well as missense variants in the gold-standard ccRCC set (Extended Data Fig. 8a,b). While missense variants scored lowly by SGE are generally well supported by computational prediction, many missense SNVs scored neutrally are predicted to be deleterious (Extended Data Fig. 8c–e). The absence of such discordantly scored variants from the gold-standard ccRCC set indicates that the computational models lack specificity for this phenotype, particularly for highly conserved variants.

## Discovery of new mechanisms explaining clinical phenotypes

To ask whether function scores for SNVs might partially reflect HIF-independent effects, we repeated SGE experiments for SNVs in exons 2 and 3 in HIF1A-knockout cells ($n = 797$ SNVs). All variant effects were effectively eliminated (Fig. 5a,b), indicating function scores specifically reflect effects on HIF regulation.

Previous studies have shown that certain germline variants conferring high pheochromocytoma risk cause less HIF upregulation[41]. Therefore, we grouped pathogenic variants by phenotypic annotations from the VHL Database (VHLdb)[21]. Variants associated only with type 1 disease or ccRCC were deemed 'type 1', whereas variants associated only with type 2 disease or predominantly pheochromocytomas were deemed 'pheo-predominant'. Critically, the median function score for type 1 SNVs was 2.9-fold lower than for pheo-predominant SNVs (Fig. 5c). Pathogenic variants not classifiable in this manner spanned a range of scores, as did SNVs absent from ClinVar. This result confirms that variants causing high pheochromocytoma risk typically impair HIF regulation to a lesser extent than variants associated with type 1 disease.

To evaluate whether SGE data may be useful for stratifying patients with VHL disease, we used the Freiburg VHL Registry to assess the age-related risk of ccRCC. We observed a markedly higher penetrance of ccRCC for patients with LOF1 variants compared to LOF2 variants (Fig. 5d and Extended Data Fig. 9). This finding was consistent across analyses, including all SNVs, missense SNVs only and SNVs lacking ClinVar interpretations. Collectively, these results show that SGE data can be useful for predicting differential tumor risk across tissues.

In light of the SGE data's high predictive power, we reasoned function scores may be valuable for mechanistically resolving unexplained genotype–phenotype associations. We highlight two specific examples.

First, all nonsense SNVs located downstream of the C-terminal-most α-helix scored neutrally by SGE. This suggests that the last 12 amino acids of VHL are dispensable for HIF regulation. However, numerous InDels have been identified in ccRCC samples between p.L201 and p.*214, calling the function of the region into question. Mapping ccRCC-associated InDels with function scores shows InDels downstream of the last depleted nonsense SNV share a common reading frame that results in a 41-amino acid C-terminal extension (Fig. 6a,b). This is much longer than extensions created by InDels in other reading frames and nonstop SNVs.

To investigate this, we analyzed CRISPR-induced InDels in HAP1 and confirmed growth defects are reading frame-specific (Fig. 6c). We next engineered HAP1 lines expressing c.606dup, a ClinVar VUS

leading to the 41-amino acid extension. Across clonal lines, we observed loss of VHL expression and upregulation of HIF1A by microscopy and western blot (Fig. 6d,e). A similar degree of HIF1A upregulation was observed for c.620_624del but not c.606del, confirming reading frame specificity. Collectively, these results indicate that between p.R200 and p.*214, frameshifting InDels promote ccRCC development via a long C-terminal extension that destabilizes VHL.

A second observation concerns c.264G>A (p.W88*), a nonsense variant shown to segregate with VHL disease in a family[24]. c.264G>A is highly unusual among nonsense variants in that it was associated with type 2 disease marked by early onset pheochromocytomas. Interestingly, c.264G>A did not score as depleted in our assay (function score = −0.16), in contrast to all other nonsense variants between p.M54 and p.L198, including c.263G>A, a nonsense variant at the same codon that causes type 1 disease (function score = −2.53; Extended Data Fig. 10a). c.264G>A is also absent from cBioPortal ccRCCs. Together, the clinical evidence and SGE data suggest that c.264G>A may preserve some ability to regulate HIF.

We reasoned that this may be due to stop-codon readthrough (SCR). c.264G>A creates an opal codon, in context–5′-UGAC. Opal codons followed by pyrimidines are the most permissive to SCR[42], a trend reflected in function scores (Extended Data Fig. 10b,c). We introduced c.264G>A by editing and observed faint residual expression of VHL by western blot (Fig. 6e). HIF1A expression was upregulated compared to controls, but to a lesser extent than in cell lines harboring c.263G>A or a frameshifting InDel.

To specifically assess SCR, we used a flow cytometry-based assay[43] to compare c.264G>A to other *VHL* nonsense variants. Indeed, c.264G>A led to substantial readthrough, whereas c.263G>A did not (Fig. 6f and Extended Data Fig. 10d–f). We also tested c.351G>A, a second variant that creates the same 4-bp stop-codon context as c.264G>A. Likewise, considerable readthrough was detected for c.351G>A but not c.350G>A, albeit less than observed for c.264G>A, consistent with c.351G>A having a lower function score. These experiments indicate that differences in residual expression of *VHL* nonsense variants can affect the degree of functional impairment. More broadly, these examples illustrate how highly accurate functional measurements reveal mechanisms underlying complex genotype–phenotype relationships.

## Discussion

Here we applied a highly optimized SGE protocol to quantify the effects of nearly all SNVs across the complete coding sequence of *VHL*. Variant effects were dependent on HIF1A and predicted pathogenicity with high accuracy. Combined with human phenotypic data, the SGE data constitute a mechanistically informative variant effect map of *VHL*. Data for *VHL* SNVs scored by SGE are available to search and explore in relation to protein structure, computational predictors and disease association via https://vhl-board.onrender.com.

Despite improvements to the SGE protocol that promise to make the method more broadly applicable, limitations remain. Even with substantial optimization, we were unable to confidently score many variants introduced to the GC-rich, 5′-region of exon 1. However, the fact that no variants tested before p.M54 scored as depleted suggests this region harbors few, if any, SNVs of clinical importance.

Only a small fraction of pathogenic SNVs in ClinVar could not be distinguished from neutral variants using SGE data alone. The fact that none of these variants were observed in ccRCC suggests some fraction may be misclassified. However, type 2 VHL disease variants have been linked to HIF-independent effects that may be important for pheochromocytoma development[44,45]. Such effects were not represented in our assay, as function scores were dependent on HIF1A expression. Despite this, the degree of functional impairment measured was highly predictive of specific features of VHL disease.

Overall, the high accuracy with which function scores distinguish pathogenic variants indicates the data may be used as strong evidence

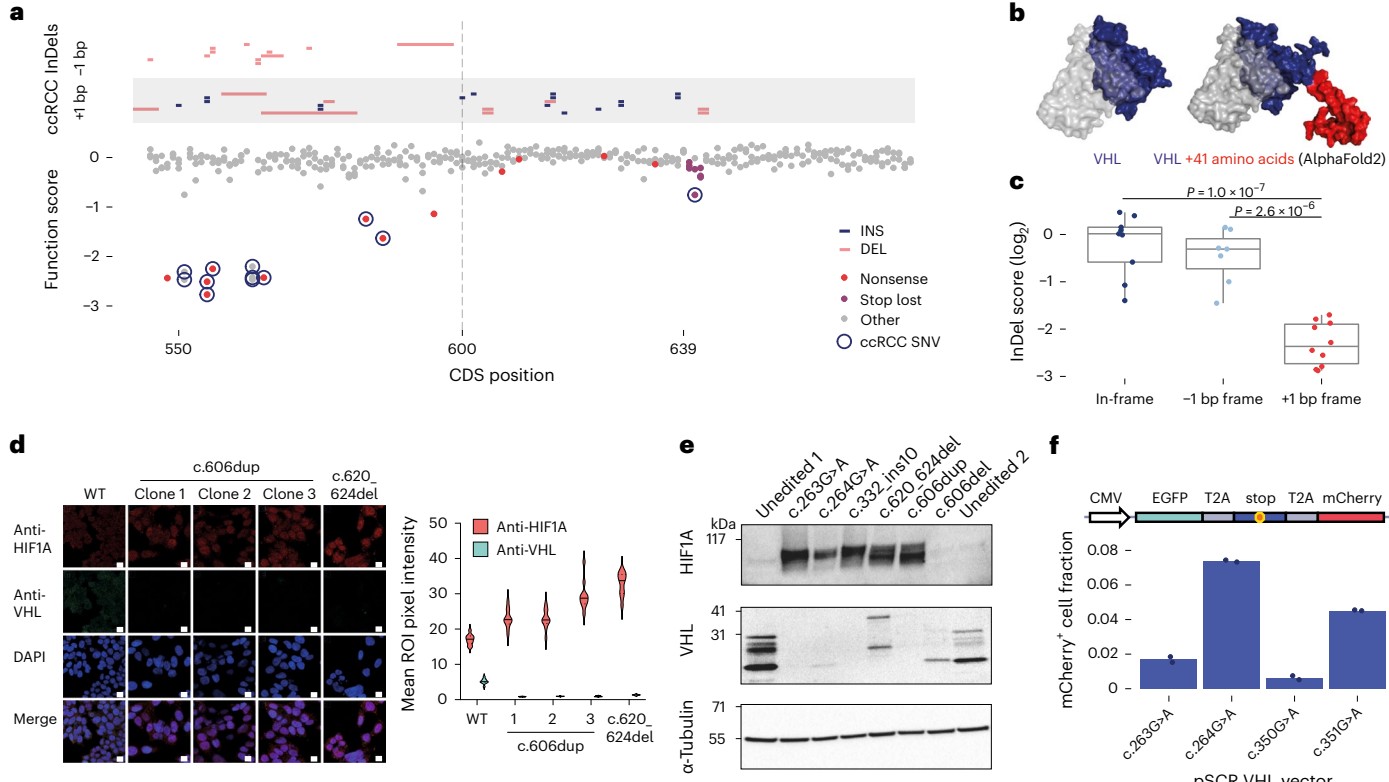

**Fig. 6 | Insights into mechanisms underlying genotype–phenotype associations. a**, Frameshifting InDels present in cBioPortal ccRCC samples are grouped by reading frame and plotted above SGE function scores in the 3′-most region of exon 3. From p.R200, all frameshifting InDels seen in ccRCC result in the same reading frame, which extends the protein by 41 amino acids. **b**, The crystal structure of VHL in complex[58] is shown next to an AlphaFold prediction[59] of the full-length VHL protein plus the 41-amino acid C-terminal extension common to InDels observed in ccRCC. **c**, InDel scores for CRISPR-induced edits located between p.R200 and p.*214 are plotted, with InDels grouped by reading frame. $n$ = 10 InDels resulting in a +1 bp frameshift scored significantly lower than $n$ = 9 in-frame InDels and $n$ = 7 −1 bp frameshifting InDels (one-way ANOVA:

$P = 1.0 \times 10^{-7}$ and $P = 2.6 \times 10^{-6}$, respectively; boxplot center line, median; box limits, upper and lower quartiles; whiskers, 1.5× interquartile range). **d**, Clonal HAP1 lines harboring variants leading to the 41-amino acid C-terminal extension (c.606dup and c.620_624del) were stained for endogenous expression of VHL and HIF1A and imaged using confocal microscopy. (Data are representative of two independent stains; scale bar, 10 μm; ROI, region of interest.) **e**, A western blot was performed to assess VHL and HIF1A protein expression in clonally isolated HAP1 lines harboring specific variants. (Results are representative of two independent blots.) **f**, A dual-fluorophore SCR reporter was used to quantify the readthrough of nonsense variants as the proportion of mCherry+ cells by flow cytometry. Points show each of $n$ = 2 replicate transfections.

to support the classification of germline variants[11]. With the recently proven efficacy of the HIF2α-inhibitor belzutifan for the treatment of VHL disease[14], accurate classification of pathogenic variants will facilitate access to better care. We anticipate function scores may also help stratify patients with VHL disease by the risk of specific tumors, as we have shown for ccRCC. Closely monitored cohorts will be valuable for exploring additional uses of the data, for instance, predicting tumor features, cancer progression, overall disease burden and treatment response.

We also envision our data being useful for adjudicating somatic *VHL* variants observed in tumors, especially in light of ongoing trials of belzutifan for sporadic ccRCC[46]. Across all cancers, low-scoring variants were predominantly found in tumors previously associated with *VHL* mutations (Fig. 4b–d). Although *VHL* mutations occur much less frequently in renal cancers other than ccRCC, our data support their functional significance when present, a finding that suggests molecular profiling may identify additional RCCs that would respond favorably to treatments targeting HIF signaling. Conversely, one in ten *VHL* variants deemed 'oncogenic' in cBioPortal scored neutrally by SGE, including several SNVs found in ccRCC. In these cases, treatments targeting HIF may provide less benefit.

The RNA scores determined by SGE provide a means of examining the interplay between dosage effects at the mRNA level and functional output. This is an important relationship in the context

of *VHL* because variants leading to complete loss of protein activity cause type 1 VHL disease. We show relatively few coding variants reduce mRNA dosage enough to impair VHL function in the assay. While we cannot preclude the clinical significance of variants impacting *VHL* mRNA levels to a lesser extent, particularly as they relate to recessive diseases such as congenital polycythemia[22,29], this finding suggests only noncoding variants with large effects on mRNA expression should warrant suspicion for a dominantly inherited cancer predisposition syndrome. RNA scores may provide additional value for identifying rare variants contributing to recessive phenotypes, such as c.222C>A. This synonymous SNV linked to VHL deficiency[23] had a marginally low function score of −0.40 but was highly depleted in mRNA (RNA score = −4.5), consistent with its documented effect on splicing.

We failed to score several variants causing recessive polycythemia as depleted by SGE, consistent with the functional impairment caused by these variants being insufficient to predispose patients to tumor development. This includes recently described variants near the pseudoexon region of intron 1 (Supplementary Table 3), where we observed no LoF variants. This result suggests there are unlikely to be many undetected variants causing VHL disease or sporadic ccRCC in this region, but we cannot rule out the possibility that our cell model is inadequate for studying pseudoexon inclusion, as effects on splicing may differ between cell types.

Collectively, this study and other recent implementations of SGE[31,47,48] show how relatively simple assays reflective of cell-intrinsic effects can identify variants driving human disease with high accuracy. Our analysis also highlights how orthologous lines of evidence can be leveraged to minimize classification errors, as evidenced by the perfect concordance between function scores and gold-standard ccRCC classifications (Fig. 4f,g).

For *VHL*, the high predictive power of SGE stems from the accurate detection of LoF variants that cause reduced growth via upregulation of HIF1A. VHL's ability to regulate HIF is crucial to its tumor suppressor function across tissues[49], making the SGE phenotype relevant to cancer despite VHL loss leading to reduced HAP1 growth. It is notable that HAP1 cells do not express *EPAS1* (ref. 50), which encodes HIF2A. The roles of HIF1 and HIF2 are thought to be opposing in ccRCC development, with the former acting as a tumor suppressor and the latter promoting growth in vivo[51–53]. While our study does not address why *VHL* loss and subsequent HIF upregulation promote growth specifically in cells that give rise to ccRCC, the hypoxic environment of the kidney and renal lineage-specific transcription factors are thought to have important roles[54,55]. Therefore, a key challenge going forward will be to extend SGE to more cell types and assays.

With continued genomic profiling of patients and tumors and the growing use of multiplexed assays to systematically study variants[56], our ability to map genotypes to phenotypes across a wide spectrum of functional effects will continue to improve. In this context, we anticipate that this analysis of *VHL* will prove highly valuable for clinical variant interpretation while also guiding future efforts to elucidate complex genotype–phenotype relationships across additional genes.

## Online content

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

[1]The Genome Function Laboratory, The Francis Crick Institute, London, UK. [2]Renal Division, Department of Medicine, Medical Center—University of Freiburg, Faculty of Medicine, University of Freiburg, Freiburg, Germany. [3]Genetics and Genome Biology Program, The Hospital for Sick Children, Toronto, Ontario, Canada. [4]The Cancer Dynamics Laboratory, The Francis Crick Institute, London, UK. [5]Renal and Skin Units, The Royal Marsden Hospital, London, UK. [6]Melanoma and Kidney Cancer Team, The Institute of Cancer Research, London, UK. [7]Scientific Computing, The Francis Crick Institute, London, UK. [8]Department of Molecular Genetics, University of Toronto, Toronto, Ontario, Canada. [9]Division of Cancer Research, Department of Thoracic Surgery, Medical Center—University of Freiburg, Faculty of Medicine, Freiburg, Germany. [10]German Cancer Consortium (DKTK), Partner Site Freiburg, A Partnership Between DKFZ and University Medical Center Freiburg, Freiburg, Germany. [11]Department of Physiology, University of Toronto, Toronto, Ontario, Canada. [12]These authors contributed equally: Athina Ganner, Laura Cubitt, Reid Brewer, Dong-Kyu Kim. ✉e-mail: greg.findlay@crick.ac.uk

## Methods

### Ethics and consent

This study complies with all relevant ethical regulations. Use of anonymized data from the Freiburg VHL Registry was approved by the ethics committee of Freiburg University (EK-FR 79/20), and all patients provided written informed consent.

### Homology-directed DNA repair (HDR) library design and cloning

SGE experiments were designed as previously[31] using *VHL* transcript ENTS000000256474.3 (CCDS2597). Six SGE regions were designed to cover the entire coding sequence, including exonic sequences and exon-proximal regions of introns. An additional SGE region was designed in intron 1 to cover positions associated with splicing alterations[29].

Oligonucleotide libraries were designed for each SGE region. First, synonymous substitutions were designed at two CRISPR protospacer adjacent motif (PAM) sequences to prevent Cas9 recutting following HDR and to distinguish HDR-derived SNVs in NGS. These substitutions were included in template sequences to which SNVs were introduced (Supplementary Table 4). An unedited genomic sequence was appended to facilitate PCR amplification and cloning. The final libraries contained molecules representing all SNVs within each SGE region, spanning regions of hg38 chromosome 3 from 10,141,841–10,142,202, 10,142,743–10,142,876, 10,146,499–10,146,644 and 10,149,760–10,150,002.

For two SGE regions (exon 1-5′ and exon 1-3′), additional synonymous substitutions were added to reduce high-GC content in a second library. All oligonucleotide libraries were synthesized (Twist Bioscience) in a pool and resuspended at 5 ng μl$^{-1}$. Primers complementary to appended sequences were used to amplify SGE region-specific oligos with KAPA HiFi ReadyMix (Roche) in 25 μl reactions with 500 pg of oligonucleotide pool as template. All PCR reactions were monitored in real-time by spiking SYBR green (Thermo Fisher Scientific) into reactions at 0.4× final concentration, and cycling was stopped upon amplification.

For each SGE region, a homology arm plasmid was generated by PCR amplification of HAP1 gDNA and cloning of products into linearized pUC19 (InFusion, Takara Bioscience). Homology arms were 200–1,300 bp. Homology arm plasmids were subsequently linearized by inverse PCR using primers with 15–20 bp of overlap with amplified oligo libraries and 10 pg of template per 50 μl reaction. Products were DpnI-digested.

Amplified oligo pools and PCR-linearized homology arm vectors were purified using AMPure XP (Beckman Coulter). To generate final HDR libraries, amplified oligo libraries (50 ng) were InFusion-cloned into linearized pUC19-homology arm vectors (65 ng). The resulting products were transformed into stellar-competent *Escherichia coli* (Takara Bioscience). In total, 1% of transformed cells were plated on ampicillin to ensure adequate transformation efficiency (at least tenfold library coverage), and the remaining transformants were cultured overnight at 37°C in 150 ml of luria broth with carbenicillin (100 μg ml$^{-1}$).

### CRISPR gRNA and pegRNA cloning

Target sites for *Streptococcus pyogenes* Cas9 were designed for each SGE region to cleave within the coding sequence such that synonymous substitutions could disrupt re-editing. gRNAs were cloned into pX459 as described[60], including PlasmidSafe DNase (Lucigen) treatment. Products were transformed into stellar-competent *E. coli*. Sequence-verified plasmids were purified with the ZymoPure Maxiprep Kit (Zymo Research) and eluted in nuclease-free water (Invitrogen).

For epegRNA plasmid construction, eBlocks were purchased from Integrated DNA Technologies. The pU6-tevopreq1-PuroR vector was linearized by inverse PCR and gel-extracted using the GeneJet Gel Extraction Kit (Thermo Fisher Scientific). Each epegRNA was integrated into the linearized vector using NEBuilder HiFi Assembly (NEB), and the product was transformed into One Shot TOP10 competent cells (Thermo Fisher Scientific). Plasmids were verified by Sanger sequencing.

### Tissue culture: subculture routine

To perform SGE, HAP1 cells with a *LIG4* frameshifting deletion (HAP1-LIG4-KO)[31] were cultured with Iscove's modified Dulbecco's medium containing L-glutamine and 25 nM HEPES (Gibco) with 10% FBS (Gibco) and 1% Penicillin–Streptomycin (Gibco). Cells were thawed 1 week before transfection and maintained under 80% confluency. For each passage, cells were washed twice with 1× DPBS (Gibco), trypsinized with 0.25% trypsin–EDTA (Gibco), resuspended in media, centrifuged at 300*g* for 5 min and resuspended. At least 6 million cells were split, either 1:5 across 2 days or 1:10 across 3 days. Apart from initial SGE experiments without DAB, 2.5 μM DAB (Stratech) was added to the media each passage.

### Tissue culture: SGE experiments

One day before transfection, 15 million cells were seeded on a 10-cm dish in 10 ml of media. For each SGE region, cells were cotransfected with 30 μg of pX459 plasmid expressing a gRNA and 10 μg of HDR library. Xfect (Takara Bioscience) transfection reagent was used according to the manufacturer's protocol except for the following modifications: at transfection (day 0), cells were 80–90% confluent, 0.6 μl of Xfect polymer was used per microgram of DNA, 40 μg of total plasmid DNA was transfected per 10-cm dish and the final volume of transfection buffer, DNA and Xfect polymer was 800 μl.

Two replicate transfections were performed per SGE region. On day 1, cells were washed and transferred to a 15-cm dish with puromycin (Cayman Chemical) added to the media at 1 μg ml$^{-1}$. During puromycin selection, DAB was not added. On day 4, the cells were passaged to media with DAB and without puromycin. Cells were collected on days 6, 13 and 20, taking at least 10 million cells as pellets stored at −80°C.

Negative control samples for each SGE region were cotransfected with HDR library and a pX459 plasmid containing a gRNA targeting *HPRT1*. The same transfection conditions were used but scaled proportionally to a six-well plate. Negative control samples were collected on day 6.

### Tissue culture: generation of HAP1–HIF1A–KO cells

HAP1-LIG4-KO cells were transfected with pX459 targeting *HIF1A*. Cells were diluted to 0.8 cells per 100 μl and aliquoted into a 96-well plate. Light microscopy was used to identify wells with a single colony. gDNA was extracted from clones (DNeasy Blood & Tissue Kit; Qiagen), PCR-amplified and Sanger-sequenced. A HIF1A-knockout clone containing a 7-bp deletion was selected as HAP1–HIF1A–KO.

### Tissue culture: essentiality testing

To test essentiality, a pX459 vector expressing a gRNA targeting *VHL* exon 2 was transfected into HAP1-LIG4-KO (day 0) using 5 μg of DNA per well of a six-well plate in replicate. Samples were collected on days 6 and 13.

### gDNA and RNA extraction

Collected cell pellets were thawed. QIAshredder columns (Qiagen) were used to homogenize cells, and the AllPrep DNA/RNA Kit (Qiagen) was used to purify RNA and gDNA, as per manufacturer protocol. Yields were measured using Nanodrop UV spectrometry and the Qubit BR DNA Kit (Thermo Fisher Scientific).

### RNA preparation for sequencing

cDNA was generated from 5 μg RNA per sample using SuperScript IV First-Strand Synthesis (Invitrogen). A *VHL*-specific primer

complementary to the 3′-UTR was used for priming. cDNA samples were subsequently prepared for sequencing starting with PCR2.

### PCR1: amplifying gDNA

PCR primers are provided in Supplementary Table 4. Per PCR, 2 µg of gDNA was amplified using at least one primer binding outside homology arm regions. Annealing temperatures and cycling times were optimized using gDNA from unedited HAP1. Up to eight 100 µl reactions were performed per SGE sample using KAPA HiFi 2× ReadyMix (Roche) with magnesium chloride added to 5 mM. Reactions for each sample were pooled and purified using AMPure XP (Beckman Coulter). The same procedure was carried out for negative controls and SGE samples from each timepoint.

### PCR2: adding Nextera adapters

A nested PCR for each sample was performed using 1 µl of purified product from PCR1. Primers with Nextera sequencing adapters were designed for each SGE region to produce amplicons such that 300-cycle Illumina kits would provide full-length coverage. Products were verified using gel electrophoresis and purified with AMPure XP (Beckman Coulter). The same reaction was performed to prepare HDR libraries for sequencing. cDNA samples were similarly amplified using cDNA-specific primers to yield amplicons spanning SGE regions and at least one exon junction.

### Indexing and sequencing

Each sample was dual-indexed by PCR using custom indexes and purified using AMPure XP. Samples were quantified using Qubit HS (Invitrogen). Illumina protocols were followed to dilute and denature samples before sequencing on an Illumina NextSeq (mid- or high-output 300-cycle kits). Approximately 5 million reads were allocated per SGE gDNA sample, except for samples from high-GC regions (exon 1–5′ and exon 1–3′), which were allocated 10 million reads. In total, 1 million, 2 million and 3 million reads were allocated for the negative control, library and RNA samples, respectively. PhiX (20–30%; Illumina) was included in sequencing.

### Deriving function scores for SNVs

A pipeline to process sequencing data to variant counts was used. Briefly, paired-end reads were adapter-trimmed and merged if the overlapping sequence matched perfectly (SeqPrep). Merged reads containing N bases were removed before alignment with needleall (EMBOSS v6.6.0.0) against a reference amplicon for each SGE region. The resulting SAM files were processed using custom scripts to calculate SNV frequencies and annotate variants with data from CADD v1.6 (hg19)[25]. For calculating function scores, reads for each variant were only included if at least one silent PAM edit was present.

In a series of quality filters, SNVs were removed from analysis if any of the following were true: SNV HDR library frequency was less than $1.0 \times 10^{-4}$; SNV day 6 frequency in either replicate was less than $1.0 \times 10^{-5}$; SNV day 20 to day 6 $\log_2$ ratios were highly discordant across replicates (that is, a difference greater than 1.5, unless both less than −1.0); SNV day 13 to day 6 $\log_2$ ratio differed markedly from the day 20 to day 13 $\log_2$ ratio (that is, a difference greater than 2.0, unless both less than −0.5); estimates of sequencing error suggested the ratio of SNV observations arising from error was greater than 0.5. Finally, where a variant was engineered in the same codon as a synonymous PAM edit, the variant was excluded if the resulting amino acid change was different from what it would be without the PAM edit present.

To calculate function scores, the mean day 20 to day 6 $\log_2$ ratio for each variant across replicates was normalized to the sample's median synonymous variant. Final function scores were then normalized across exons using the range of effect sizes observed for synonymous and nonsense variants. Scores per region were scaled linearly such that the median nonsense variant for the region equaled the global median nonsense variant and the median synonymous variant scaled to 0. (Only nonsense variants between p.54 and p.198 were used for scaling.)

### Calculating RNA scores

RNA scores were determined for exonic variants using variant frequencies in targeted sequencing of cDNA. Variant frequencies were calculated only from reads in which both the variant and at least one PAM edit indicative of HDR were present, and the sequence on either side of the target exon matched the reference transcript perfectly. RNA scores for day 6 and day 20 were derived by normalizing each coding SNV's frequency in RNA to its frequency in the corresponding gDNA sample. Noncoding variants were not assigned RNA scores because such variants are not detectable in spliced transcripts. RNA scores were also not generated for variants in the exon 1–5′ SGE region. RNA scores were only assigned to SNVs that passed quality filtering for function scores.

### Producing a single score set

For variants covered by overlapping SGE regions, final function scores and RNA scores were determined as follows: for variants in both exon 1–5′ and exon 1-mid, the scores from the exon 1-mid experiment were used; for variants in both exon 1-mid and exon 1–3′, the mean of the scores was used; for variants in both exon 3–5′ and exon 3–3′, the mean of the scores was used.

### Analysis of InDels

To analyze InDel selection, CIGAR strings from each timepoint were generated via global alignment. 'InDel scores' for each CIGAR string were defined as the $\log_2$ ratio of day 13 frequency to day 6 frequency, averaged across replicates. Only editing outcomes observed in more than 0.1% of reads with at most one InDel and 100 bp of matching sequence on the 5′ end of the alignment were included.

### Statistics and reproducibility

Statistical tests were performed in R (v3.6.3) using RStudio (v1.4.1106). Unless indicated, all tests were two-sided. Sample sizes were determined by including all SNVs assayed across SGE regions provided they passed quality control. All analyses excluded no variants, samples or patients unless explicitly indicated. No blinding or randomization was performed.

To determine function score significance, a null model (assumed to be normal but not formally tested) was fit for each SGE region using all synonymous variants. Where RNA scores were ascertained, synonymous variants were excluded from null if their day 6 RNA score was less than −1.0. For experiments without RNA scores, all synonymous SNVs were included, and for the intron 1 SGE region, all SNVs were included in the null model. $P$ values were calculated using the 'pnorm' function in R and adjusted using 'p.adjust' (Benjamini–Hochberg) to produce $q$ values. An FDR of 0.01 was used to define significant scores.

### DepMap analysis

CRISPR screening data for *VHL* was downloaded via https://depmap.org/portal/. 'Primary.Disease' was used to define lines of kidney origin, and lines with *VHL* mutations defined as 'Hotspot', 'Damaging' or 'Other nonconserving' were labeled mutant for *VHL*.

### Structural analysis

Exposure labels defining missense mutations as 'protein core', 'interaction core', 'interaction periphery' and 'surface' were obtained[61]. FoldX[35] 5.0 was downloaded (https://foldxsuite.crg.eu/) and run locally to calculate ΔΔG predictions. For this task, the VHL protein structure in complex with ELOC, ELOB and the HIF1A peptide (Protein Data Bank (PDB): 1LM8)[58] was repaired using 'RepairPDB' before running 'PositionScan' to calculate ΔΔG values for all missense substitutions from p.R60 to p.Q209.

For visualizing function scores in relation to structure, average function scores of missense SNVs at each residue from p.M54 were calculated, excluding variants with RNA scores below −2.0, and mapped to the VHL structure (PDB: 1LM8) in PyMol v2.5.4. AlphaFold's colab notebook was used to model a structure for VHL with the C-terminus 41-amino acid extension (https://colab.research.google.com/github/deepmind/alphafold/blob/main/notebooks/AlphaFold.ipynb).

### ClinVar and cBioPortal analyses

*VHL* entries with at least a one-star assertion criteria rating in ClinVar were obtained on 4 May 2023. Variants classified as both 'pathogenic' and 'likely pathogenic' were labeled 'likely pathogenic', and variants deemed both 'benign' and 'likely benign' were labeled 'likely benign'.

cBioPortal data were accessed on 2 October 2022. Mutation data from the 'curated set of nonredundant studies' were exported by querying samples with *VHL* mutations. Data were parsed to sum the number of times each variant was seen across each cancer type. For analyses involving allele frequencies in tumors, each independent sample was plotted. ccRCC samples 'with sarcomatoid features' were grouped with all ccRCC samples. To assign a single cancer type for coloring variants in multiple cancers, the following order of preference was used: ccRCC, pheochromocytoma, PNET, chromophobe RCC, papillary RCC, RCC not otherwise specified and other.

### Population sequencing analysis

UKB variants in *VHL* were ascertained via GeneBass, which includes data from 394,841 individuals (7 June 2022 release). TOPMed (freeze 8) *VHL* variants were obtained, as were nonoverlapping gnomAD v2 and v3 datasets. A combined population allele count was defined as the sum of gnomAD v2 and v3 allele counts, TOPMed heterozygous allele counts and UKB allele counts.

### Defining function classes

We derived four function classes for variants scored by SGE. First, variants with $q$ values greater than 0.10 and function scores greater than −0.2188 (that is, greater than the fifth percentile of gold-standard neutral variants) were deemed 'neutral'. Second, variants with function scores lower than −1.26 (that is, lower than the 95th percentile of gold-standard ccRCC variants) were deemed 'LOF1'. Third, variants with $q$ values less than 0.01 and function scores less than −0.3875 (the lowest scoring gold-standard neutral variant) were deemed 'LOF2'. All remaining variants were deemed 'Intermediate'.

### VHLdb analysis

Mutation data from VHLdb[21] were downloaded (28 July 2022) and parsed to count disease features of entries for each variant. Where there was only one possible match to SGE data, the nucleotide change was inferred from the amino acid substitution if only the latter was provided. When the provided nucleotide and protein changes were discordant, the entry was removed, as were entries flagged 'needs revision'.

'Pathogenic' and 'likely pathogenic' variants in ClinVar, excluding variants with recessive phenotypes and those seen in two or more population control individuals, were assigned to categories of 'type 1', 'pheo-predominant' or 'type unclear' by VHLdb disease type (if provided) and/or associated tumors. Variants only associated with type 1 disease or ccRCC were deemed 'type 1'. Variants associated only with type 2 disease or with more pheochromocytoma entries than ccRCC entries were deemed 'pheo-predominant'. The remaining variants either had mixed associations, lacked phenotypic information or were explicitly type 2B variants. All were deemed 'type unclear'.

Curated clinical phenotype data were obtained for benchmarking[61], including the number of kindreds analyzed, the occurrence of ccRCC and pheochromocytoma and the type of VHL disease. Variants in at least two kindreds were included, excluding those seen more than once in combined population sequencing unless annotated as type 2.

### Comparisons to computational predictions

CADD scores (v1.6)[25] were obtained for SNVs. The following scores were obtained for missense variants: REVEL scores[5] (1 September 2022), boostDM scores for *VHL* variants in the 'Renal Clear Cell Carcinoma' model (12 August 2022), EVE scores (15 August 2022) and VARITY scores (16 August 2022). For comparisons between scores, only variants for which all metrics provided a score were used. This excluded variants before p.M54 and variants after p.A207 due to a lack of EVE scores. Receiver operating characteristic curves were generated in R using the 'geom_roc' function.

SpliceAI scores for each SNV were obtained among CADD annotations. A single SpliceAI score per variant was defined as the maximum score among independent scores for 'acceptor gain', 'acceptor loss', 'donor gain' and 'donor loss'.

### Clinical study design and statistical analysis

The Freiburg VHL Registry includes patients screened at least once until 2023 at the von Hippel–Lindau Outpatient Clinic of the University Medical Center Freiburg. As of 1 January 2024, the Freiburg VHL Registry included 552 participants with data on ccRCC status. The inclusion criterion for this retrospective analysis was the detection of a *VHL* germline mutation (class 4 or 5 according to the American College of Medical Genetics and Genomics and the Association for Molecular Pathology). Patients lacking clinical data were excluded. In total, 375 (67.9%) patients had a *VHL* mutation classified as LOF1 or LOF2 by SGE (mean age ± s.d. = 45.5 years ± 17.6, 52% female). In total, 122 participants had LOF1 mutations (age: 41.4 ± 14.3 years, 47.5% female). In total, 253 participants had a LOF2 mutation (age: 47.5 ± 18.7 years, 54.2% female). In total, 46 different LOF1 mutations and 11 different LOF2 mutations were present.

Clinical data including age, gender and diagnostic results were recorded in a predefined database. Surveillance was performed according to VHL disease guidelines (VHL Active Surveillance Guidelines) and included a magnetic resonance imaging or computed tomography scan of the abdomen. For ccRCC, the first radiologic description was considered the initial diagnosis. Registrants were grouped by the function class of their variant (LOF1 or LOF2). The Kaplan–Meier estimator was used to assess age-related ccRCC penetrance. Registrants without ccRCC were censored at the age of their last visit. Log-rank tests were used for pairwise comparisons of age-related penetrance curves using GraphPad Prism 10.1.2.

### Generation of clonal lines with VHL variants

To engineer SNVs via HDR, oligonucleotides containing c.264G>A and c.606dup were ordered for InFusion cloning into homology arm vectors including synonymous edits at PAM sites. HAP1-LIG4-KO cells were transfected with pX459 and each vector in six-well plates. On day 6 post-transfection, cells were split by limiting dilution into 96-well plates. Individual clones were sequenced to identify correctly edited lines. Additional lines harboring unintended edits were isolated as controls, including a negative control line with only a synonymous PAM edit, a line with a 10-bp insertion (c.332_333_insCTACCGAGGT, abbreviated 'c.332ins10') and a line with a frameshifting deletion (c.620_624del).

Additional lines were created by prime editing in HAP1 *MLH1* knockout cells (Horizon Discovery, HZGHC000343c022). On day 0, 0.5 µg of each epegRNA and PEmax plasmid (Addgene, 180020) were cotransfected into 90,000 cells per well in a 12-well plate using FuGENE HD (Promega). On day 1, cells were treated with puromycin (Thermo Fisher Scientific) at 2 µg ml⁻¹ and maintained for 2 days. On day 3, puromycin was removed and cells were incubated for one more day before being sorted into a 96-well plate using a Sony MA900 Cell Sorter. After 12 days, gDNA was extracted and Sanger-sequenced. Lines were verified to harbor SNVs without additional edits, including c.228C>G, c.222C>A, c.191G>C, c.292T>C, c.334T>C, c.371C>A,

c.462A>C, c.539T>A, c.329A>C, c.302T>C, c.458T>C, c.263G>A and c.606del.

## Western blots

To assess VHL and HIF1A protein expression, 0.3 million cells were seeded in six-well plates. After 2–3 days, cells were lysed in RIPA buffer (50 mM Tris–HCL at pH 7.4, 150 mM NaCl, 1 mM EDTA, 1% sodium deoxycholate, 1% NP-40 and 1% Triton X-100) containing protease inhibitors (Thermo Fisher Scientific). Protein concentration was determined by BCA assay (Thermo Fisher Scientific). In total, 20 μg of total protein was run per well using Mini Protein Gels (Thermo Fisher Scientific) and transferred to nitrocellulose membrane using iBlot 2 Dry Blotting System (Thermo Fisher Scientific). The membrane was blocked with a blocking solution containing 5% skim milk in Tris-buffered saline with Tween-20 (TBST; 100 mM Tris–HCl, 150 mM NaCl, 0.1% Tween-20, pH 7.5) at room temperature for 1 h. The membrane was then incubated at 4 °C overnight with primary antibodies including mouse tubulin antibody (Sigma-Aldrich, T6199; 1:3,000), rabbit VHL antibody (Cell Signaling Technology, 68547; 1:1,1,000) and mouse HIF1A antibody (BD Transduction Laboratories, 610959; 1:1,000). After washing with TBST for 30 min, the membrane was incubated at room temperature for 1 h with the following secondary antibodies: goat anti-mouse IgG–HRP (Abcam, ab205719; 1:10,000) or goat anti-rabbit IgG–HRP (Sigma-Aldrich, AP307P; 1:10,000). After washing with TBST for 30 min, the membrane was treated with SuperSignal West Pico Plus Chemiluminescent Substrate (Thermo Fisher Scientific, PI34577) and visualized on ChemiDoc XRS+ system (Bio-Rad).

## Immunofluorescence microscopy

Cells plated on tissue culture-treated 35 mm imaging dishes (Ibidi) were washed with 1× DPBS (Gibco) and fixed with 10% neutral-buffered formalin (Sigma-Aldrich) for 10 min, then washed twice with ice-cold 1× DPBS. Cell permeabilization was carried out using 0.2% Triton X-100 (Thermo Fisher Scientific) for 10 min, followed by three washes with 1× PBST (1× DPBS and 0.1% Tween-20 (Thermo Fisher Scientific)). Samples were blocked by washing once and incubating in a blocking buffer (1× DPBS and 1% BSA (Sigma-Aldrich)) for 1 h. Samples were then incubated for 1 h at room temperature with primary antibodies in blocking buffer, including rabbit anti-VHL (Cell Signaling Technology, 68547; 1:200) and mouse anti-HIF1α (Novus Biologicals, NB100-105; 1:50). Dishes were washed three times with 1× DPBS for 5 min, then incubated for 1 h at room temperature with secondary antibodies in blocking buffer and then washed three times with 1× DPBS. Secondary antibodies were donkey anti-rabbit IgG Alexa Fluor 555 (Thermo Fisher Scientific, A-31572; 1:500) and goat anti-mouse IgG Alexa Fluor 647 (Thermo Fisher Scientific, A-21235; 1:500). Cells were mounted using mounting medium containing DAPI (Ibidi) and imaged using a ×40 oil objective on a point scanning confocal microscope (Zeiss LSM880). Images were analyzed using Fiji ImageJ2 (v2.14.0).

## SCR experiments

A dual-fluorophore SCR reporter cassette[43] was cloned into a pUC19 backbone with a cytomegalovirus promoter. The SCR reporter expresses a transcript encoding EGFP and mCherry, separated by a sequence of interest flanked by T2A sequences. *VHL*-derived sequences of 141 bp centered on select nonsense variants were cloned into the SCR reporter and transfected into HAP1-LIG4-KO cells. On day 5 post-transfection, EGFP and mCherry expression were assessed by flow cytometry (BD Fortessa X20) for at least 150,000 cells. The fraction of EGFP+ cells that were mCherry+ was determined (FlowJo v10.10) and normalized to a vector without a stop codon (pSCR-VHL-no-stop).

## Reporting summary

Further information on research design is available in the Nature Portfolio Reporting Summary linked to this article.

## Data availability

All function scores and RNA scores are included in Supplementary Table 1, as well as NGS read counts. Function scores are also available for visualization at https://vhl-board.onrender.com/ and have been deposited to MAVE-DB[62] (urn:mavedb:00000675-a) with RNA scores. Fastq files are publicly available (European Nucleotide Archive accession: PRJEB75229). Unprocessed western blots are included as Source Data. Structural data (PDB: 1LM8) was accessed from the PDB (https://www.rcsb.org/structure/1lm8). ClinVar, cBioPortal and VHLdb data are available at https://www.ncbi.nlm.nih.gov/clinvar/, https://www.cbioportal.org/ and http://vhldb.bio.unipd.it/, respectively. UKB, TOPMed and gnomAD data are accessible at https://app.genebass.org/, https://bravo.sph.umich.edu/freeze8/hg38/ and https://gnomad.broadinstitute.org/. CADD scores can be found at https://cadd.gs.washington.edu/download, and missense variant scores from REVEL, boostDM, EVE and VARITY are available at https://sites.google.com/site/revelgenomics/downloads, https://www.intogen.org/boostdm/search?ttype=RCCC&gene=VHL, https://evemodel.org/ and http://varity.varianteffect.org/, respectively. Source data are provided with this paper.

## Code availability

Code used in this study is available on GitHub (https://github.com/TheGenomeLab/VHL-SGE) and has been archived to Zenodo (https://zenodo.org/records/11065771)[63].

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

## Acknowledgements

We thank R. Goldstone and M. Gavrieldes for assisting with data access; J. Nicod, M. Crawford and the Advanced Sequencing STP for performing sequencing; E. Sahai and D. Balchin for helpful insights and feedback on the manuscript; A. Waters and D. Adams for experimental advice; P. Ratcliffe and lab for discussion of VHL-related research; the Cell Services STP for maintenance of cell lines; I. Toledano, B. Lehner and F. Supek for sharing materials and advice for performing SCR reporter experiments. We are also deeply thankful for H. Neumann's immense contributions to the establishment of the Freiburg VHL Registry. This work was supported by the Francis Crick Institute, which receives its core funding (to G.M.F.) from Cancer Research UK (CC2190), the UK Medical Research Council (CC2190) and the Wellcome Trust (CC2190), by a grant from Cancer Research UK (CG-MAVE to G.M.F.), and by funding from the VHL Foundation UK/Ireland (PRJ20071 to G.M.F. and S.T.). A.G. and E.N.-H. are funded by the Deutsche Forschungsgemeinschaft (DFG; German Research Foundation, Project ID 431984000-SFB 1453), and S.D. is funded by DFG (Di 1421/9-2). The funders had no role in study design, data collection and analysis, decision to publish or preparation of the manuscript. For the purpose of Open Access, the authors have applied a CC BY public copyright license to any author-accepted manuscript version arising from this submission.

## Author contributions

M.B. and G.M.F. conceived the project and designed experiments. M.B., L.C., R.B., D.K., C.M.K., N.F., P.D., J.D.J., M.M. and G.M.F. performed experiments. G.M.F., M.B., C.T., A.G., L.C., C.M.K., S.T.C.S. and C.S. analyzed data. C.T. developed the data visualization platform. A.G.,

E.N.-H. and S.D. ascertained and analyzed clinical data. G.M.F., E.A.I. and S.T. supervised the project. G.M.F. and M.B. wrote the paper with input from all authors.

## Funding

## Competing interests

The authors declare no competing interests.

## Additional information

**Extended data** is available for this paper at https://doi.org/10.1038/s41588-024-01800-z.

**Correspondence and requests for materials** should be addressed to Gregory M. Findlay.

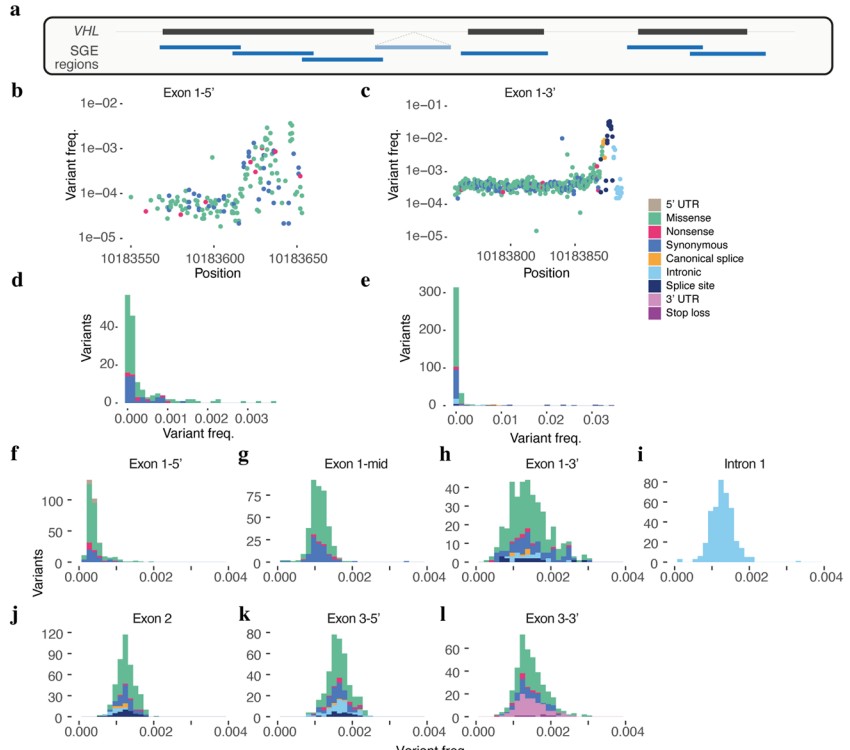

**Extended Data Fig. 1 | Optimizing SGE libraries to tile the complete *VHL* coding sequence. a**, A schematic showing the seven SGE regions tiling across *VHL*. **b**,**c**, Frequency of SNVs plotted by position in the initial libraries for exon 1–5′ (**b**) and exon 1–3′ (**c**). **d**,**e**, Histograms of variant frequency for the initial libraries for exon 1–5′ (**d**) and exon 1–3′ (**e**). Based on these distributions, additional synonymous SNVs were added to final library designs. **f**–**l**, Frequency of SNVs in the final SGE libraries used for each region.

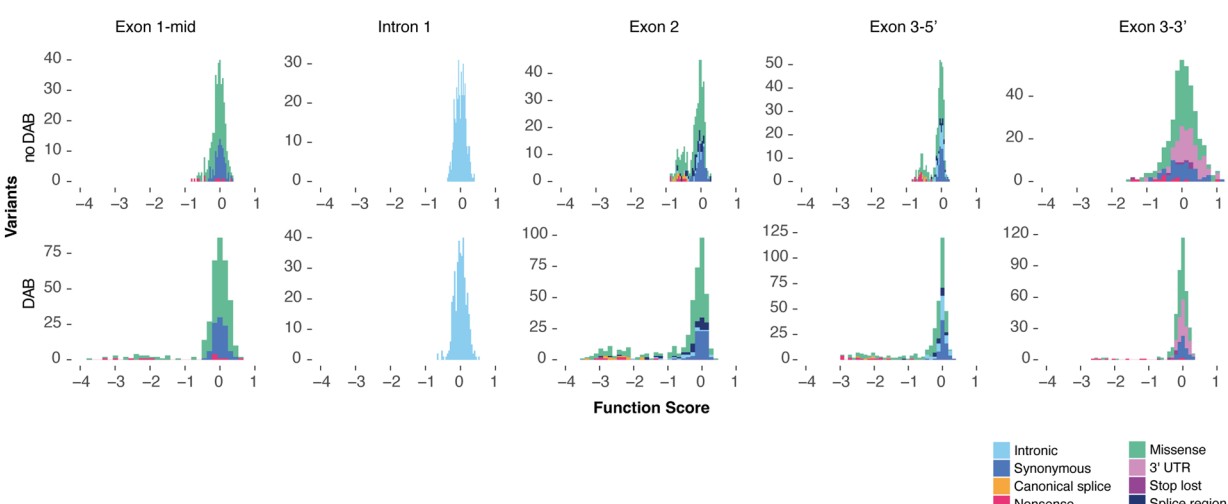

**Extended Data Fig. 2 | Addition of DAB to SGE experiments improves data quality.** Histograms of function scores for regions where SGE was performed in normal HAP1 growth media (top) and media supplemented with 2.5 μM DAB (bottom). Function scores span a greater range when derived using DAB.

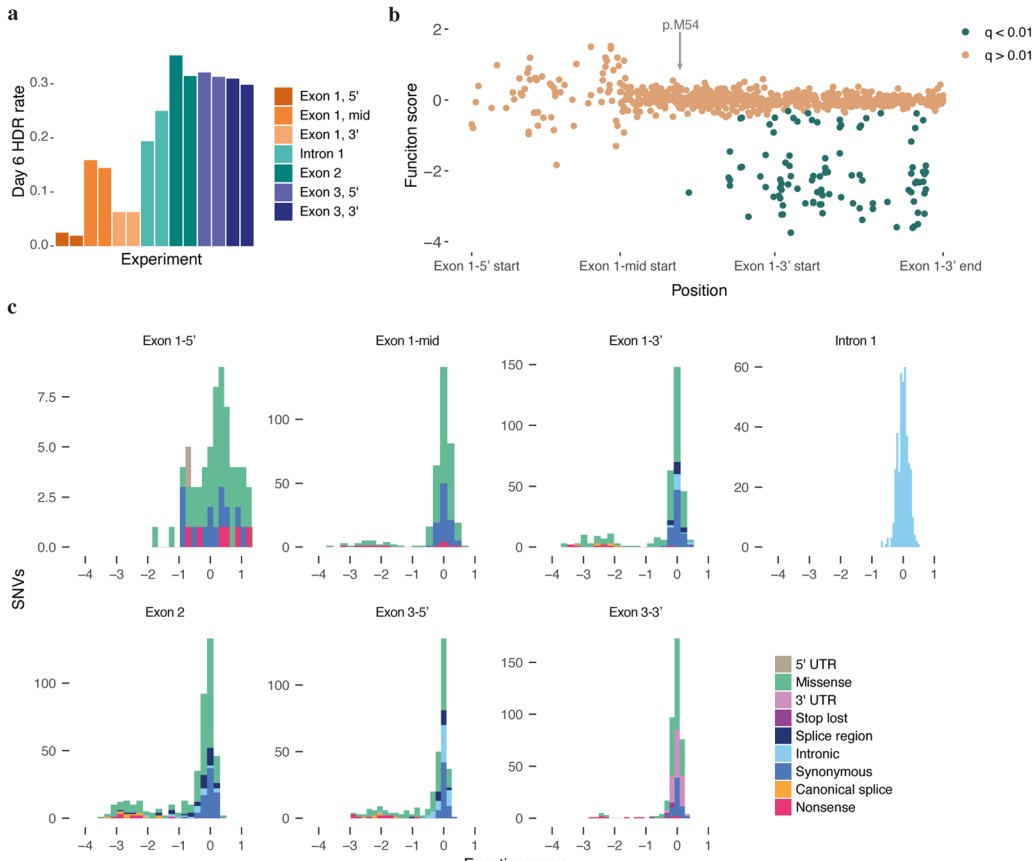

**Extended Data Fig. 3 | An absence of LoF variants in the 5' coding region of exon 1. a**, The rate of editing by HDR as measured by NGS is plotted for each replicate SGE experiment, sampled on day 6 post-transfection. **b**, Function scores for variants in exon 1 are plotted by genomic position and colored by *q* value. Positions of the three different SGE regions tiling exon 1 are indicated on the *x* axis. **c**, Histograms of function scores colored by mutation consequence are shown for each SGE region. Nonsense variants consistently score lowly across SGE regions, with the exception of the exon 1–5' region.

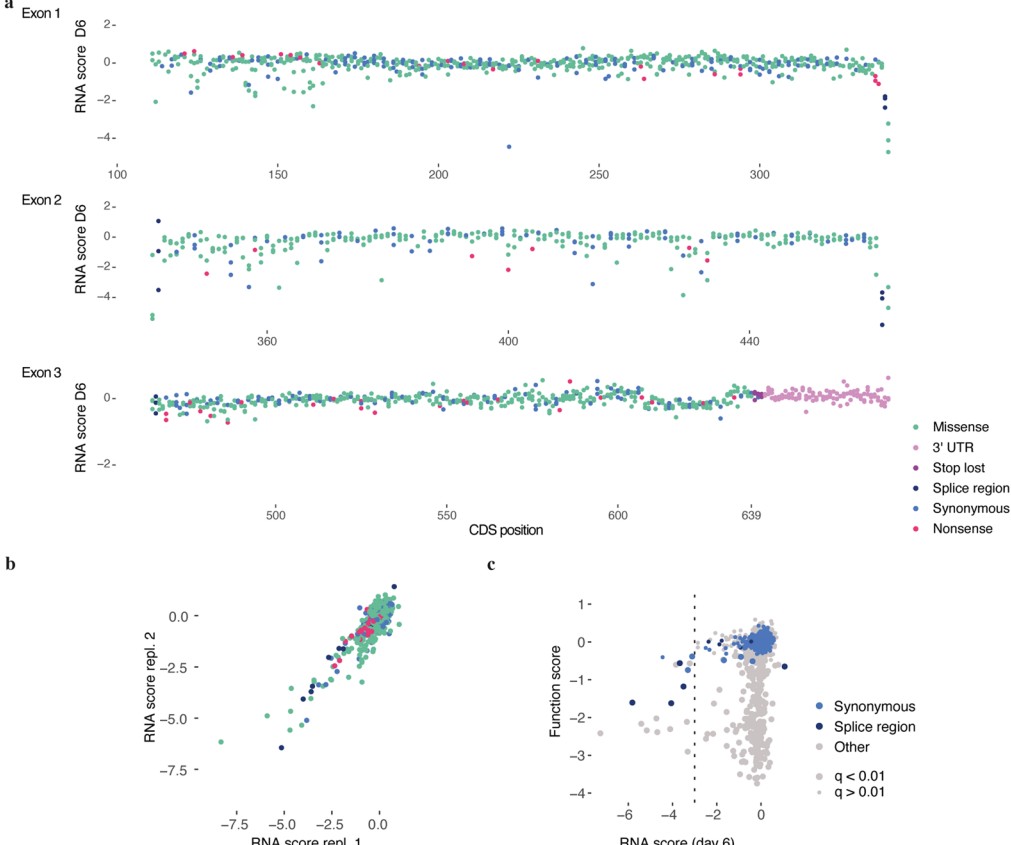

**Extended Data Fig. 4 | A map of RNA scores for *n* = 1,626 SNVs in *VHL*. a**, RNA scores, defined as each SNV's abundance in cDNA normalized to its abundance in gDNA, are plotted by transcript position. RNA scores shown are from samples collected 6 days post-transfection. **b**, Day 6 RNA scores from individual replicates are highly correlated (Pearson's *R* = 0.87). **c**, Comparison of function scores and RNA scores indicates that below an RNA score threshold of −3.0 (dashed line), 6 of 7 synonymous variants were significantly depleted.

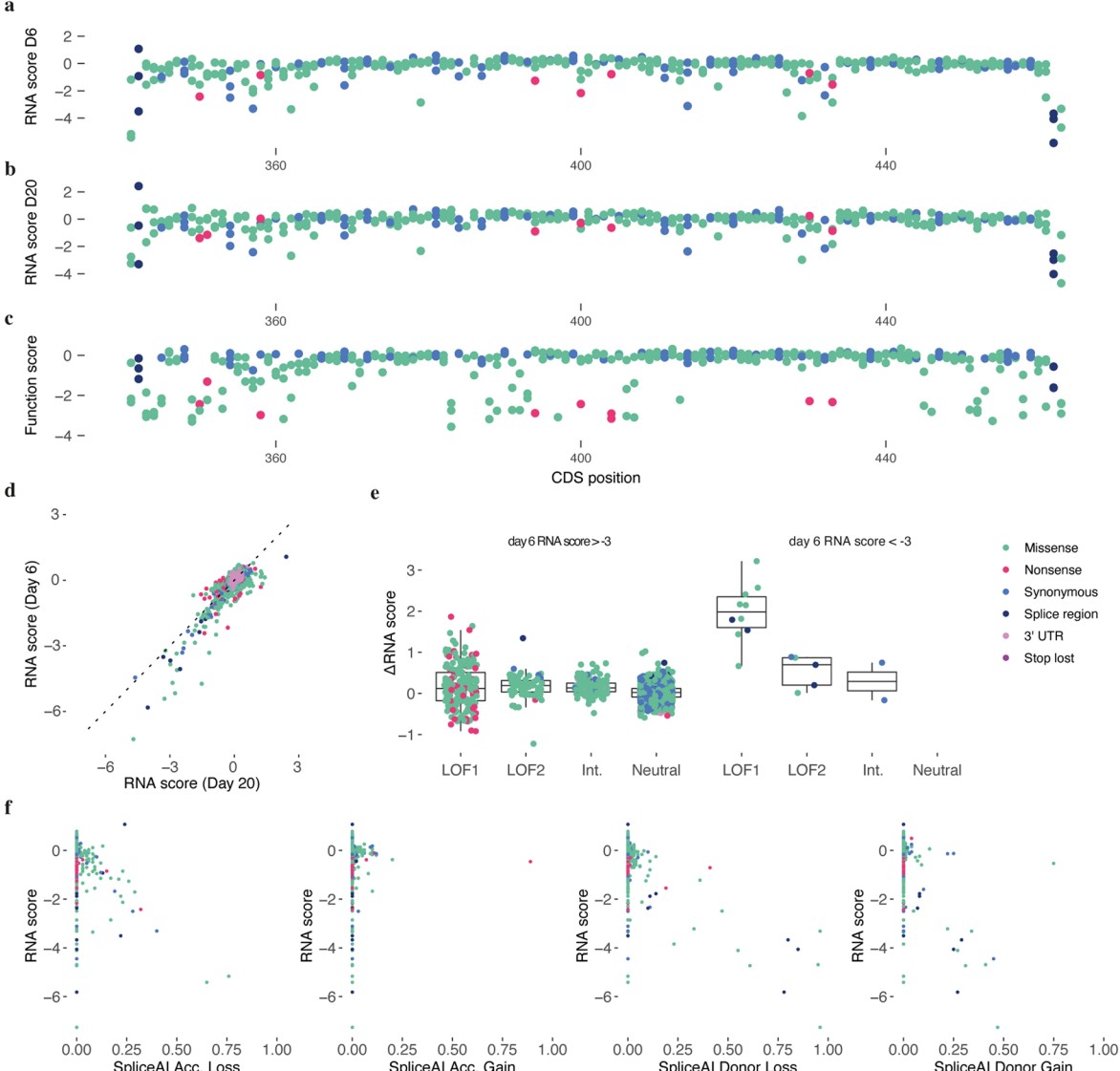

**Extended Data Fig. 5 | Comparing RNA scores across timepoints. a–c,** Scores for *n* = 356 SNVs analyzed in exon 2 are plotted by transcript position. RNA scores are plotted for samples collected 6 days post-transfection (**a**) and 20 days post-transfection (**b**). Function scores are plotted for the same set of exon 2 variants (**c**). **d,** RNA scores correlate across timepoints (Pearson's *R* = 0.86). Many variants with low RNA scores on day 6 have relatively higher RNA scores on day 20. (*y* = *x* plotted as a dashed line for reference.) **e,** The ΔRNA score for each SNV, defined as the day 20 RNA score minus the day 6 RNA score, is plotted for *n* = 184 LOF1,

*n* = 81 LOF2, *n* = 118 intermediate and *n* = 1,226 neutral SNVs. Variants with day 6 RNA scores below the threshold of −3.0 are plotted separately for *n* = 10 LOF1, *n* = 5 LOF2 and *n* = 2 intermediate SNVs. (Boxplots: center line, median; box limits, upper and lower quartiles; whiskers, 1.5× interquartile range; all points shown.) **f,** SpliceAI component scores predict specific splice alterations, including acceptor loss, acceptor gain, donor loss and donor gain. Component SpliceAI scores are plotted against RNA scores for exonic SNVs.

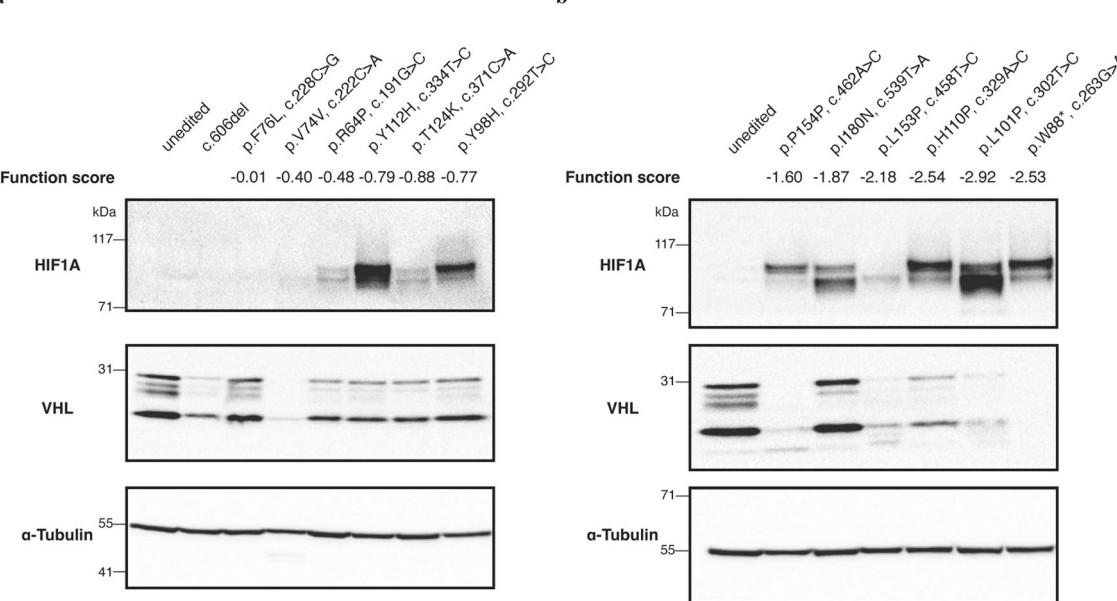

**Extended Data Fig. 6 | Expression of HIF1A and VHL in clonal HAP1 lines with variants assayed in SGE. a,b**, Clonal HAP1 cell lines were isolated containing SNVs introduced independently via prime editing. Western blots were performed to assess VHL and HIF1A protein levels, with α-tubulin stained as a loading control. SNVs scored as significantly depleted in SGE (all except c.228C>G, c.222C>A and c.191G>C) showed increased levels of HIF1A expression compared to unedited HAP1 and cells expressing p.F76L, a variant scored neutrally by SGE. Of note, c.222C>A and c.462A>C had RNA scores of −4.45 and −5.82, respectively. Clonal variability may account for subtle differences between results from the SGE assay and the degree of HIF1A upregulation observed by western blot. (Results are representative of 2 independent blots per line.)

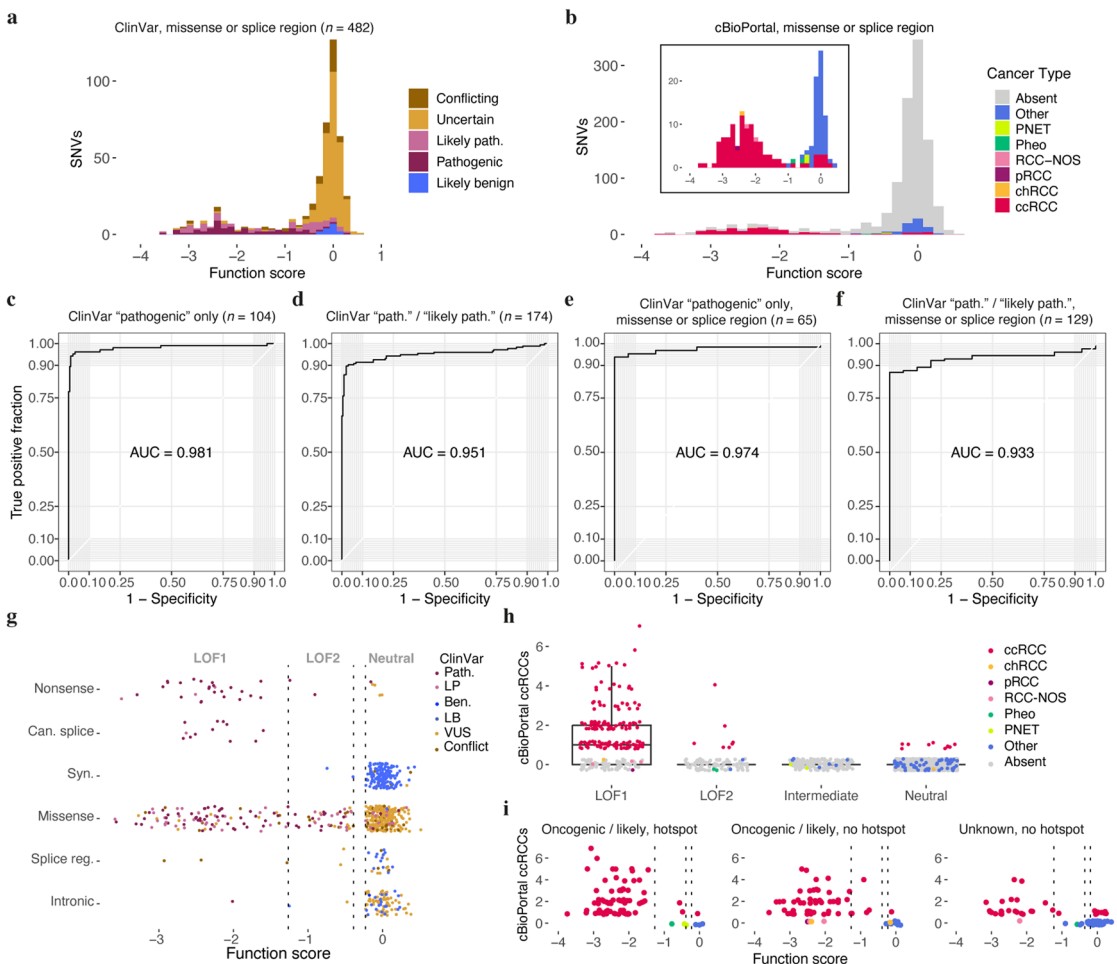

**Extended Data Fig. 7 | Function scores accurately predict pathogenicity of germline and somatic variants. a**, The distribution of function scores for missense and splice region SNVs reported in ClinVar is shown (*n* = 482 SNVs, including 129 'pathogenic' and 'likely pathogenic' SNVs and 15 'likely benign' SNVs). **b**, Missense and splice region SNVs observed in cBioPortal are plotted by function score (inset shows variants present in at least one sample). **c**,**d**, Receiver operating characteristic (ROC) curves are shown for the classification of ClinVar variants using SGE function scores. 'Pathogenic' SNVs (**c**) or 'pathogenic' and 'likely pathogenic' SNVs (**d**) were distinguished from *n* = 190 'benign' or 'likely benign' SNVs. **e**,**f**, The same analyses were repeated as in (**c**) and (**d**), restricting to only missense and splice region SNVs. **g**, Function classes, defined from SGE data, are illustrated to show performance at separating ClinVar variants by

mutation consequence. Thresholds for distinguishing LOF1 (less than −1.26), LOF2 (less than −0.39) and neutral (greater than −0.23) classes are indicated. (Intermediately scored variants are not plotted.) **h**, The number of ccRCC entries in cBioPortal is plotted by function class for *n* = 225 LOF1, *n* = 102 LOF2, *n* = 173 intermediate and *n* = 1,700 neutral SNVs. (Boxplot: center line, median; box limits, upper and lower quartiles; whiskers, 1.5× interquartile range; all points shown.) **i**, For each unique SNV in cBioPortal (*n* = 233 SNVs), function score is plotted versus the number of ccRCC samples in which the SNV was observed. Variants are split by OncoKB annotation and mutational hotspot status in cBioPortal, with thresholds for defining LOF1, LOF2 and neutral variants indicated.

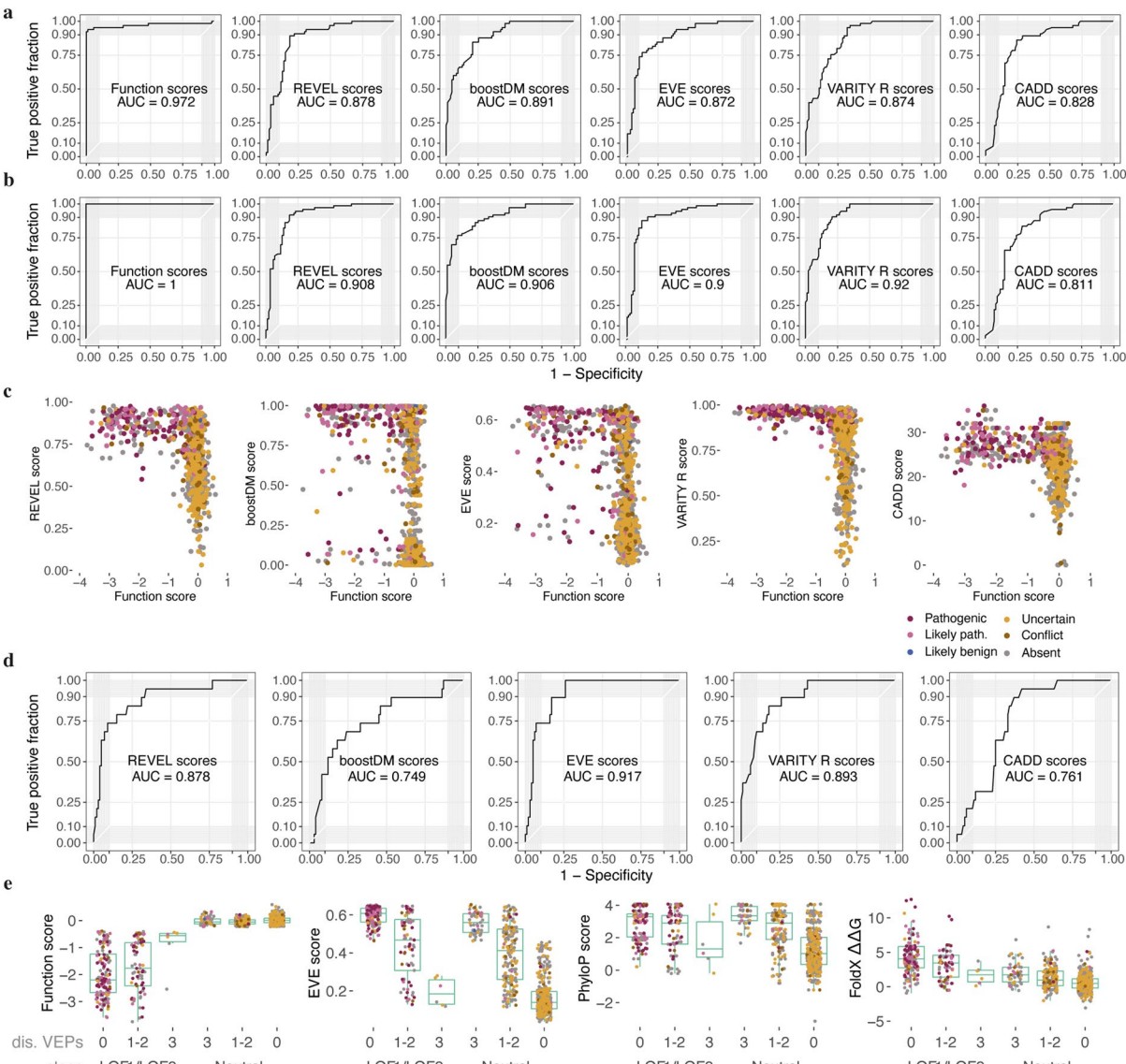

**Extended Data Fig. 8 | Function scores for missense variants outperform predictions from computational models. a,b**, ROC curves indicate the performance of different metrics at distinguishing disease-associated missense variants in *VHL*. The metrics evaluated were SGE function scores, REVEL scores[5], boostDM scores from the VHL-ccRCC model[8], EVE scores[7], VARITY R scores[9] and CADD scores[25]. Missense SNVs were included if scored by all metrics (that is, those present in SGE data from p.M54 to p.A207). In **a**, *n* = 65 missense variants deemed 'pathogenic' in ClinVar were distinguished from *n* = 87 missense SNVs deemed neutral (as in Fig. 4g). In **b**, missense variants present in the gold-standard set of ccRCC-associated SNVs (*n* = 73) were classified against the same neutral set of variants as in (**a**). **c**, Function scores for *n* = 953 missense SNVs are plotted versus scores from each computational predictor, colored by ClinVar

status. **d**, Function scores were used to define two sets of unseen variants (that is, those absent from ClinVar, cBioPortal, population sequencing and VHLdb). Each metric was assessed on its ability to distinguish unseen missense SNVs with function scores below −0.479 (*n* = 19) from the set of missense SNVs with function scores closest to 0 (*n* = 100). **e**, Missense variants classified by SGE as LOF1/LOF2 or neutral were grouped by whether they were discordantly classified by 0, 1 to 2 or all 3 top variant effect predictors (VARITY, EVE and REVEL). Function scores, EVE scores, vertebrate phyloP scores and FoldX predictions are shown across groups (boxplot: center line, median; box limits, upper and lower quartiles; whiskers, 1.5× interquartile range; all points shown except *n* = 22 SNVs with FoldX scores greater than 12.0 in the right panel).

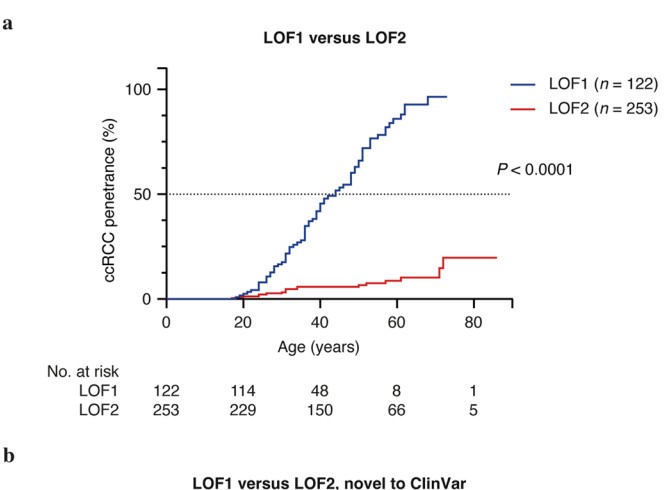

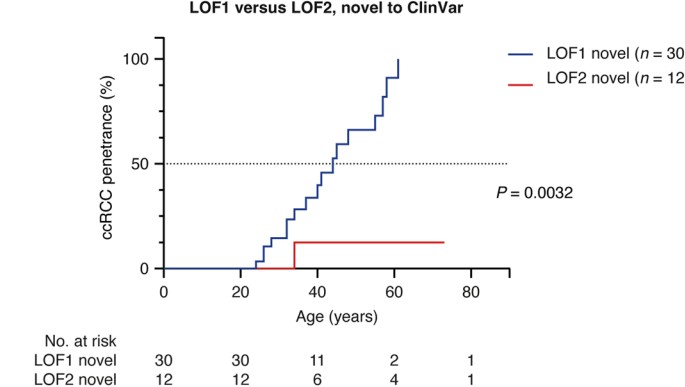

**Extended Data Fig. 9 | Stratification of patients with VHL disease by SGE function class. a**, Patients in the Freiburg VHL Registry were grouped according to whether their germline *VHL* variant was functionally classified as LOF1 or LOF2 by SGE, and a Kaplan–Meier estimator was used to assess differences in age-related ccRCC penetrance with log-rank test for significance (additional details in Methods). **b**, The same analysis and log-rank test were repeated including only patients with variants not reported to be pathogenic in ClinVar at time of analysis (that is, absent, VUS or conflicting interpretations).

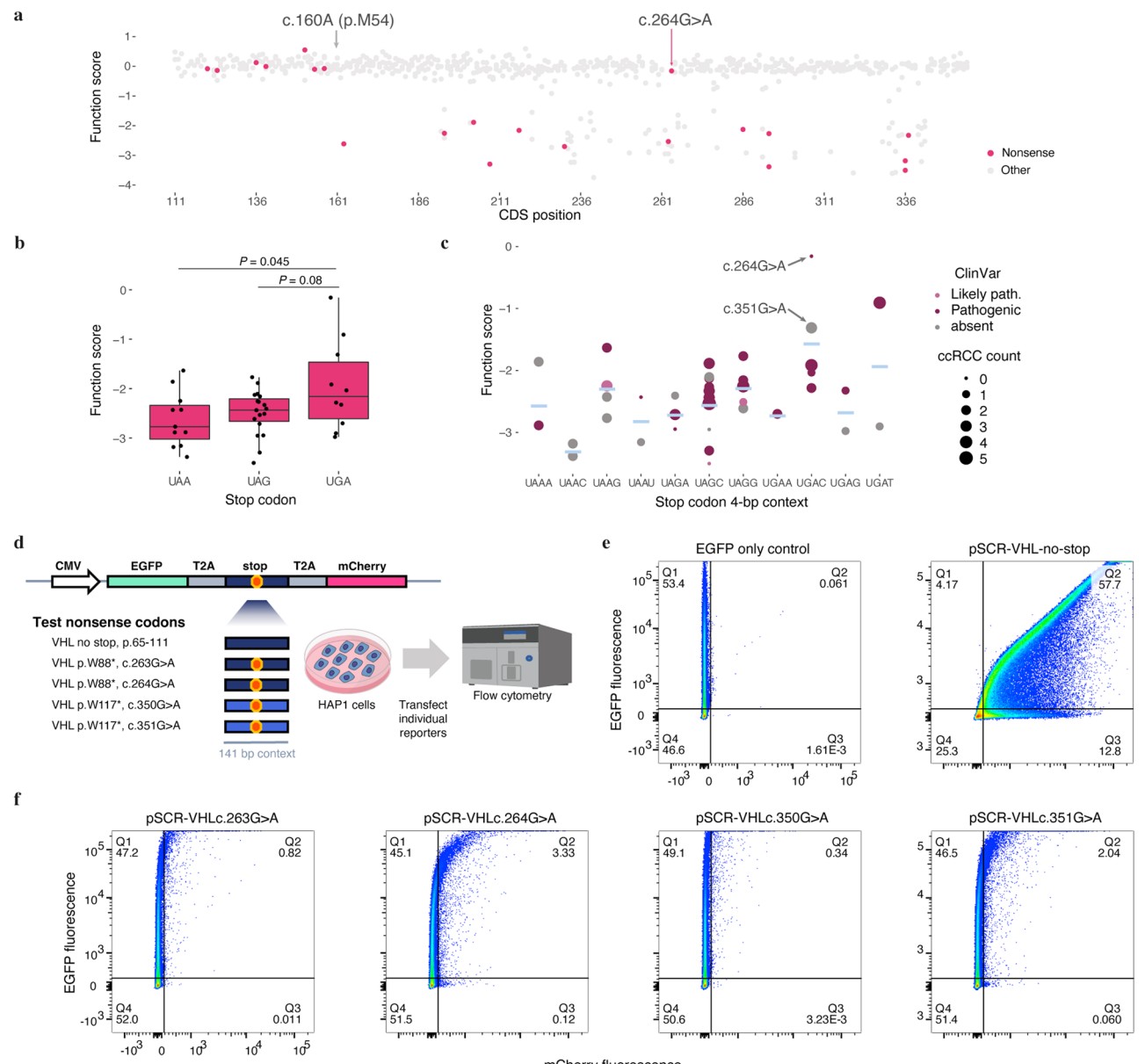

**Extended Data Fig. 10 | Functional effects of nonsense variants in relation to position and stop codon context. a**, Function scores are plotted by position in exon 1 of *VHL* and colored to highlight nonsense variants. All nonsense SNVs tested between c.160A and c.601C scored as depleted, except for c.264G>A, a variant associated with type 2 VHL disease[24]. **b,c**, Function scores for *n* = 40 nonsense variants between c.160 and c.601 are plotted by termination codon (**b**) and 4-bp termination codon context (**c**). Differences between function scores by termination codon were tested using a one-way ANOVA. (Boxplot: center line, median; box limits, upper and lower quartiles; whiskers, 1.5× interquartile range; all points shown.) In **c**, the blue line indicates the mean score for each stop codon context and the size of each dot corresponds to the number of cBioPortal

ccRCC samples in which the SNV has been observed. **d**, A dual-fluorophore stop-codon readthrough (SCR) reporter assay was used to quantify readthrough of nonsense variants assayed by SGE. Nonsense variants with 138 bp of surrounding *VHL* sequence were cloned between EGFP and mCherry, such that mCherry expression only occurs if the nonsense codon fails to terminate translation. **e,f**, Flow cytometry data for live populations of single cells are shown for each plasmid tested, with gating to determine the fraction of transfected cells (EGFP⁺) positive for mCherry expression. Data were normalized to a control vector without a stop codon (pSCR-VHL-no-stop). Control plasmids are in (**e**), and plasmids containing *VHL* nonsense codons are in (**f**).

# Reporting Summary

## Statistics

For all statistical analyses, confirm that the following items are present in the figure legend, table legend, main text, or Methods section.

| n/a | Confirmed | |
|---|---|---|
| ☐ | ☒ | The exact sample size (*n*) for each experimental group/condition, given as a discrete number and unit of measurement |
| ☐ | ☒ | A statement on whether measurements were taken from distinct samples or whether the same sample was measured repeatedly |
| ☐ | ☒ | The statistical test(s) used AND whether they are one- or two-sided<br>*Only common tests should be described solely by name; describe more complex techniques in the Methods section.* |
| ☐ | ☒ | A description of all covariates tested |
| ☐ | ☒ | A description of any assumptions or corrections, such as tests of normality and adjustment for multiple comparisons |
| ☐ | ☒ | A full description of the statistical parameters including central tendency (e.g. means) or other basic estimates (e.g. regression coefficient) AND variation (e.g. standard deviation) or associated estimates of uncertainty (e.g. confidence intervals) |
| ☐ | ☒ | For null hypothesis testing, the test statistic (e.g. *F*, *t*, *r*) with confidence intervals, effect sizes, degrees of freedom and *P* value noted<br>*Give P values as exact values whenever suitable.* |
| ☒ | ☐ | For Bayesian analysis, information on the choice of priors and Markov chain Monte Carlo settings |
| ☒ | ☐ | For hierarchical and complex designs, identification of the appropriate level for tests and full reporting of outcomes |
| ☐ | ☒ | Estimates of effect sizes (e.g. Cohen's *d*, Pearson's *r*), indicating how they were calculated |

*Our web collection on statistics for biologists contains articles on many of the points above.*

## Software and code

Policy information about availability of computer code

| | |
|---|---|
| Data collection | Illumina sequencing .bcl files were processed to .fastq files using bcl2fastq2 (v2.17.1.14). |
| Data analysis | Analysis of sequencing data was performed as fully described in Methods, using a custom pipeline to process .fastq files to variant-level function scores and RNA scores. This pipeline uses SeqPrep v1.3.2 for read merging, needleall (EMBOSS v6.6.0.0) for sequence alignment, and custom scripts written in Python v2.7.5. Subsequent statistical analysis was performed in R v3.6.3 using RStudio v1.4.1106. Micrographs were analyzed using Fiji ImageJ2 v2.14.0 and flow cytometry data were analyzed with FlowJo v10.10. FoldX v5.0 and  PyMol v.2.5.4 were used for structural analysis. GraphPad Prism v10.1.2  was used to analyze age-related ccRCC penetrance.<br><br>Code used in this study is available on GitHub (https://github.com/TheGenomeLab/VHL-SGE) and has been archived to Zenodo (https://zenodo.org/records/11065771) |

For manuscripts utilizing custom algorithms or software that are central to the research but not yet described in published literature, software must be made available to editors and reviewers. We strongly encourage code deposition in a community repository (e.g. GitHub). See the Nature Portfolio guidelines for submitting code & software for further information.

## Data

Policy information about availability of data

All manuscripts must include a data availability statement. This statement should provide the following information, where applicable:

- Accession codes, unique identifiers, or web links for publicly available datasets
- A description of any restrictions on data availability
- For clinical datasets or third party data, please ensure that the statement adheres to our policy

All function scores and RNA scores are included in Supplementary Table 1, as well as NGS read counts. Function scores are also available for visualization at https://vhl-board.onrender.com/ and have been deposited to MAVE-DB (urn:mavedb:00000675-a). Fastq files are publicly available (European Nucleotide Archive accession: PRJEB75229). Unprocessed western blots are included as Source Data.

Structural data (PDB: 1LM8) was accessed from the Protein Data Bank (https://www.rcsb.org/structure/1lm8). ClinVar, cBioPortal, and VHLdb data are available via https://www.ncbi.nlm.nih.gov/clinvar/, https://www.cbioportal.org/, and http://vhldb.bio.unipd.it/, respectively. UK Biobank, TOPMed, and gnomAD data are accessible via https://app.genebass.org/, https://bravo.sph.umich.edu/freeze8/hg38/, and https://gnomad.broadinstitute.org/, respectively. CADD scores can be found at https://cadd.gs.washington.edu/download, and missense variant scores from REVEL, boostDM, EVE, and VARITY are available at   https://sites.google.com/site/revelgenomics/downloads, https://www.intogen.org/boostdm/search?ttype=RCCC&gene=VHL, https://evemodel.org/, and http://varity.varianteffect.org/, respectively.

## Research involving human participants, their data, or biological material

Policy information about studies with human participants or human data. See also policy information about sex, gender (identity/presentation), and sexual orientation and race, ethnicity and racism.

| Reporting on sex and gender | The Freiburg VHL Registry includes patients screened at least once until 2023 at the von Hippel-Lindau Outpatient Clinic of the University Medical Center Freiburg.  As of January 1, 2024, the Freiburg VHL Registry included 552 participants with data on ccRCC status. Patients lacking clinical data were excluded. In total, 375 (67.9%) patients had a VHL mutation classified as LOF1 or LOF2 by SGE (mean age ± SD = 45.5 years ± 17.6, 52% female). 122 participants had LOF1 mutations (age: 41.4 ± 14.3 years, 47.5% female). 253 participants had a LOF2 mutation (age: 47.5 ± 18.7 years, 54.2% female). 46 different LOF1 mutations and 11 different LOF2 mutations were present.<br><br>Other human-derived data used in analysis are publicly available, having been previously published and/or released by others, meaning no human participants were specifically recruited for this study. Sources of human genetic data analyzed in this study include ClinVar (https://www.ncbi.nlm.nih.gov/clinvar/), cBioPortal (https://www.cbioportal.org/), GeneBass (https://app.genebass.org/), TOPMed (https://bravo.sph.umich.edu/freeze8/hg38/) and VHLdb (http://vhldb.bio.unipd.it/). Sex and gender are not consistently reported in these database, precluding further analysis. |
| --- | --- |
| Reporting on race, ethnicity, or other socially relevant groupings | The Freiburg VHL Registry includes patients screened at least once until 2023 at the von Hippel-Lindau Outpatient Clinic of the University Medical Center Freiburg.<br><br>Other human-derived data used in analysis are publicly available, having been previously published and/or released by others, meaning no human participants were specifically recruited for this study. Sources of human genetic data analyzed in this study include ClinVar (https://www.ncbi.nlm.nih.gov/clinvar/), cBioPortal (https://www.cbioportal.org/), GeneBass (https://app.genebass.org/), TOPMed (https://bravo.sph.umich.edu/freeze8/hg38/) and VHLdb (http://vhldb.bio.unipd.it/). Race, ethnicity, and other socially relevant groupings are not consistently reported in these database, precluding further analysis. |
| Population characteristics | The Freiburg VHL Registry includes patients screened at least once until 2023 at the von Hippel-Lindau Outpatient Clinic of the University Medical Center Freiburg.  As of January 1, 2024, the Freiburg VHL Registry included 552 participants with data on ccRCC status. Patients lacking clinical data were excluded. In total, 375 (67.9%) patients had a VHL mutation classified as LOF1 or LOF2 by SGE (mean age ± SD = 45.5 years ± 17.6, 52% female). 122 participants had LOF1 mutations (age: 41.4 ± 14.3 years, 47.5% female). 253 participants had a LOF2 mutation (age: 47.5 ± 18.7 years, 54.2% female). 46 different LOF1 mutations and 11 different LOF2 mutations were present.<br><br>Clinical data including age, gender and diagnostic results were recorded in a predefined database. |
| Recruitment | The Freiburg VHL Registry includes patients screened at least once until 2023 at the von Hippel-Lindau Outpatient Clinic of the University Medical Center Freiburg. All included patients have provided written informed consent. Inclusion criterion for this retrospective analysis was the detection of a VHL germline mutation assayed by SGE. Patients for whom no clinical data were available were excluded. |
| Ethics oversight | Use of the anonymised data for further analysis was approved by the ethics committee of Freiburg University (EK-FR 79/20). |

Note that full information on the approval of the study protocol must also be provided in the manuscript.

# Field-specific reporting

Please select the one below that is the best fit for your research. If you are not sure, read the appropriate sections before making your selection.

☒ Life sciences ☐ Behavioural & social sciences ☐ Ecological, evolutionary & environmental sciences

For a reference copy of the document with all sections, see nature.com/documents/nr-reporting-summary-flat.pdf

# Life sciences study design

All studies must disclose on these points even when the disclosure is negative.

| | |
|---|---|
| Sample size | We set out to study all possible single nucleotide variants across the coding sequence of VHL. This number is determined by multiplying the length of the DNA sequence being studied (in base pairs) by 3 (the number of possible single nucleotide variants at each position). In each analysis, all variants assayed falling into a specific category were included (e.g. all pathogenic or benign variants in ClinVar, all variants seen in a particular type of cancer). No specific sample sizes were chosen, but rather all qualifying variants successfully assayed in each category were included. The number of genetic variants present in each analysis are thus bounded by existent variation in the human population. |
| Data exclusions | A small fraction of experimental data were excluded on the basis of high experimental noise owing to low rates of CRISPR-mediated gene editing at certain genomic positions. These exclusions for technical reasons are discussed in the manuscript (see Results, Methods, and Supplementary Fig. 3). Such exclusions were not predetermined, but instead made using consistently applied thresholds designed to ensure data quality. ClinVar entries not meeting ClinVar's predetermined assertion criteria were excluded from analysis. |
| Replication | Two biological replicates (i.e. separate experiments from transfection forward) were performed and used to derive independent scores for each variant. Scores across replicates were well correlated (Fig. 2a). Where indicated, the average score for each variant across replicate experiments was used for analysis. |
| Randomization | One of many variants being assayed per experiment was introduced by chance to each cell via homologous recombination. Cells with different genetic variants were then treated the same, as part of a single pool of cells. For analysis, groups of variants were determined by patterns of genetic variation reported elsewhere (e.g. variants seen in cancer, variants seen in human germline testing, variants deemed pathogenic by clinicians). Therefore, randomization was not performed in this study. Covariates were not consistently reported across external human genetics databases analyzed, precluding their analysis. |
| Blinding | Experiments were internally controlled through the use of multiplexed assays, meaning variants being compared were all treated the same as part of a large pool of cells. In such instances, experimenters are inherently blind to which variants are present while carrying out assays, eliminating the need for blinding of individual samples. Blinding during analysis of SGE data was not performed, as all variants passing technical filters for data quality were treated identically without exclusions.<br><br>A final set of SGE data was used to retrospectively analyze clinical data, including pathogenicity assertions in ClinVar and cBioPortal. All analyses included all variants for which clinical data was available without exclusions. Clinical data was likewise analyzed in that all patients in a pre-defined cohort, the Freiburg VHL registry, were included for analysis without exclusion and analyzed retrospectively. |

# Reporting for specific materials, systems and methods

We require information from authors about some types of materials, experimental systems and methods used in many studies. Here, indicate whether each material, system or method listed is relevant to your study. If you are not sure if a list item applies to your research, read the appropriate section before selecting a response.

## Materials & experimental systems

| n/a | Involved in the study |
|---|---|
| ☐ | ☒ Antibodies |
| ☐ | ☒ Eukaryotic cell lines |
| ☒ | ☐ Palaeontology and archaeology |
| ☒ | ☐ Animals and other organisms |
| ☐ | ☒ Clinical data |
| ☒ | ☐ Dual use research of concern |
| ☒ | ☐ Plants |

## Methods

| n/a | Involved in the study |
|---|---|
| ☒ | ☐ ChIP-seq |
| ☐ | ☒ Flow cytometry |
| ☒ | ☐ MRI-based neuroimaging |

## Antibodies

| | |
|---|---|
| Antibodies used | Western blots: Mouse tubulin antibody (Sigma-Aldrich, T6199, 1:3,000), rabbit VHL antibody (Cell Signaling Technology, 68547, 1:1,1000), mouse HIF1A antibody (BD Transduction Laboratories, 610959, 1:1,000). Secondary antibodies: goat anti-mouse IgG-HRP (Abcam, ab205719, 1:10,000), goat anti-rabbit IgG-HRP (Sigma, AP307P, 1:10,000).<br><br>Immunofluorescence microscopy primary: rabbit anti-VHL (Cell Signaling Technologies, 68547, 1:200), mouse anti-HIF1α (Novus Biologicals, NB100-105, 1:50), |

Immunofluorescence secondary:
Donkey anti-Rabbit IgG Alexa Fluor 555 (Thermo Fisher, A-31572, 1:500), goat anti-Mouse IgG Alexa Fluor 647 (Thermo Fisher, A-21235, 1:500)

| | |
|---|---|
| Validation | All antibodies used are well-validated for use in the indicated applications in human cells, as evidenced by a wealth of citations and validation statements and data described on the manufacturer websites:<br><br>Mouse tubulin antibody (Sigma-Aldrich, T6199) is a Sigma "enhanced validation" antibody with at least 10 references (https://www.sigmaaldrich.com/GB/en/product/sigma/t6199?icid=sharepdp-clipboard-copy-productdetailpage).<br><br>Rabbit VHL antibody (Cell Signaling Technology, 68547) has 95 citations (https://www.cellsignal.com/products/primary-antibodies/vhl-antibody/68547).<br><br>Mouse HIF1A antibody (BD Transduction Laboratories, 610959) has 7 citations. (https://www.bdbiosciences.com/en-gb/products/reagents/microscopy-imaging-reagents/immunofluorescence-reagents/purified-mouse-anti-human-hif-1.610959)<br><br>Mouse anti-HIF1α (Novus Biologicals, NB100-105) has been cited over 1,000 times: https://www.novusbio.com/products/hif-1-alpha-antibody-h1alpha67_nb100-105#datasheet.<br><br>In data provided in the manuscript, specificity of all primary antibodies for HIF1A and VHL was confirmed via genetic manipulation of HAP1 cells (i.e. VHL knock-out), leading to the expected change in staining by both western blot and immunofluorescence (EDF6).<br><br>Secondary antibodies used for immunofluorescence microscopy were assessed for background staining using both knockout lines lacking VHL and staining controls lacking primary antibody. |

# Eukaryotic cell lines

Policy information about cell lines and Sex and Gender in Research

| | |
|---|---|
| Cell line source(s) | The parental HAP1 cell line was originally obtained from the commercial supplier formerly known as Haplogen (now Horizon Discovery). HAP1 cells were derived from a male human but are haploid and lack a Y chromosome. |
| Authentication | The HAP1 cells used in this study were commercially sourced and were not independently authenticated. Sanger sequencing was performed on clonal populations where indicated to confirm genetic manipulations (i.e. knock-out generation). |
| Mycoplasma contamination | All cell lines used in this study tested negative for mycoplasma contamination. |
| Commonly misidentified lines (See ICLAC register) | None used. |

# Clinical data

Policy information about clinical studies

All manuscripts should comply with the ICMJE guidelines for publication of clinical research and a completed CONSORT checklist must be included with all submissions.

| | |
|---|---|
| Clinical trial registration | n/a |
| Study protocol | This was a retrospective cohort analysis without a study protocol |
| Data collection | The Freiburg VHL Registry includes patients screened at least once through 2023 at the von Hippel-Lindau Outpatient Clinic of the University Medical Center Freiburg. Clinical data such as age, gender and diagnostic results were recorded in a predefined database. Clinical surveillance was performed according to international guidelines for VHL disease (VHL Active Surveillance Guidelines) and included an MRI and/or CT scan of the abdomen for the diagnosis of ccRCC. |
| Outcomes | For ccRCC, the first radiologic description was considered the initial diagnosis of ccRCC. Registrants without ccRCC were censored at the age of their last visit. |

# Flow Cytometry

## Plots

Confirm that:

☒ The axis labels state the marker and fluorochrome used (e.g. CD4-FITC).

☒ The axis scales are clearly visible. Include numbers along axes only for bottom left plot of group (a 'group' is an analysis of identical markers).

☒ All plots are contour plots with outliers or pseudocolor plots.

☒ A numerical value for number of cells or percentage (with statistics) is provided.

## Methodology

| | |
|---|---|
| Sample preparation | pSCR plasmids containing sequences of interest between EGFP and mCherry genes were cloned (see Methods). Each vector was transfected into HAP1-LIG4KO cells as described. On day 5 post-transfection, cells were trypsinized, washed, and resuspended in FACS buffer. |
| Instrument | BD Fortessa X20 flow cytometer |
| Software | FlowJo v10.10 |
| Cell population abundance | Data was recorded for at least 150,000 single cells per sample. |
| Gating strategy | Quadrant gating was performed (FlowJo) to determine the fraction of transfected cells (EGFP+) that were mCherry+. Prior gating on living, single cells was performed using FSC/SSC. |

☒ Tick this box to confirm that a figure exemplifying the gating strategy is provided in the Supplementary Information.

