## [Peer Review File · Nature Genetics]

Peer Review Information

Manuscript Title: Saturation Genome Editing Maps the Functional Spectrum of Pathogenic *VHL* Alleles

Corresponding author name(s): Dr Gregory (M) Findlay

Reviewer Comments & Decisions:

Decision Letter, initial version:

21st Jun 2023

Dear Gregory,

hope this email finds you well.

Your Article, "Saturation Genome Editing Resolves the Functional Spectrum of Pathogenic *VHL* Alleles" has now been seen by 2 referees. You will see from their comments copied below that while they find your work of considerable potential interest, they have raised substantial concerns that must be addressed. In light of these comments, we cannot accept the manuscript for publication, but would be very interested in considering a revised version that addresses these concerns.

In particular, Reviewer #1 thinks the paper is too focused on the methodology from your previous work and should develop more in the direction of explaining the novel *VHL* variants functionally. Reviewer #2 is quite positive, saying that the paper is of broad interest and has mainly technical comments about the splicing analysis and the interpretation of the variants.

We agree with Reviewer #1 that a more mechanistic explanation of your findings would strengthen your study.

However, we are committed to providing a fair and constructive peer-review process. Do not hesitate to contact us if there are specific requests from the reviewers that you believe are technically impossible or unlikely to yield a meaningful outcome.

We hope you will find the referees' comments useful as you decide how to proceed. If you wish to submit a substantially revised manuscript, please bear in mind that we will be reluctant to approach the referees again in the absence of major revisions.

If you choose to revise your manuscript taking into account all reviewer and editor comments, please highlight all changes in the manuscript text file. At this stage we will need you to upload a copy of the

manuscript in MS Word .docx or similar editable format.

*2) If you have not done so already please begin to revise your manuscript so that it conforms to our Article format instructions, available here. Refer also to any guidelines provided in this letter.

Please be aware of our guidelines on digital image standards.

[redacted]

If you wish to submit a suitably revised manuscript we would hope to receive it within 6 months. If you cannot send it within this time, please let us know. We will be happy to consider your revision so long as nothing similar has been accepted for publication at Nature Genetics or published elsewhere. Should your manuscript be substantially delayed without notifying us in advance and your article is eventually published, the received date would be that of the revised, not the original, version.

Nature Genetics is committed to improving transparency in authorship. As part of our efforts in this direction, we are now requesting that all authors identified as 'corresponding author' on published papers create and link their Open Researcher and Contributor Identifier (ORCID) with their account on the Manuscript Tracking System (MTS), prior to acceptance. ORCID helps the scientific community achieve unambiguous attribution of all scholarly contributions. You can create and link your ORCID from the home page of the MTS by clicking on 'Modify my Springer Nature account'. For more information please visit please visit www.springernature.com/orcid.

Thank you for the opportunity to review your work.

My best wishes,
Chiara

Chiara Anania, PhD
Associate Editor
Nature Genetics
<https://orcid.org/0000-0003-1549-4157>

Referee expertise:

Referee #1: cancer biology/functional genomics

Referee #2: human genetics, saturation mutagenesis

Reviewers' Comments:

Reviewer #1:

Remarks to the Author:

Buckley et al applied Saturation Genome Editing (SGE) to VHL to understand the consequences of mutations on VHL function. They argue that their method outperforms current standard computational models to predict VHL pathogenic alleles. They identified differential risk across tumor types and revealed how new variants. The most interesting findings relate to the novel VHL variant effects identified in their screen (e.g. C-term extension and W88*). This information will be helpful for those analyzing VHL variants in clinical samples.

Although the screens were well executed, the overall impact is modest. The analysis of the method provides little insight over their prior work. Although they identified some potentially interesting alleles, further mechanistic studies are necessary to credential these variants.

Major Comments:

1. The SGE technology based on homology-directed repair has already been vetted (Findlay, GM Nature 2018). The current article provides too much emphasis on re-qualifying the same method, with minimal characterization and validation of novel biological findings pertaining to VHL.
2. Authors claim "Function scores capture fitness effects mediated at both the transcript and protein level" but provide no evidence at the protein expression level, nor any biophysical/biochemical evidence of impact on HIF1a interaction and HIF1a activity. These data should be provided to substantiate their claims.
3. The method for deriving RNA scores is clever but may result in false positives/negatives in the case that low abundance gDNA can artificially conflate a high RNA score (RNA/gDNA ratio) and vice versa. The authors should provide further details in methods if they utilize an abundance cutoff to account for this, as well as orthogonally validate their top variants with differential expression by qPCR. Authors should also provide mechanistic rationale for why some missense variants should result in destabilized RNA.
4. The most impactful findings of these studies are the novel variants identified, yet this portion of the study is inadequately addressed. Authors need to validate both the impact of indels resulting in 42-AA extension of VHL as well as functionally evaluate its effect on HIF1a. A clear mechanism of how this impairs VHL's ability to regulate HIF should be provided. Furthermore, the W88* opal codon hypothesis should be tested.

Minor Comments:

1. Please provide the % of all possible VHL variants observed in initial timepoint, post-transfection. It is unclear how truly comprehensive their representation is.
2. Authors claim, "Frameshifting indels are strongly selected against in parental HAP1." However, the data seem to indicate in-frame editing events also are strongly selected against. The authors should separate out mutation type and show the distributions more clearly if they want to make this claim.
3. For better visualization and understanding of relevant "hits," provide labels for some of the top scoring variants in Fig2.
4. Several VHL variants have been well characterized/studied (e.g. R167Q, Y112*, etc). These should be benchmarked/labeled throughout the figures within the manuscript wherever relevant (both main figure and supplement).
5. The claim that only low RNA scores reliably predict LoF at the cellular level is unconvincing based on the data provided. Please provide a quantitative/statistical justification for this claim.
6. In Figure 3, authors should separate out average score shown on structure based on the average of

amino acids that share biophysical properties.

7. Relevant contacting amino acid positions should be labeled in all structural data shown in Fig 3.
8. Fig 3d should show ELOC interaction, much like HIF1a.
9. In Fig3, authors should provide a figure showing the distribution of variant scores that reside at important protein interaction interfaces vs all others.
10. The correlations observed in Fig4c&d are unconvincing (e.g. $\rho = \sim 0.2$). Authors should provide rationale for why certain profoundly LoF variants are so infrequently observed while others are highly frequent. Perhaps, incorporating mutational signatures will help provide clarity (similar to Giacomelli AO, et al Nature Genetics 2018).
11. Authors should provide more concrete numbers (e.g. % variants) for the claim "Variants observed repeatedly in population sequencing nearly always scored neutrally." It is hard to gauge this from the way the data is currently visualized.
12. Authors should validate VHL variants of the pheochromocytoma phenotype by determining HIF1a activity biochemically and looking at downstream markers of HIF1a activity (VEGFA, CHGA, etc.) or reporter assay, comparing this with type 1 disease variants.
13. Please provide statistics for Fig 5c, similar to Fig 5f.
14. It is hard to visually tell the difference between conditions in Supp Fig 1. Authors should provide image analysis quantification of confluence or viability readout normalized to VHL control.
15. Given the authors claim that their SGE is comprehensive, they should provide a full heatmap of functional scores for all missense substitutions at all positions represented in the screen to better visualize the impact of each variant per position.
16. Are there any primary patient samples the authors could validate some of their novel variants in?
17. A legend for Supplementary Table headers should be provided as it is hard to interpret some of the columns.

Reviewer #2:

Remarks to the Author:

Findlay and colleagues present a MAVe assay covering most of the possible SNVs within the tumor suppressor gene VHL, using some refinements to their 'saturation genome editing' (SGE) approach. The resulting measurements strongly correlate with standing clinical variant interpretations for type 1 VHL disease, and they highlight some interesting exceptions with supporting phenotypic correlations from the literature. Their results are of technically very high quality, and are clearly presented. This study is another demonstration of MAVEs' promise for interpreting germline variants and will no doubt be of broad interest.

As I describe below, I did have a few concerns related the calibration of the data (i.e., threshold-setting) for clinical variant interpretation, and about the analysis of splicing impacts.

1. RNA scores. The authors sequentially measure variants' depletion from the pool of spliced RNA at two timepoints. In the case of VHL, it appears that there was not much constraint at the level of splicing or RNA stability among exonic variants.
 - a. A key question which arises in the context of splice disruption is to determine a threshold of expression defect which is pathogenic. I am not sure whether that has been done here, but hope that such an analysis could be attempted given this rich dataset. Along those lines, how concordant are the scores (splicing and function) with those of splicing effect predictors like MaxEntScan and SpliceAI? (A related limitation which could be more prominently mentioned is the lack of RNA measurement for intronic variants, where there might be the broadest range of effects)
 - b. Lines 190-191 (+338-339) are unclear. Are the authors suggesting that some RNA-depleting variants can be rescued by feedback/upregulation in order to survive to day 20? Why would a similar effect not influence the results of missense variants with stability/abundance (but not RNA) defects?
 - c. What was the basis for including the intron 1 region (given the apparent lack of effects?) I notice there is a cryptic exon in this region (eg PMID 31996412) . Is this system informative as to the functional effect of that exon's inclusion?

2. Structure-function analysis. The authors noted that LOF-scoring missense variants clustered in regions of secondary structure, which is unsurprising. I wonder if this part of the analysis could be developed further. Did these tend to be destabilizing (by Western or by FoldX/rosetta/etc prediction?) What fraction of the observed LOF effects are due to stability defects vs interaction defects?

3. The poor specificity exhibited by the bioinformatic callers (Fig S8c) aligns well with others' observations from MAVE screens. Are the discordant variants (predicted by algorithm to be deleterious but neutral by MAVE) concentrated at any particular residues or domains? Is it correlated with conservation?

4. Calibration/clinical variant interpretation

a. Fig 4 and Supp Fig 7 should be limited to missense/splice variants; otherwise the plots are somewhat dominated by syn and non/fs/indel which are peripheral to the interpretation challenge this paper addresses.

b. I wonder about the robustness of the thresholds in Figure 4f-h. It is great that 100% separation can be obtained between ccRCC and neutral variants after some curation, but I wonder how sensitive this may be to the curation criteria. Also, in order to achieve this separation, the pathogenic/benign thresholds are very closely spaced, with a score difference of only 0.091 (-0.479 - (-0.388)), which appears (Fig 2A) to be considerably less than the average inter-replicate variation. I would encourage the authors to consider a less aggressive threshold selection (i.e., broadening the intermediate range or accepting some FP/FNs), and/or tempering the language about classification performance, lest marginal scores be given too much weight for novel variants.

c. Fig 4h / lines 249-253. The suggestion here is that the function score may quantitatively measure the (i) degree of LOF, and (ii) risk conferred, within the low-scoring range. This is an interesting point and gets to a broader question of MAVEs' calibration. The comparison made here is specifically among SNVs (a) absent from ClinVar and (b) with function score < -0.47, between those seen in ccRCC vs not seen in ccRCC. There does seem to be a convincing difference (though it seems weaker when condition (a) is dropped; not obvious why this would be?) What is notable then that very few of the seen-in-ccRCC variants fall between -0.479 and -2. This would seem to argue against the pathogenicity of the novel variants falling in to this range, but it also seems at odds with these variants' apparent depletion in population datasets (4e).

5. Discussion

- As the authors acknowledge, a key limitation with MAVE assays is understanding how the cellular phenotypes map to the effects in vivo. In this case, the direction of effect appears to be opposite: VHL loss results in loss of fitness in culture but is tumorigenic in vivo (though the scores, with the sign flipped, are nevertheless highly predictive). It might be instructive for the authors to speculate (or cite any relevant literature) as to why constitutive HIF activity is toxic in culture.

Minor

- Fig 1d. A scale bar in base pairs would be helpful. Also I presume this is so but the counts should be only variants for which the assay is informative, ie SNVs

- Fig 4c,d – shouldn't correlation stats be negative?

- Fig 4e is overplotted and two of the status (conflicting and pathogenic) are shaded in very similar colors. Maybe separate by status and plot separately in a supplementary figure?

- Fig 7c – is the x label "Not pathogenic" accurate (as some have quite negative function scores)? Or are these merely not seen in VHLdb?

- L290 "SNVs not deemed pathogenic" ◊ if this primarily refers to SNVs deemed absent from ClinVar I would rephrase as such

Author Rebuttal to Initial comments

Point-by-point response to reviewer comments

Reviewer comments are in black.

Responses are in blue.

Sections of new or revised text in the manuscript are reproduced in green with original text for context in black.

Reviewers' Comments:

Reviewer #1:

Remarks to the Author:

Buckley et al applied Saturation Genome Editing (SGE) to VHL to understand the consequences of mutations on VHL function. They argue that their method outperforms current standard computational models to predict VHL pathogenic alleles. They identified differential risk across tumor types and revealed how new variants. The most interesting finding relate to the novel VHL variant effects identified in their screen (e.g. C-term extension and W88*). This information will be helpful for those analyzing VHL variants in clinical samples.

Although the screens were well executed, the overall impact is modest. The analysis of the method provides little insight over their prior work. Although they identified some potentially interesting alleles, further mechanistic studies are necessary to credential these variants.

We thank the reviewer for their positive comments on the high level of technical execution and the mechanistic interest of the newly studied variants and we fully agree that our findings would benefit from further mechanistic investigation. In our revised manuscript, we detail several additional experiments and analysis to further delineate the mechanisms by which variants act. We describe each of these results in detail below.

We think the study's impact will be significant both because we demonstrate these novel mechanisms and because the data will be used directly to improve variant classification once published. As we report, hundreds of VUS in *VHL* currently exist. This is the first experimental evidence to support classification for the vast majority of such variants. We show SGE data reliably predict disease risk, such that in accordance with ACMG guidelines¹ the data can be used to support reclassifying VUS to either pathogenic or benign. Therefore, once our data are released

they will improve the diagnosis of germline predisposition (VHL disease) and enable stratification of patients by cancer risk (as we now show in revision). Identifying causal variants can also improve selection of precision therapies in the context of sporadic renal cancer. This is particularly timely with the recent FDA approval and growing use of belzutifan for the treatment of ccRCC.

Major Comments:

1. The SGE technology based on homology-directed repair has already been vetted (Findlay, GM Nature 2018). The current article provides too much emphasis on re-qualifying the same method, with minimal characterization and validation of novel biological findings pertaining to VHL.

We thank the reviewer for the broad feedback on the manuscript's interest. In our revised manuscript, we have much expanded our mechanistic investigations into why specific variants cause effects, focusing on the variants acting via newly discovered mechanisms, specifically c.264G>A (p.W88*) and indels leading to long C-terminal extensions. New experiments detailed in subsequent responses include:

1. Derivation of monoclonal cell lines harboring specific *VHL* variants and analysis of effects on VHL and HIF1A protein expression via western blot and immunofluorescence microscopy.
2. Use of a flow cytometry-based assay to directly quantify stop codon readthrough for nonsense variants.
3. Direct demonstration that long C-terminal extensions lead to loss of protein expression and subsequent HIF1A upregulation in a reading frame-specific manner.

We now also include deeper mechanistic analyses, showing how variants with low RNA scores are predicted to impact splicing, and how missense variants' with low function scores are predicted to impact protein stability.

Following the reviewer's suggestion to minimize characterization of the method, we have revised the manuscript to limit technical details while still highlighting the clinical performance of the assay for identifying causal variants. Clinical geneticists require careful assay validation if functional data is to be used for variant interpretation, making this essential, and the high accuracy we show for categorizing ccRCC-associated variants will be of high interest. As there are a growing number of labs attempting SGE studies, we initially emphasized technical optimizations because the increase

in the dynamic range of the assay is much improved (**Fig. 1f,g, SFig.2**). While we think this will make our optimized protocol more attractive to other labs, we do appreciate the technical benchmarking is less critical to this audience. Accordingly, we have summarized the key changes to the protocol as follows:

With the aim of measuring more subtle effects on growth, SGE experiments were performed using a highly optimized protocol modified from published work² to feature improved transfection efficiency, a longer time course, and addition of 10-deacetyl-baccatin-III (DAB) to maintain haploidy³ (**Fig. 1f-g, Supplementary Figs. 1,2, Supplementary Note 1**).

We have moved the assay optimization section into **Supplementary Note 1**:

Supplementary Note 1. Optimizing the SGE protocol to assay *VHL*.

We substantially optimized SGE to enable accurate measurement of functional effects of *VHL* SNVs. Sequencing of initial libraries revealed two regions of exon 1 to have skewed variant distributions at sites of repetitive, GC-rich sequence (**Supplementary Fig. 2a-e**). Additional synonymous mutations were therefore engineered in these regions, resulting in improved library uniformity (**Supplementary Fig. 2f-l**).

For four exonic regions initially assayed using normal HAP1 culture media, only modest growth defects were observed for expected LoF variants (**Fig. 1f, Supplementary Fig. 3**). HAP1 cells can revert to diploidy with prolonged culture⁴, a phenomenon that could weaken recessive effects measured in multiplex. Recently, 10-deacetyl-baccatin-III (DAB) was identified via small molecule screening to select for haploid cells³. Therefore, we next performed SGE for all *VHL* regions in media containing 2.5 μ M DAB. This led to a substantial improvement in dynamic range (**Fig. 1g, Supplementary Fig. 3**). In exon 2, for example, the median function score of nonsense and canonical splice site SNVs dropped 4-fold, from -0.62 to -2.49. Across all SGE regions assayed with and without DAB, there were 39.3% more depleted SNVs identified in DAB-treated cells. Therefore, we used only data from SGE experiments performed with DAB to calculate final function scores.

2. Authors claim “Function scores capture fitness effects mediated at both the transcript and protein level” but provide no evidence at the protein expression level, nor any biophysical/biochemical evidence of impact on HIF1a interaction and HIF1a activity. These data should be provided to substantiate their claims.

We thank the reviewer for this feedback. Firstly, this figure title was intended to indicate our SGE data reflect variant effects at both the level of RNA (e.g. splicing) and at the level of protein function (e.g. protein destabilization). We did not intend to imply we're *directly* measuring protein levels or interactions, but rather that by editing the genome to install variants, our assay captures functional effects mediated at the level of mRNA expression and at the level of protein function. We have revised this figure title in line with changes to the figure to better reflect our intended meaning:

Fig. 3: Function scores capture fitness effects secondary to splicing alterations and impairment of protein function.

More importantly, to broadly validate our SGE assay and to show specific defects in VHL protein expression and HIF1A regulation, we now include western blot data generated using HAP1 clonal lines harboring individual *VHL* variants. Here, we show broad validation of VHL variants' effects on VHL and HIF1A protein levels in our system. We also include new western blot and immunofluorescence data to test effects of specific variants, as described in response to main comment #4.

We first introduced 12 variants assayed with SGE to HAP1 cells and confirmed via Sanger sequencing the presence of a single VHL allele. Variants spanned a range of function scores from 0.0 to -2.9 and included both well-studied variants for benchmarking as well as newly classified variants. We also included two intermediately classified variants, and two variants with low RNA scores. (Our revised classification system is detailed below in response to reviewer #2; briefly, LOF1 variants score lowest, LOF2 variants are significantly depleted but not as low as ccRCC-associated variants, and intermediate variants score near neutral but are mildly depleted.)

Response Table 1. Variants engineered in clonal cell lines for western blot analysis.

cHGVS	pHGVS	function score	q-value	rna score	SGE class	ClinVar	phenotype
c.228C>G	p.F76L	-0.009	0.82	0.162	Neutral	VUS	Unknown
c.222C>A	p.V74V	-0.405	0.04	-4.450	Intermediate	absent	Rec. VHL def.
c.191G>C	p.R64P	-0.477	0.01	-0.275	Intermediate	Likely path.	Type 2C
c.292T>C	p.Y98H	-0.770	2.19E-09	0.169	LOF2	Pathogenic	Type 2A
c.334T>C	p.Y112H	-0.789	8.34E-10	-0.343	LOF2	Pathogenic	Type 2A
c.371C>A	p.T124K	-0.882	1.29E-10	-0.109	LOF2	absent	Unknown

c.462A>C	p.P154P	-1.605	3.99E-32	-5.819	LOF1	absent	Unknown
c.539T>A	p.I180N	-1.869	6.33E-67	-0.045	LOF1	absent	Unknown
c.329A>C	p.H110P	-2.541	1.11E-91	-0.587	LOF1	absent	Unknown
c.302T>C	p.L101P	-2.918	2.59E-120	-0.250	LOF1	Likely path.	Type 1
c.458T>C	p.L153P	-2.183	3.95E-58	0.083	LOF1	absent	Unknown
c.263G>A	p.W88*	-2.532	4.22E-91	-0.1910589	LOF1	Pathogenic	Type 1

a

b

Supplementary Fig. 6: Expression of HIF1A and VHL in clonal HAP1 lines with variants assayed in SGE. a,b, Clonal HAP1 cell lines were isolated containing SNVs introduced independently via prime editing. Western blots were performed to assess VHL and HIF1A protein levels, with α -Tubulin stained as a loading control. SNVs scored as significantly depleted in SGE (all except c.228C>G, c.222C>A, and c.191G>C) showed increased levels of HIF1A expression compared to unedited HAP1 and cells expressing p.F76L, a variant scored neutrally by SGE. Of note, c.222C>A and c.462A>C had RNA scores of -4.45 and -5.82, respectively.

The western blot results confirm that SGE a.) accurately identifies variants leading to upregulation of HIF1A, b.) function scores generally reflect the degree of HIF1A upregulation seen in clonal lines, c.) variants with low RNA scores lead to decreases in VHL protein levels. Notably, none of the missense variants with modestly low SGE scores (p.R64P, p.Y98H, p.Y112H, and P.T124K)

lead to severe reductions in VHL levels. We introduce this data with a new **Results** paragraph:

SNVs spanning with a wide range of scores were introduced independently to HAP1 cells for validation. Western blots to measure VHL and HIF1A confirmed that 9 of 9 variants with significantly reduced function scores lead to increases in HIF1A expression compared to unedited HAP1 cells and a line harboring a missense VUS scored neutrally by SGE (**Supplementary Fig. 6**). Two synonymous variants with low RNA scores, c.222C>A and c.462A>C, result in reduced VHL protein expression, but only c.462A>C led to clear HIF1A upregulation, consistent with its lower RNA score and lower function score. Overall, these data confirm variants depleted in SGE lead to upregulation of HIF1A in HAP1 cells.

A limitation to the analysis of clonal cell lines is inherent variability between clonal populations. An advantage of SGE is that each variant is introduced hundreds of times independently in the experiment, mitigating stochastic effects. Unfortunately, we lack high-throughput methods for assaying VHL variants' effects on HIF1A protein levels and activity directly, which would be required to study these relationships for more variants. This is what motivated us to repeat two SGE experiments in the HIF1A-KO line to ask if any variants' effects were *not* dependent on HIF1A. As shown in **Fig. 5a**, the effects of all low-scoring variants among $n = 797$ SNVs tested in isogenic lines depend completely on HIF1A. We think this is strong evidence to conclude that increased HIF1A activity secondary to VHL functional impairment causes reduced growth in the SGE assay.

Additionally, the analysis of predicted protein stability we now include in **Fig. 3e** also provides computational insight into missense variants effects on VHL protein levels. A full description of this analysis is included in our response to major comment #2 from Reviewer #2, below.

Figure 3e, FoldX-predicted $\Delta\Delta G$ values were higher for missense SNVs depleted in SGE experiments (median depleted variants = 3.63 compared to 0.70 for other SNVs; boxplot: center line, median; box limits, upper and lower quartiles; whiskers, 1.5x interquartile range).

We describe our main findings in a new **Results** paragraph as follows:

To explore features of missense variants impacting function, we examined where depleted mutations map to the protein's structure. The greatest proportion of depleted variants occur at core residues; 55.8% of mutations in the protein core and 49.4% of mutations in an interface core were depleted by SGE, whereas only 26.7% of peripheral interface mutations and 11.1% of other surface mutations were depleted (**Fig. 3d**). We next used FoldX⁵ to predict stability effects of missense variants. FoldX-predicted $\Delta\Delta G$ values were higher for variants with low function scores (median $\Delta\Delta G$ = 3.63 for depleted SNVs versus median $\Delta\Delta G$ = 0.70 for other missense; **Fig. 3e**). Indeed, 76.9% of depleted missense variants had predicted $\Delta\Delta G$ values greater than 2.0, compared to 20.6% of missense variants not deemed depleted by SGE. Restricting analysis to only depleted missense variants, FoldX $\Delta\Delta G$ values correlate inversely with function scores (Spearman's ρ = -0.41), indicating the degree of destabilization is further predictive of functional impairment.

3. The method for deriving RNA scores is clever but may result in false positives/negatives in the case that low abundance gDNA can artificially conflate a high RNA score (RNA/gDNA ratio) and vice versa. The authors should provide further details in methods if they utilize an abundance cutoff to account for this, as well as orthogonally validate their top variants

with differential expression by qPCR. Authors should also provide mechanistic rationale for why some missense variants should result in destabilized RNA.

Thanks for the opportunity to clarify. We only assigned RNA scores to variants for which we assigned function scores, meaning the same cut-off for gDNA reads on day 6 is used, without any additional threshold imposed for RNA-derived reads. We have added a sentence to the Methods section on “Calculating RNA scores” to make this explicit:

RNA scores were only assigned to SNVs that passed quality filtering for function scores, meaning the variant’s day 6 gDNA frequency in both replicates exceeded 1×10^{-5} .

The risk of low variant counts in gDNA conflating RNA scores is a good concern. Yet, we do not see this effect in our data. Here, we plot RNA scores as a function of average gDNA variant frequency across replicates to specifically show this relationship (dotted line at RNA score = 0).

Response Fig. 1, RNA scores as a function of genomic DNA coverage.

Variants with the lowest RNA scores (below the dashed line at -3.0) are seen across a range of gDNA frequencies. Further, the median RNA score for variants in the lowest decile of gDNA frequency was very near zero (0.063), as was the median RNA score for variants in the highest decile of gDNA frequency (-0.057). Additionally, we observe a strong correlation of RNA scores across independent replicates, which we now include in our revised manuscript:

Supplementary Fig. 5b, Day 6 RNA scores from individual replicates are highly correlated (Pearson's $R = 0.87$).

For orthogonal validation, we have used a computational predictor informative of splicing mechanisms (as suggested by Reviewer #2). A key advantage of this analysis is that it is applicable to the complete data set, informing predicted mechanisms of splice disruption and any systematic differences between computational and experimental approaches. We generated the following figure panels comparing SpliceAI scores to RNA scores of exonic variants and function scores of intronic variants.

Figure 3b,c, The maximum SpliceAI score for each SNV is plotted against RNA scores for exonic SNVs (Pearson's $R = -0.70$) and function scores for intronic SNVs ($R = -0.90$).

As SpliceAI is a model trained on sequence features of spliced transcripts, these strong correlations both validate our results and implicate aberrant splicing as the causal mechanism for

exonic variants with low RNA scores (defined as < -3.0 ; $n = 17$) and for intronic variants depleted in SGE ($n = 47$). Regarding more specific mechanistic rationale, SpliceAI offers predictions specifically for acceptor gain, acceptor loss, donor gain, and donor loss:

Supplementary Fig. 6f, SpliceAI component scores predict specific splice alterations, including acceptor loss, acceptor gain, donor loss, and donor gain. Component SpliceAI scores are plotted against RNA scores for exonic SNVs.

As to be expected for exonic variants, SNVs with low RNA scores are often predicted to act via acceptor loss or donor loss.

We describe these new findings in the manuscript in **Results**:

Predictions from SpliceAI⁶, a computational predictor of splicing outcomes, were strongly correlated with RNA scores (**Fig. 3b, Supplementary Fig. 5f**). 16 of 17 SNVs with RNA scores less than -3.0 had SpliceAI scores greater than 0.08 (median of 0.61), whereas 83% of variants with RNA scores greater than -3.0 had SpliceAI scores of 0.00. The only variant with a low RNA score and a SpliceAI score of 0.00, c.414A>G, is a known pathogenic variant in the middle of exon 2 shown to promote exon 2 skipping⁷. While we are unable to measure RNA scores for intronic SNVs, SpliceAI scores also correlate highly with function scores for intronic variants ($R = -0.90$, **Fig. 3c**), implicating splice disruption as the primary mechanism driving functional effects of intronic variants.

Text newly added to **Methods**:

To compare RNA scores to a state-of-the-art computational splice predictor, we obtained SpliceAI

scores for each SNV (included with CADD data). A single SpliceAI score per variant was defined as the maximum score for any predicted splice change among the independent SpliceAI scores for “acceptor gain”, “acceptor loss”, “donor gain” and “donor loss”. SpliceAI scores were compared to RNA scores for exonic variants and to function scores for intronic variants.

We also validated expression defects for two variants with low RNA scores by analyzing VHL protein abundance in monoclonal lines. Critically, both c.222C>A and c.462A>C showed highly reduced expression of VHL (**Supplementary Fig. 6**). From a technical perspective, we do not expect qPCR to be more precise than the NGS-based approach used to calculate RNA scores considering the high reproducibility we have shown.

4. The most impactful finding of these studies are the novel variants identified, yet this portion of the study are inadequately addressed. Authors need to validate both the impact of indels resulting in 42-AA extension of VHL as well as functionally evaluate its effect on HIF1a. A clear mechanism of how this impairs VHL’s ability to regulate HIF should be provided. Furthermore, the W88* opal codon hypothesis should be tested.

We agree that these interesting findings should be addressed further and now include much new data that provides independent validation and further mechanistic insights.

Regarding the long C-terminal extension, we originally validated the importance of the reading frame effect by analyzing CRISPR-induced indels in a reading-frame specific manner, showing only indels leading to the +1 bp reading frame were significantly depleted in HAP1, now in **Fig. 6c** reproduced here:

(We previously referred to this as a 42-AA extension, but the length of the extension common to

all variants in cBioPortal is correctly 41 AA, so we have revised accordingly.)

In revision, confirm this effect and investigate the mechanism of disruption by testing a specific variant in ClinVar, a VUS that we predict to be loss-of-function because it leads to the 41-AA C-terminal extension, c.606dup. We used homology-directed repair to engineer 3 independent HAP1 clonal lines harboring c.606dup. We also isolated a clone harboring a 5-bp deletion leading to the same C-terminal extension, c.620_624del. Cells were stained with anti-VHL and anti-HIF1A antibodies to perform confocal microscopy. Compared to controls, we observed a loss of VHL staining and an increase in HIF1A staining for all c.606dup clones and the c.620_624del clone. Signal intensities were quantified for at least 10 cells of each clone, using DAPI to quantify nuclear HIF1A staining specifically.

Figure. 6d, Clonal HAP1 lines harboring variants leading to the 41-AA C-terminal extension (c.606dup and c.620_624del) were stained for endogenous expression of VHL and HIF1A and imaged using confocal microscopy.

The lack of VHL staining suggested the 41-AA extension may lead to protein destabilization. Therefore, we compared protein levels directly in cells harboring c.606dup and c.620_624del to cells harboring other variants, including c.606del, a frameshifting variant at the same position that does not lead to the 41-AA extension. Western blot confirmed c.606dup leads to near-complete

loss of VHL protein, while c.620_624del leads to partial loss. Importantly, both showed strong HIF1A upregulation, whereas c.606del leads to a partial reduction in VHL protein without a concomitant increase in HIF1A. This is now included as **Fig. 6e**.

Figure 6e, A western blot was performed to assess VHL and HIF1A protein expression in clonally isolated HAP1 lines harboring specific variants of interest.

In summary, these data confirm that variants resulting in the 41-AA C-terminal extension cause VHL protein destabilization and subsequent HIF1A upregulation. This explains the enrichment of indels in a reading frame-specific manner present in ccRCC (**Fig. 6a**) and implicates germline variants such as c.606dup as being pathogenic for VHL disease. We anticipate this finding will enable definitive clinical interpretation of these variants.

We next validated our opal codon readthrough hypothesis for c.264G>A (p.W88*) with multiple additional lines of evidence. We isolated a c.264G>A HAP1 line, which shows a small amount of residual VHL expression via western blot compared to lines harboring frameshifting indels and a line with the c.263G>A (p.W88*) variant, which scores typically of other nonsense variants by SGE (**Fig. 6e**). Notably, HIF1A protein levels in cells with c.264G>A were elevated, but not to the same extent as in cells with either c.263G>A or a frameshifting indel in exon 1, consistent with this variant retaining partial ability to regulate HIF1A.

Next, to confirm whether this effect was due to stop-codon readthrough, we used a flow cytometry-based, dual-fluorophore reporter assay in which mCherry is only expressed if readthrough of a stop codon of interest occurs⁸. The result from this experiment is now included in **Fig. 6f**, with FACS data for each sample in **Supplementary Fig. 10d-f**.

Fig. 6f, A dual-fluorophore stop-codon readthrough (SCR) reporter assay was used to quantify readthrough of nonsense variants as the proportion of mCherry+ cells by flow cytometry. Error bars show the standard deviation of $n = 2$ replicate transfections.

Supplementary Fig. 10d, A dual-fluorophore stop-codon readthrough (SCR) reporter assay was used to quantify readthrough of nonsense variants assayed by SGE. Nonsense variants with 138 bp of surrounding *VHL* sequence were cloned between EGFP and mCherry such that mCherry expression only occurs if the nonsense codon fails to terminate translation. **e,f**, Flow cytometry

data is shown for each plasmid tested, with gating to determine the fraction of transfected cells positive for mCherry expression. Data was normalized to a control vector without a stop codon (pSCR-VHL-no-stop). Control plasmids are in (e) and plasmids containing *VHL* nonsense codons are in (f).

This assay shows c.264G>A to have high readthrough activity, whereas c.263G>A, a known ccRCC-causing mutation also leading to p.W88*, does not (**Fig. 6e**, **Supplementary Fig. 10d-f**). Additionally, we tested a second opal variant that creates the same 4-bp sequence context, c.351G>A. c.351G>A scored as LOF1 in SGE but was more mildly depleted than the corresponding amber codon at the same position (c.351G>A function score = -1.31 vs. c.350G>A function score = -2.44). We observed considerable readthrough activity for c.351G>A, whereas c.350G>A displayed almost zero readthrough. The readthrough activity for c.351G>A was less than for c.264G>A, consistent with the differences in function scores.

In summary, our new data corroborate stop-codon readthrough as a means of preserving partial *VHL* function. This is consistent with the published clinical description of c.264G>A as a type 2 *VHL* disease variant causing pheochromocytoma⁹. This finding has importance for how we understand *VHL*'s dose-dependent role in tumor formation across tissues. To our knowledge, there are very few instances in which stop-codon readthrough of a truncating variant has been shown to retain partial function and thus alter a monogenic disease phenotype.

We describe these validation experiments in the **Results** as follows:

To next assess the impact of the 41-AA extension, we introduced c.606dup, a ClinVar VUS leading to the 41-AA extension, to HAP1 cells. Across multiple clonal c.606dup lines, we observed loss of *VHL* expression and upregulation of HIF1A by confocal microscopy (**Fig. 6d**). This result was confirmed by western blot (**Fig. 6d**), which revealed a similar degree of HIF1A upregulation for c.620_624del but not c.606del, confirming reading frame specificity. Collectively, these results indicate that between p.R200 and p.*214, frameshifting indels promote ccRCC development via a long C-terminal extension that destabilizes *VHL* and impairs its ability to regulate HIF.

...

We introduced c.264G>A by gene editing and observed faint residual expression of *VHL* by western blot (**Fig. 6e**). HIF1A expression was upregulated compared to control cells, but to a

lesser extent than that observed in cell lines harboring c.263G>A or a frameshifting indel in exon 1. To specifically assess stop-codon readthrough potential, we used an independent, flow cytometry-based readthrough assay⁸ to compare c.264G>A to other *VHL* nonsense variants. Indeed, c.264G>A led to substantial readthrough, whereas c.263G>A did not (**Fig. 6f**, **Supplementary Fig. 10d-f**). We also tested c.351G>A in this system, a second variant that creates the same 4-bp stop codon context as c.264G>A. Likewise, considerable readthrough was detected for c.351G>A to compared to c.350G>A, albeit less than observed for c.264G>A, consistent with its lower function score. Together, these experiments indicate that differences in residual expression of *VHL* nonsense variants can affect the degree of functional impairment.

Details of how the experiments were performed are now provided in **Methods**:

Generation of clonal lines with *VHL* variants

To engineer specific SNVs via HDR, the following variants were ordered as oligonucleotides for In-Fusion cloning into HDR vectors: c.264G>A and c.606dup. Vectors matched the HDR library used to perform SGE, including the presence of synonymous edits at PAM sites. HAP1-Lig4KO cells were transfected using Xfect in 6-well plates, as above. Each transfection consisted of a pX459 plasmid and a single vector containing the desired edit. On day 6 post-transfection, cells were split by limiting dilution into a 96-well plate. Individual clones were picked and sequence-verified to identify lines with only the desired edit. In this process, additional lines harboring unintended edits in *VHL* were identified to use as controls. These included a negative control line that only received a synonymous PAM edit, a line with a 10-bp exon 1 insertion (c.332_333_insCTACCGAGGT, or 'c.332ins10' for short) and a line with a frameshifting deletion in exon 3, c.620_624del.

Additional cell lines for validating the SGE assay were created by prime editing in HAP1 *MLH1* knock-out cells (Horizon Discovery, HZGHC000343c022). On day 0, 0.5 ug of each epegRNA and PEmax plasmid (Addgene, 180020) were co-transfected into cells seeded at 90,000 cells per well in a 12-well plate using FuGENE HD transfection reagent (Promega). On day 1, cells were treated with puromycin (Thermo Fisher Scientific) at a final concentration of 2 ug/ml and maintained for 2 days. On day 3, puromycin was removed and cells were incubated one more day prior to being sorted into a 96-well plate using a Sony MA900 Cell Sorter. After 12 days of

propagation, gDNA was extracted using DNeasy Blood & Tissue Kit (Qiagen) and sequenced by Sanger sequencing. Lines generated in this fashion were verified to harbor SNVs without additional edits, including c.228C>G, c.222C>A, c.191G>C, c.292T>C, c.334T>C, c.371C>A, c.462A>C, c.539T>A, c.329A>C, c.302T>C, c.458T>C, c.263G>A, and c.606del.

Western blots

To assess protein expression of VHL and HIF1A in engineered cell lines by western blot, 0.3 million cells were seeded per well of a 6-well plate. After 2-3 days, cells were lysed in RIPA buffer (50 mM Tris-HCL at pH7.4, 150 mM NaCl, 1 mM EDTA, 1% Sodium Deoxycholate, 1% NP-40, 1% Triton X-100) containing protease inhibitors (Thermo Fisher Scientific). Protein concentration was determined by BCA assay (Thermo Fisher Scientific). 20 ug of total protein was loaded per well of Mini Protein Gels (Thermo Fisher Scientific). After gel running, protein was transferred to NC membrane using iBlot 2 Dry Blotting System (Thermo Fisher Scientific). The membrane was blocked with blocking solution containing 5% skim milk in TBST (100 mM Tris-HCl, 150 mM NaCl, 0.1% Tween-20, pH7.5) at RT for 1 hour. The membrane was then incubated at 4 °C overnight with primary antibodies including mouse tubulin antibody (Sigma-Aldrich, T6199, 1:3,000), rabbit VHL antibody (Cell Signaling Technology, 68547, 1:1,1000), and mouse HIF1A antibody (BD Transduction Laboratories, 610959, 1:1,000). After washing with TBST for 30 min, the membrane was incubated at RT for 1 hour with secondary antibodies including goat anti-Ms IgG (1:10,000) or goat anti-Rb IgG (1:10,000). After washing with TBST for 30 min, the membrane was treated with SuperSignal West Pico Plus Chemiluminescent Substrate (Thermo Fisher Scientific, PI34577) and the target protein band was visualized on ChemiDoc XRS+ system (Bio-Rad).

Immunofluorescence microscopy

Cells were seeded onto tissue culture-treated 35 mm imaging dishes (Ibidi). Dishes were washed with 1X DPBS (Gibco) and fixed with 10% Neutral-buffered formalin (Sigma-Aldrich) for 10 min, then washed twice with ice-cold 1X DPBS. Cell permeabilization was carried out using 0.2% Triton-X-100 (Thermo Scientific) for 10 min, followed by three washes with 1X PBST (1X DPBS, 0.1% Tween-20 (Thermo Scientific)). Samples were blocked by washing once and incubating in blocking buffer (1X DPBS, 1% BSA (Sigma-Aldrich)) for 1 hour. Samples were then incubated for 1 hour at room temperature with primary antibodies in blocking buffer. The primary antibodies

were used at the following concentrations: rabbit anti-VHL (Cell Signaling Technologies, 68547, 1:200), mouse anti-HIF1 α (Novus Biologicals, NB100-105, 1:50). Dishes were washed three times with 1X DPBS for 5 min each wash, and then incubated for 1 hour at room temperature with secondary antibodies in blocking buffer followed by three washes with 1X DPBS for 5 min each. The following secondary antibodies were used at a 1:500 dilution: donkey anti-Rabbit IgG Alexa Fluor 555 (Thermo Fisher, A-31572) and goat anti-Mouse IgG Alexa Fluor 647 (Thermo Fisher, A-21235). After the final wash, cells were mounted using mounting medium containing DAPI (Ibidi). Cells were imaged with a point scanning confocal microscope (Zeiss LSM880) using a 40X oil objective, and analyzed using FIJI.

Stop codon readthrough reporter experiments

A dual-fluorophore stop codon readthrough (SCR) reporter cassette⁸ was cloned into a pUC19 backbone with a CMV promoter for expression in mammalian cells. The SCR reporter vector expresses a single transcript encoding EGFP and mCherry, separated by a sequence of interest flanked by T2A sequences (**Supplementary Fig. 10d**). These sequences are in-frame, such that only if the sequence of interest reliably terminates translation will EGFP be expressed without mCherry. Conversely, if the sequence of interest does not terminate translation (for instance, if no stop codon is present), mCherry will be expressed proportionately to EGFP. The effectiveness of different stop codons at terminating translation can therefore be tested by cloning them into the reporter and assessing relative fluorophore levels via cytometry.

VHL-derived sequences consisting of 141 bp centered on each nonsense variant assayed were purchased (Twist Biosciences) and cloned into the SCR reporter vector to assess effects on translation termination. Each vector was transfected into HAP1-LIG4KO cells as described above. On day 5 post-transfection, EGFP and mCherry expression was assessed using a BD Fortessa X20 flow cytometer. Data was recorded for at least 150,000 single cells. Two transfection replicates were performed and sorted separately for each vector. Quadrant gating was performed (FlowJo) to determine the fraction of transfected cells (EGFP+) that were mCherry+. Data for individual stop codons was normalized to a vector containing a VHL sequence without any stop codon (pSCR-VHL-no-stop).

Minor Comments:

1. Please provide the % of all possible VHL variants observed in initial timepoint, post- transfection. It is unclear how truly comprehensive their representation is.

Thanks for letting us know this wasn't clear. In **Results**, we have edited the following sentence to provide a more precise percentage of variants assayed:

After stringent quality filtering, function scores for $n = 2,268$ SNVs were obtained, comprising 85.4% of all possible SNVs in SGE regions (**Supplementary Table 1**).

We've also added a sentence to **Methods** detailing regions for which all possible SNVs were designed:

The final designed oligonucleotide libraries contained molecules representing all possible SNVs within each SGE region, spanning regions of hg38 chr3 from 10,141,841-10,142,202, 10,142,743-10,142,876, 10,146,499-10,146,644 and 10,149,760-10,150,002.

(To clarify, all possible SNVs means 3 substitution variants per genomic position. Libraries span a total of 885 nucleotides of genomic sequence. Therefore: $2,268 / (885 \times 3) = 85.4\%$. The quality filters used for retaining variants are included in **Methods**, including to remove variants if the day 6 frequency in either replicate was less than 1.0×10^{-5} .)

2. Authors claim, "Frameshifting indels are strongly selected against in parental HAP1."
However, the data seem to indicate in-frame editing events also are strongly selected against. The authors should separate out mutation type and show the distributions more clearly if they want to make this claim.

The reviewer raises a good point. We focused our analysis on frameshifting indels to ask if loss-of-function leads to reduced growth, but there is strong selection against in-frame indels, too. In this experiment, Cas9 cleaves in sequence encoding L128, a core residue sensitive to missense variation by SGE. Clearly, in-frame indels are not well tolerated here. The key comparison, however, is that all indels are much more strongly depleted over time in wild-type cells compared to HIF1A-KO cells. Therefore, we have revised the figure as suggested and re-worded our claim:

Fig. 1b,c, CRISPR-induced editing of *VHL* was performed in HAP1 cells (day 0) and outcomes were quantified by next-generation sequencing (NGS). Distributions of indel scores, calculated as the log₂-ratio of abundance on day 13 to day 6, show that both frameshifting and in-frame indels are strongly depleted in parental HAP1 (**b**) compared to in HIF1A-knockout cells (**c**) (median indel score -3.20 vs. -0.20, respectively; Wilcoxon rank-sum $P < 2.2 \times 10^{-16}$).

Results:

Robust depletion of indels over time confirmed the essentiality of *VHL* for normal HAP1 proliferation (**Fig. 1b**). The strong selection against indels was eliminated by prior knockout of *HIF1A* (**Fig. 1c**), indicating *VHL* loss confers a HIF-dependent growth defect in HAP1 cells.

3. For better visualization and understanding of relevant “hits,” provide labels for some of the top scoring variants in Fig2.

We fully agree with the need for enhanced visualization. We have added labels to many low scoring missense variants in **Fig. 2c**. We think this better demonstrates the dynamic range of the assay. Well-known type 1 variants including W117C, S65W, and N78S score lowly, whereas well-known type 2 disease variants such as Y98H and Y112 are significantly below baseline but less reduced.

Fig. 2c, Function scores are plotted by genomic position for each coding and intronic region assayed, with β -sheets and α -helices of VHL's secondary structure¹⁰ shown above. Scores of select well-studied VHL disease variants are indicated by amino acid substitution, with SGE data for additional variants of well characterized phenotype in **Supplementary Table 2**.

In total, there are 347 SNVs with significantly reduced function scores. To allow better visualization of all hits we created a website. Users can search any variant of their choosing, hover the mouse over a variant to see its identity, plot function scores versus other metrics such as computational predictors, and interact with scores mapped to VHL's structure. The tool, which will be made public at time of publication, is available at: <https://vhl-board.onrender.com>. The login information for review is:

Username: vhl_viewer

Password: fun456

A sentence introducing the visualization tool has been added at the start of the **Discussion**:

Data for VHL SNVs scored by SGE are available to search and explore in relation to protein structure, computational predictors, and disease association via <https://vhl-board.onrender.com>.

(All code for the website is available on GitHub.)

4. Several VHL variants have been well characterized/studied (e.g. R167Q, Y112*, etc). These should be benchmarked/labeled throughout the figures within the manuscript wherever

relevant (both main figure and supplement).

Thanks for this suggestion. Many of the labels we have now added to **Fig. 2c** are for well-studied variants, such as R167Q. We now also highlight 2 variants with low RNA scores shown to disrupt splicing^{11,12} in **Fig. 3a,b** and low scoring residues with known pathogenic mutations in **Fig. 3g,h**.

We think our visualization tool will be helpful for identifying well-studied variants rapidly. However, we have also curated a list of clinically well-described variants to produce the new **Supplementary Table 2**, specifically for benchmarking SGE data.

Supplementary Table 2. SGE data for variants with established clinical phenotypes. To identify variants for benchmarking, curated clinical phenotype data was obtained¹³, including the number of kindreds analyzed, occurrence of ccRCC and pheochromocytoma, and the reported type of VHL disease. Variants seen in at least two kindreds were included for annotation with SGE data, excluding those seen in more than 1 individual in population sequencing (e.g. gnomAD) unless annotated as type 2.

5. The claim that only low RNA scores reliably predict LoF at the cellular level is unconvincing based on the data provided. Please provide a quantitative/statistical justification for this claim.

Thanks for the chance to clarify. We've addressed this point in response to a major comment from Reviewer #2. Briefly, we've added a demonstration of this threshold effect in **Supplementary Fig. 4c**. Our full response to reviewer #2 is reproduced as follows:

We attempted to show the dosage effect in **Fig. 3a**:

However, the dosage relationship is partially obscured because many different types of variants are plotted, with most LoF variants acting independently of mRNA dosage. While this global view is important, we can also focus only on synonymous variants to approximate a dosage curve (among exonic variants, “splice region” variants are synonymous variants within 3 bp of the exon junction). To make this clear, we have generated a new panel, **Supplementary Fig. 4c** that indicates the threshold below which variants tend to have significant SGE function scores:

Supplementary Fig. 4c, Comparison of function scores and RNA scores indicates that below an RNA score threshold of -3.0 (dashed line), 6 of 7 synonymous variants were significantly depleted.

There are relatively few variants to calibrate this dosage threshold, but all 7 synonymous/splice region variants with RNA scores below -3.0 (dashed line) have q-values below 0.05 (function scores from -1.62 to -0.38, mean = -0.93). The 3 synonymous variants with RNA scores between -2.0 and -3.0 have a mean function score of -0.12, and the 8 synonymous variants with RNA scores between -1.0 and -2.0 have a mean function score of -0.12, indicating a higher threshold

is unwarranted. From this analysis, an RNA score threshold of -3.0 seems appropriate for identifying variants whose mRNA depletion is expected to cause LoF. We refer to these as “low-RNA” variants. Indeed, all 10 low-RNA missense SNVs are depleted by SGE, validating the utility of the threshold.

We now highlight the dosage threshold better in text:

Analysis of synonymous variants reveals an RNA score threshold of -3.0 accurately predicts variants depleted in SGE experiments (**Supplementary Fig. 4c**). Indeed, 16 of 17 SNVs with RNA scores below -3.0 were depleted (function scores from -2.9 to -0.38), reflecting a minimum mRNA dosage required for normal growth.

6. In Figure 3, authors should separate out average score shown on structure based on the average of amino acids that share biophysical properties.

This comment is not entirely clear to us, but we interpret it to mean a version of original **Figure 3b** should be made showing only the average of conservative amino acid substitutions on the structure with the intent of highlighting residues highly sensitive to mutation. We have made this figure as follows, showing function scores for only conservative amino acid substitutions (left):

Response Fig. 2. Residue tolerance to conservative amino acid substitutions. Conservative missense variants were defined as those in the same Dayhoff category (a-f) as reference. We calculated an average missense function score for each residue, using only conservative missense variants (left) or all missense variants (right). 13 positions where no SNV results in an amino acid in the same category as reference are shown in gray. (Missense variants with RNA scores less than -2.0 were excluded.)

The new version of the figure highlights many of the same residues that are bright red using the initial strategy we employed, but shows less data overall. Some residues have no conservative mutations scored owing to the codon. Therefore, we have opted to show the data as before (averaging all amino acid substitutions at each site), noting **Fig. 3f-h** are now improved by showing more detail and labeling many of the lowest scoring residues.

Importantly, the visualization tool also allows users to click on an amino acid and see all the SGE scores of missense mutations at that position, which we think enables the best exploration of the data.

7. Relevant contacting amino acid positions should be labeled in all structural data shown in Fig

We have now done this (see next response).

8. Fig 3d should show ELOC interaction, much like HIF1a.

Thanks for these feedback points. We have substantially revised **Fig. 3g,h** to show contacting amino acids scoring lowly by SGE and the VHL-ELOC interface better, zooming in to residues scored lowly by SGE at the HIF1A and ELOC interactions. We have included labels for the lowest scoring contacting residues, inclusive of their average function scores.

9. In Fig3, authors should provide a figure showing the distribution of variant scores that reside at important protein interaction interfaces vs all others.

Thanks for this good suggestion. We agree this is a nice way of summarizing where loss-of-function missense mutations occur and have added a main figure panel to show this. We used

exposure labels for VHL missense mutations¹³ to quantify what fraction of missense mutations score as depleted in each of 4 categories: protein core, interaction core, interaction periphery, and surface.

Figure 3d, The proportion of missense variants scoring as depleted by SGE ($q < 0.01$) is displayed for each residue exposure label.

We describe these findings in **Results** as follows:

To explore features of missense variants impacting function, we examined where depleted mutations map to the protein's structure. The greatest proportion of depleted variants occur at core residues; 55.8% of mutations in the protein core and 49.4% of mutations in an interface core were depleted, whereas only 26.7% of peripheral interface mutations and 11.1% of other surface mutations score as depleted (**Fig. 3d**).

10. The correlations observed in Fig4c&d are unconvincing (e.g. $\rho = \sim 0.2$). Authors should provide rationale for why certain profoundly LoF variants are so infrequently observed while others are highly frequent. Perhaps, incorporating mutational signatures will help provide clarity (similar to Giacomelli AO, et al Nature Genetics 2018).

We agree with the reviewer that the correlations are highly imperfect. This is partly because we include all *VHL* variants across all cancer types, meaning noise is introduced by passenger mutations in cancer types in which *VHL* mutations are not causal for tumor development. Indeed, if we restrict the analysis in Fig. 4c to ccRCC cases only instead of all cancers, the correlation

between function score and cBioPortal entries strengthens substantially ($\rho = -0.58$). We have added this finding to the figure legend:

$\rho = -0.58$, $P < 2 \times 10^{-16}$ when analysis restricted to $n = 172$ SNVs seen in ccRCC

Allele frequencies are highly variable across tumor samples, owing to large differences in tumor purity and tumor heterogeneity. This means that while there is a significant correlation between function score and allele frequency, the high variability observed is to be expected.

Among significantly depleted variants, those with lower function scores are indeed seen in ccRCC more than variants with higher function scores. Our new function classes help to show this. While both LOF1 and LOF2 SNVs are significantly depleted in SGE experiments, it is the lower scoring SNVs in the LOF1 category that mostly drive ccRCC.

Supplementary Fig. 7h, The number of ccRCC entries in cBioPortal per variant is plotted by function class.

This is now referenced with statistics in **Results**:

Among SNVs depleted in SGE, LOF1 variants were seen far more frequently in ccRCC samples compared to LOF2 variants ($n = 225$ LOF1 SNVs seen in 1.30 ccRCC samples on average versus $n = 102$ LOF2 SNVs seen in 0.11 samples on average; **Supplementary Fig. 7h**), indicating the degree of functional impairment is strongly linked to the likelihood a ccRCC will develop once a mutation arises somatically.

We found the mutational signature analysis suggested by the reviewer intriguing and have looked into it. In the case of *TP53*¹⁴, the mutation signature analysis was very well motivated because

many variants were highly enriched in a cell-type specific manner. Looking at our data for *VHL* mutations in ccRCC scoring as LOF1, we do not see such a large range in cancer observations:

Response Fig. 3, Counts of ccRCC samples for LOF1 and LOF2 variants.

Most variants have been seen at least once, and no variant has been seen in more than 7 samples, reflecting a pattern similar to the Poisson distribution, which predicts ~1% of variants would be seen 5 or more times by chance. This contrasts the striking pattern of hotspot mutations in *TP53*, as Giacomelli et al. identified certain *TP53* variants in over 200 samples. Without other tissues for comparison, we attempted to take the ten LOF1 variants seen at least 5 times in ccRCC samples and run COSMIC's SigProfiler Assignment to ask if there was a specific mutational signature explaining their preponderance. However, the result was uninformative owing to the lack of power. In conclusion, while mutation frequencies are likely playing some role in determining which variants occur more often, we were not powered to see this.

11. Authors should provide more concrete numbers (e.g. % variants) for the claim "Variants observed repeatedly in population sequencing nearly always scored neutrally." It is hard to gauge this from the way the data is currently visualized.

Thanks for letting us know this was unclear. We have revised **Fig. 4e** to make our finding more apparent, using dashed lines to indicate which variants are seen in more than one individual, and also grouping variants by ClinVar status for visual clarity:

Fig. 4e, A combined population allele count for each SNV assayed was determined by summing independent observations from gnomAD¹⁵, UK Biobank¹⁶, and TOPMed¹⁷. Of $n = 233$ variants observed more than once in population sequencing, 96.6% were not significantly depleted.

(In the main text, additional details are provided as follows:

...among $n = 119$ SNVs seen at least five times in total across the UK Biobank¹⁶, gnomAD¹⁵, and TOPMed¹⁷ databases, no SNV scored below -0.40 (mean = -0.03 , s.d. = 0.14). Likewise, the lowest function score for any SNV seen at least twice was -0.77 .)

12. Authors should validate VHL variants of the pheochromocytoma phenotype by determining HIF1a activity biochemically and looking at downstream markers of HIF1a activity (VEGFA, CHGA, etc.) or reporter assay, comparing this with type 1 disease variants.

The major motivation for performing SGE experiments in the HIF1A-KO line was to ask whether variants associated with pheochromocytoma had HIF-independent effects on growth in HAP1. We did not see this, as all significant function scores (both LOF1 and LOF2) were dependent on HIF1A (**Fig. 5a**, reproduced below). This indicates HIF1A activity determines the phenotype we measure in SGE for type 1 and type 2 VHL disease variants alike.

Fig. 5a, $n = 797$ SNVs in exons 2 and 3 were assayed in HIF1A-KO HAP1 cells. Compared to previous data (top), variants were well-tolerated in HIF1A-KO cells independent of consequence.

As we have detailed in response to major comment #2, we now include western blot data to validate effects of type 2 variants on HIF1A levels, as well as variants scored as LOF2 by SGE. These include c.191G>C (p.R64P), c.292T>C (p.Y98H), c.334T>C (p.Y112H), and c.371C>A (p.T124K). We also used our SGE data to identify a new mechanism that may explain why c.264G>A causes type 2 disease despite being a nonsense mutation, and have now confirmed that c.264G>A leads to stop codon readthrough. We feel further specific validations of how type 2 disease variants act downstream of HIF are beyond the scope of the current work.

13. Please provide statistics for Fig 5c, similar to Fig 5f.

Thanks for pointing this out. We now provide the statistics requested in the Figure legend:

1-way ANOVA: $P = 0.004$ between 'pheno-predominant' and 'type unclear', $P < 1.0 \times 10^{-7}$ for all other comparisons.

14. It is hard to visually tell the difference between conditions in Supp Fig 1. Authors should provide image analysis quantification of confluence or viability readout normalized to VHL control.

Thanks for the feedback. We have removed the old **Supplementary Fig. 1** from the manuscript in line with the broad feedback to focus more on the mechanistic consequences of newly identified variants. This figure was an initial result that suggested HIF1A-KO may reverse effects of VHL loss in HAP1. We provide quantitative proof of this effect in revised **Fig. 1b,c**, and again for individual SNVs in **Fig. 5a**.

15. Given the authors claim that their SGE is comprehensive, they should provide a full heatmap of functional scores for all missense substitutions at all positions represented in the screen to better visualize the impact of each variant per position.

We cannot find a claim in the text using the word “comprehensive”, so it is unclear what this comment references specifically. We agree with the sentiment that any one assay cannot be truly comprehensive, and have clarified the coverage of all possible SNVs in response to minor comment 1, which was 85.4%.

In SGE, we aim to introduce all possible SNVs and not all possible missense variants per codon. This choice is justified because SNVs are more common in clinical sequencing and our goal was to obtain the cleanest data possible. Only with extensive optimization was this possible. Adding more variants to the library would have come at a cost of data quality.

As our focus was on SNVs in both coding and non-coding sequence, any heatmap of missense variants would be mostly incomplete because most possible AA substitutions are not made by SNVs. This is why we have opted to create an interactive visualization tool instead, which includes an option to visualize variants by nucleotide type in heatmap form (showing all possible base substitutions), as well as the option to click on residues of the protein structure to see data for missense mutations at the selected position.

To visualization tool is here: <https://vhl-board.onrender.com> Username: vhl_viewer
Password: fun456

Examples of various forms of displaying the data are shown below:

Data is also available for download in **Supplementary Table 1** for users interested in more specific analyses.

16. Are there any primary patient samples the authors could validate some of their novel variants in?

We have included new analyses of patient data from a VHL disease registry including 375 patients with variants assayed by SGE to be either LOF1 or LOF2. We retrospectively asked how well our function classes stratify risk of ccRCC in this cohort, first including all variants regardless of their novelty.

Supplementary Figure 9. Stratification of patients with VHL disease by SGE function class.
a, Patients in the Freiburg VHL Registry were grouped according to whether their germline *VHL* variant was functionally classified as LOF1 or LOF2 by SGE, and a Kaplan-Meier estimator was used to assess differences in age-related ccRCC penetrance with log-rank test for significance (additional details in **Methods**).

As advised by reviewer #2, we feature the same analysis restricted to missense variants due to the unique challenge they pose for clinical variant interpretation. This is now **Fig. 5d**:

Figure 5d, Patients in the Freiburg VHL Registry with missense variants assayed by SGE ($n = 321$) were grouped based on the function class of their germline variant. Due to the high prevalence of p.Y98H in this cohort, an additional LOF2 group excluding p.Y98H was analyzed, as well. A Kaplan-Meier estimator was used to assess age-related ccRCC penetrance (log-rank test for significance).

While this confirms the utility of our data in a different clinical context, more specifically to the reviewer’s question, a subset of patients in the registry harbor variants that lack definitive interpretations in ClinVar (e.g., VUS, absent). Therefore, we also performed a sub-analysis on this group and found that SGE data stratify ccRCC penetrance in a highly similar manner:

Supplementary Figure 8b, The same analysis was repeated including only patients with variants not reported to be pathogenic in ClinVar at time of analysis (i.e., absent, VUS, or conflicting interpretations).

Overall, the large difference in penetrance of ccRCC between LOF1 and LOF2 groups across these analyses further validates that our data inform disease risk, including for recently discovered variants.

17. A legend for Supplementary Table headers should be provided as it is hard to interpret some of the columns.

Thanks for this. We have included an appendix with the table to define all column headings. We have also added columns required during revision such as FoldX and SpliceAI predictions and removed extraneous columns containing results of intermediate steps in data processing.

Reviewer #2:

Remarks to the Author:

Findlay and colleagues present a MAVE assay covering most of the possible SNVs within the tumor suppressor gene VHL, using some refinements to their 'saturation genome editing' (SGE) approach. The resulting measurements strongly correlate with standing clinical variant interpretations for type 1 VHL disease, and they highlight some interesting exceptions with supporting phenotypic correlations from the literature. Their results are of technically very high quality, and are clearly presented. This study is another demonstration of MAVEs' promise for interpreting germline variants and will no doubt be of broad interest.

As I describe below, I did have a few concerns related to the calibration of the data (i.e., threshold-setting) for clinical variant interpretation, and about the analysis of splicing impacts.

We thank the reviewer for the positive comments and we agree these key points are worth exploring further.

1. RNA scores. The authors sequentially measure variants' depletion from the pool of spliced RNA at two timepoints. In the case of VHL, it appears that there was not much constraint at the level of splicing or RNA stability among exonic variants.

a. A key question which arises in the context of splice disruption is to determine a threshold of expression defect which is pathogenic. I am not sure whether that has been done here, but hope that such an analysis could be attempted given this rich dataset.

We thank the reviewer for this insightful comment and agree this is a unique opportunity to approximate a dosage threshold for function. We attempted to show the dosage effect in **Fig. 3a**:

However, the dosage relationship is partially obscured because many different types of variants are plotted, with most LoF variants acting independently of mRNA dosage. While this global view is important, we can also focus only on synonymous variants to approximate a dosage curve (among exonic variants, “splice region” variants are synonymous variants within 3 bp of the exon junction). To make this clear, we have generated a new panel, **Supplementary Fig. 4c** that indicates the threshold below which variants tend to have significant SGE function scores:

Supplementary Fig. 4c, Comparison of function scores and RNA scores indicates that below an RNA score threshold of -3.0 (dashed line), 6 of 7 synonymous variants were significantly depleted.

There are relatively few variants to calibrate this dosage threshold, but all 7 synonymous/splice region variants with RNA scores below -3.0 (dashed line) have q -values below 0.05 (function scores from -1.62 to -0.38, mean = -0.93). The 3 synonymous variants with RNA scores between -2.0 and -3.0 have a mean function score of -0.12, and the 8 synonymous variants with RNA scores between -1.0 and -2.0 have a mean function score of -0.12, indicating a higher threshold is unwarranted. From this analysis, an RNA score threshold of -3.0 seems appropriate for identifying variants whose mRNA depletion is expected to cause LoF. We refer to these as “low-RNA” variants. Indeed, all 10 low-RNA missense SNVs are depleted by SGE, validating the utility of the threshold.

We now highlight the dosage threshold better in text:

Analysis of synonymous variants reveals an RNA score threshold of -3.0 accurately predicts variants depleted in SGE experiments (**Supplementary Fig. 4c**). Indeed, 16 of 17 SNVs with RNA scores below -3.0 were depleted (function scores from -2.9 to -0.38), reflecting a minimum mRNA dosage required for normal growth.

Of note, our analysis is still limited by the relatively few variants with low RNA scores, but we feel this does provide a level of insight unique to our data set, especially as it informs the extent to which normal splicing must be disrupted for intronic variants to be depleted.

Along those lines, how concordant are the scores (splicing and function) with those of splicing effect predictors like MaxEntScan and SpliceAI? (A related limitation which could be more prominently mentioned is the lack of RNA measurement for intronic variants, where there might be the broadest range of effects)

We thank the reviewer for this great suggestion. Of many splice predictors, we focused our analysis on SpliceAI due to its strong performance in benchmarking studies¹⁸ and widespread use in clinical applications. In summary, we observe impressive correlations between SpliceAI scores and RNA scores for exonic variants, and between SpliceAI scores and function scores for intronic variants. We think this is an important point to highlight, both because it is confirmatory of splicing as the mechanism of variants with low RNA scores, and because it provides rare functional benchmarking of a widely used predictor. Accordingly, we've added two main figure panels and a supplementary figure on this analysis.

(This section of response is shared between reviewers.)

For orthogonal validation, we have used a computational predictor informative of splicing mechanisms (as suggested by Reviewer #2). A key advantage of this analysis is that it is applicable to the complete data set, informing predicted mechanisms of splice disruption and any systematic differences between computational and experimental approaches. We generated the following figure panels comparing SpliceAI scores to RNA scores of exonic variants and function

scores of intronic variants. These are now included as **Fig. 3b,c**:

Figure 3b,c, The maximum SpliceAI score for each SNV is plotted against RNA scores for exonic SNVs (Pearson's $R = -0.70$) and function scores for intronic SNVs ($R = -0.90$).

As SpliceAI is a model trained on sequence features of spliced transcripts, these strong correlations both validate our results and implicate aberrant splicing as the causal mechanism for exonic variants with low RNA scores (defined as < -3.0 ; $n = 17$) and for intronic variants scoring as depleted ($n = 47$). Regarding more specific mechanistic rationale, SpliceAI offers predictions specifically for acceptor gain, acceptor loss, donor gain, and donor loss:

Supplementary Fig. 6f, SpliceAI component scores predict specific splice alterations, including acceptor loss, acceptor gain, donor loss, and donor gain. Component SpliceAI scores are plotted against RNA scores for exonic SNVs.

As to be expected for exonic variants, SNVs with low RNA scores are often predicted to act via acceptor loss or donor loss.

We describe these new findings in the manuscript in **Results**:

Predictions from SpliceAI⁶, a computational predictor of splicing outcomes, were strongly correlated with RNA scores (**Fig. 3b**, **Supplementary Fig. 5f**). 16 of 17 SNVs with RNA scores less than -3.0 had SpliceAI scores greater than 0.08 (median of 0.61), whereas 83% of variants with RNA scores greater than -3.0 had SpliceAI scores of 0.00. The only variant with a low RNA score and a SpliceAI score of 0.00, c.414A>G, is a known pathogenic variant in the middle of exon 2 shown to promote exon 2 skipping⁷. While we are unable to measure RNA scores for intronic SNVs, SpliceAI scores also correlate highly with function scores for intronic variants ($R =$

-0.90 , **Fig. 3c**), implicating splice disruption as the primary mechanism driving functional effects

of intronic variants.

Text newly added to **Methods**:

To compare RNA scores to a state-of-the-art computational splice predictor, we obtained SpliceAI scores for each SNV (included with CADD data). A single SpliceAI score per variant was defined as the maximum score for any predicted splice change among the independent SpliceAI scores for “acceptor gain”, “acceptor loss”, “donor gain” and “donor loss”. SpliceAI scores were compared to RNA scores for exonic variants and to function scores for intronic variants.

We also validated expression defects for two variants with low RNA scores by analyzing VHL protein abundance in monoclonal lines. Critically, both c.222C>A and c.462A>C showed highly reduced expression of VHL (**Supplementary Fig. 6a**). From a technical perspective, we do not expect qPCR to be more precise than the NGS-based approach used to calculate RNA scores considering the high reproducibility we have shown (**Supplementary Fig. 4b**).

Please also note that the text now emphasizes the fact that we can only measure RNA scores for exonic variants.

b. Lines 190-191 (+338-339) are unclear. Are the authors suggesting that some RNA-depleting variants can be rescued by feedback/upregulation in order to survive to day 20? Why would a similar effect not influence the results of missense variants with stability/abundance (but not RNA) defects?

We agree this finding should be clarified. We included this point to show how RNA scores change globally over the course of the experiment, tending to increase only if low on day 6. We cannot say, however, whether this finding reflects cell-intrinsic upregulation, or whether among many 100s of cells edited to receive a given variant, those with stochastically lower *VHL* expression have been more strongly selected against by day 20. Both of these processes constitute selection for higher expression, and as there is no straightforward way of distinguishing cell-intrinsic feedback from population-level stochasticity, we stopped short of commenting on potential upregulation.

The reviewer raises a good point that there could also be selection for higher expression of hypomorphic proteins. We have checked for this by comparing the Δ RNA values (day 20 RNA score minus day 6 RNA score) across function classes, using our new classification system detailed below in which LOF1 variants score more lowly than LOF2 variants.

Supplementary Fig. 5e, The Δ RNA score for each SNV, defined as the day 20 RNA score minus the day 6 RNA score, is plotted by function class. Variants with day 6 RNA scores below the threshold of -3.0 are plotted separately.

Among variants with relatively normal RNA scores (left), Δ RNAs are subtly highest in the LOF2 class, perhaps suggesting slight selection for higher expression of hypomorphic variants, albeit the only significant differences by one-way ANOVA are between neutral variants and each of the other 3 classes (all $P < 0.0001$). Importantly, these effects are very small compared to the Δ RNA scores observed for variants with low day 6 RNA scores (right). Indeed, 11 of 17 variants with low day 6 RNA scores have day 20 RNA scores near or above the functional threshold of -3.0 , as shown in **Supplementary Fig. 5d** (reproduced):

This suggests that the functional selection creates a population in which more cells express enough *VHL* mRNA for normal growth.

There are several possible reasons as to why this effect may be restricted to low-RNA variants. The detection of these variants in full-length mRNA indicates they still enable a small amount of functional mRNA to be produced on day 6. If the defect is purely quantitative, it would follow that increased mRNA expression would inherently result in increased abundance of normal protein. On the other hand, missense and nonsense variants leading to qualitative protein defects despite normal mRNA expression may produce relatively normal amounts of protein. Cells may not be able to upregulate VHL mRNA expression if mRNA levels are already normal, and our RNA scores do not reflect post-translational upregulation. Critically, as VHL acts as part of a E3 ligase complex, it would be anticipated that functional complex levels depend on both intrinsic protein function and complex stoichiometry. This means increasing VHL protein expression above normal may not lead to more functionally active complexes.

We find it noteworthy that the relative increase in mRNA expression over time for low-RNA variants is confirmatory that these variants cause dosage-related defects. In light of this, we have replaced the original version of what is now **Supplementary Fig. 5e** with the new version above to better show that strongly positive Δ RNA values are unique to low-RNA variants.

We also modify the description of this effect in text to a) better reflect that the selection for higher mRNA is observed on the cell-population level, and b) to highlight that many low-RNA variants score above the dosage threshold of -3.0 by the end of selection:

While RNA scores across timepoints were highly correlated, variants strongly depleted in mRNA on day 6 tended to be less depleted in mRNA on day 20 (**Supplementary Fig. 5a-e**). This suggests that selection enriches for cells expressing sufficient levels of *VHL* mRNA for growth. Indeed, only 6 of 17 variants with day 6 RNA scores below -3.0 had day 20 RNA scores also below -3.0.

c. What was the basis for including the intron 1 region (given the apparent lack of effects?) I notice there is a cryptic exon in this region (eg PMID 31996412). Is this system informative as

to the functional effect of that exon's inclusion?

We thank the reviewer for giving us the opportunity to address this. Indeed, we chose this intronic region because a few variants within it had been associated with hereditary polycythemia (a recessive disease) and, putatively, type 2 VHL disease. We were curious to assess deep intronic variants in this region in an unbiased manner, as the discovery of new LoF variants would be clinically important for guiding genetic testing. However, we saw no SNVs with statistically significant function scores, nor did we see any meaningful correlation between function scores and SpliceAI scores, as shown here:

Response Fig. 4, Function scores and SpliceAI scores for SNVs in the intron 1 region.

From Lenglet et al. (2018) and Buffet et al. (2020), 6 unique SNVs reported from patients overlapped with the region of our intron 1 SGE library^{19,20}. The results for these variants are summarized in the new **Supplementary Table 3**.

In summary, the genetic evidence supporting variant phenotypes in this region is strongest for variants causing hereditary polycythemia, a recessive condition. Well known polycythemia variants in *VHL* coding regions also do not score lowly by SGE (shown in ST3). As polycythemia variants in *VHL* do not typically cause tumor predisposition, we can infer that the functional deficit of a single polycythemia variant is insufficient to be depleted in our assay.

Of note, two variants in the intron 1 region assayed have been putatively associated with VHL disease by Lenglet et al. (2018) and Buffet et al (2020), but these findings have yet to be confirmed

more broadly. Both variants are found only in individual families or single patients. The first variant, 340+617C>G, is reported as pathogenic in ClinVar but was seen in *cis* with a second variant, c.340+648T>C, a more common variant. Unfortunately, as SGE libraries only contain individual SNVs, we did not test this specific combination. Both 340+617C>G and c.340+648T>C score as neutral when tested individually.

The other variant putatively linked to VHL disease is c.340+682T>C (function score = 0.35), which Invitae recently classified as a VUS in ClinVar. Given this variant has a SpliceAI score of 0.00, was seen in a single tumor sample, and other mechanisms of VHL silencing²¹ may explain loss of expression, we find the link between this variant and VHL disease unconvincing.

On the whole, our analysis suggests the intron 1 variants assayed do not contribute to VHL disease burden individually. However, as the reviewer suggests, we cannot be fully confident our system is suitable for studying pseudoexon inclusion if the underlying splice mechanism is cell type-specific. SGE performs very well for intronic regions nearer to exons. However, without functional calibration of bonafide variants pathogenic for VHL disease that cause pseudoexon inclusion, it is impossible to rule out cell-type specific differences affecting only pseudoexon splicing. Therefore, while more detailed mechanistic work on this region is our focus, we opted to include the SGE data for intron 1 because it was a technically clean result and may inform variants discovered in this region in the future. To make the design rationale, findings, and potential limitations of the assay more clear, we have added the following excerpts.

In **Results**, concerning library design:

This intron 1 region covers the 5' end of a pseudoexon in which variants have been reported to impact *VHL* splicing in association with diverse phenotypes^{19,20}.

In **Discussion**:

We failed to score several variants causing recessive polycythemia as depleted by SGE, consistent with the functional impairment caused by these variants being insufficient to predispose patients to tumor development. This includes recently described variants near the pseudoexon region of intron 1 (**Supplementary Table 3**), where we observed no LoF variants. This result suggests there are unlikely to be a large number of undetected variants causing VHL disease or

sporadic ccRCC in this region, but we cannot rule out the possibility that our cell model is inadequate for studying pseudoexon inclusion, as variant effects on splicing may differ between cell types.

2. Structure-function analysis. The authors noted that LOF-scoring missense variants clustered in regions of secondary structure, which is unsurprising. I wonder if this part of the analysis could be developed further. Did these tend to be destabilizing (by Western or by FoldX/rosetta/etc prediction?) What fraction of the observed LOF effects are due to stability defects vs interaction defects?

Thanks for the helpful suggestions. We have looked substantially more into the structural impacts of variants, using analytical approaches and by directly measuring VHL protein levels for select variants. First, we used residue exposure labels for VHL missense mutations¹³ to quantify what fraction of missense mutations score as depleted in each of 4 categories: protein core, interaction core, interaction periphery, and surface.

Figure 3d, The proportion of missense variants scoring as depleted by SGE ($q < 0.01$) is displayed for each residue exposure label.

The greatest number of depleted missense variants ($q < 0.01$) occur at core residues; 55.8% of core mutations and 49.4% of mutations at an interface core were depleted, whereas only 26.7% of peripheral interface mutations and 11.1% of other surface mutations score as LoF. As advised

by Reviewer #1, we add this panel as **Fig. 3d**.

We looked at stability and interaction effects using FoldX, which has performed well in benchmarking of variant effect predictors that use structural information²². FoldX was used to predict $\Delta\Delta G$ values from the structure of VHL in complex with ELOC, ELOB, and the HIF1A peptide (PDB: 1LM8). We observed higher predicted $\Delta\Delta G$ values for depleted variants (median $\Delta\Delta G$ for depleted SNVs = 3.63, compared to 0.70 for other SNVs), confirming many variants scoring lowly by SGE are predicted to be destabilizing.

Figure 3e, FoldX-predicted $\Delta\Delta G$ values were higher for missense SNVs depleted in SGE experiments (median depleted variants = 3.63 compared to 0.70 for other SNVs; boxplot: center line, median; box limits, upper and lower quartiles; whiskers, 1.5x interquartile range).

Restricting analysis to only missense variants with significantly low function scores, FoldX $\Delta\Delta G$ still correlates reasonably well with function score (Spearman's $\rho = -0.41$).

Response Fig. 5, Higher FoldX-predicted $\Delta\Delta G$ values correlated with lower function scores

among low scoring SNVs. Here, analysis was restricted to only variants with significantly low function scores (Spearman's $\rho = -0.41$).

We now report this finding in text:

Restricting analysis to only depleted missense variants, FoldX $\Delta\Delta G$ values correlate inversely with function scores (Spearman's $\rho = -0.41$), indicating the degree of destabilization is further predictive of functional impairment.

We also used FoldX to obtain $\Delta\Delta G$ predictions from the structure of VHL alone (i.e., without ELOC, ELOB, and the HIF1A peptide), reasoning differences between $\Delta\Delta G$ predictions could inform interacting residues. As expected, the correlation between function score and FoldX-predicted $\Delta\Delta G$ values of missense variants was stronger using the VHL complex predictions ($\rho =$

-0.46) versus VHL alone ($\rho = -0.35$). Key residues highlighted by differences between the two approaches that also scored lowly by SGE include “interface core” and “interface periphery” residues, such as p.C162 and p.L158.

Response Fig. 6, Function and FoldX differences for residues predicted to destabilize complex interactions. The difference in FoldX-predicted $\Delta\Delta G$ values between VHL-only predictions and VHL-complex predictions is plotted. Only variants with significant function scores and VHL-only $\Delta\Delta G$ values less than 1.0 are displayed, highlighting residues important for complex stability.

We have now labeled key interaction residues C162 and L158, as well as active site residues scoring lowly by SGE, in an improved **Figs. 3g,h** to illustrate their high sensitivity to mutation.

We describe these findings in a new **Results** paragraph as follows:

To explore features of missense variants impacting function, we examined where depleted mutations map to the protein's structure. The greatest proportion of depleted variants occur at core residues; 55.8% of mutations in the protein core and 49.4% of mutations in an interface core were depleted, whereas only 26.7% of peripheral interface mutations and 11.1% of other surface mutations score as depleted (**Fig. 3d**). We next used FoldX⁵ to predict stability effects of missense variants. FoldX-predicted $\Delta\Delta G$ values were higher for variants with low function scores (median $\Delta\Delta G = 3.63$ for depleted SNVs versus median $\Delta\Delta G = 0.70$ for other missense; **Fig. 3e**). Indeed, 76.9% of depleted missense variants had predicted $\Delta\Delta G$ values greater than 2.0, compared to 20.6% of missense variants not deemed depleted by SGE. Restricting analysis to only depleted missense variants, FoldX $\Delta\Delta G$ values correlate inversely with function scores (Spearman's $\rho = -0.41$), indicating the degree of destabilization is further predictive of functional impairment.

Methods description:**Structural analysis**

Residue exposure labels for VHL missense mutations were obtained¹³, which define missense mutations as “protein core”, “interaction core”, “interaction periphery”, and “surface”. FoldX⁵ 5.0 was downloaded (<https://foldxsuite.crg.eu/>) and run locally to calculate predicted $\Delta\Delta G$ values. For this task, the VHL protein structure in complex with ELOC, ELOB, and the HIF1A peptide (PDB: 1LM8) was first repaired using “RepairPDB”, prior to running “PositionScan” to calculate $\Delta\Delta G$ values for all missense substitutions from p.R60 and p.Q209. The same process was repeated after removing ELOC, ELOB, and the HIF1A peptide chains from the VHL structure to predict $\Delta\Delta G$ values for the VHL structure alone. Missense mutations were deemed predicted to destabilize the complex if they had complex-derived $\Delta\Delta G$ values greater than 2.0 and VHL-only-derived $\Delta\Delta G$ values less than 1.0.

Lastly, we have now made our functional data available to view online with an interactive tool, such that users can look-up specific residues of interest and freely explore structure-function relationships by clicking on the structure of VHL to reveal function scores at the position. Upon publication this will be publicly accessible. Currently, the login information for review is as follows:

Username: vhl_viewer

Password: fun456

We introduce this in the first sentence of **Discussion** when summarizing our work:

Data for *VHL* SNVs scored by SGE are available to search and explore in relation to protein structure, computational predictors, and disease association via <https://vhl-board.onrender.com>.

A screenshot depicting the structure feature part of the website:

3. The poor specificity exhibited by the bioinformatic callers (Fig S8c) aligns well with others' observations from MAVE screens. Are the discordant variants (predicted by algorithm to be deleterious but neutral by MAVE) concentrated at any particular residues or domains? Is it correlated with conservation?

We agree this is a good opportunity to explore discrepancies between SGE and variant effect predictors (VEPs). To define discordantly classified variants, we first used ROC analysis to find optimal thresholds for distinguishing pathogenic *VHL* variants. We did this for each of the 3 top performing VEPs, namely, VARIETY, EVE, and REVEL, using gold-standard missense SNVs. Variants classified as neutral by SGE were then divided into those concordantly classified by the three top-performing VEPs ($n = 402$), those discordant with 1-2 VEPs ($n = 185$), and those discordant with all 3 VEPs ($n = 53$). The same procedure for LOF1/2 variants was performed.

Among neutral variants, only 53 were consistently discordant, suggesting using multiple predictors lessens the problem of poor specificity. Those that were discordant across all 3 predictors most commonly impacted core residues. However, the total fraction of neutral variants called discordantly by at least 1 predictor was comparable across regions:

Response Fig. 7, Discordantly classified missense SNVs by exposure. The fraction of neutral missense variants ($n = 640$) deemed discordant by top performing variant effect predictors (EVE, REVEL, and VARIETY) is plotted by protein exposure.

We did not observe a clear pattern regarding where discordantly classified neutral variants occur in relation to protein domains. Instead, they seem relatively evenly dispersed across the coding sequence.

Response Fig. 8, Discordantly classified missense SNVs by position. Function scores of missense variants deemed LOF1, LOF2, or neutral by SGE are plotted ($n = 867$). Neutral variants discordantly predicted by all 3 predictors (EVE, REVEL, and VARIETY) are colored by exposure. The 8 residues with 3+ neutral missense variants called discordantly are labeled.

Next, we compared function scores, EVE scores, vertebrate phyloP scores, and FoldX $\Delta\Delta G$ predictions across each category:

Supplementary Fig. 8e, Missense variants classified by SGE as LOF1/LOF2 or neutral were grouped by whether they were discordantly classified by 0, 1 to 2, or all 3 of top variant effect predictors (VARITY, EVE, and REVEL). Function scores, EVE scores, vertebrate phyloP scores, and FoldX predictions are shown across groups (boxplot: center line, median; box limits, upper and lower quartiles; whiskers, 1.5x interquartile range; all points shown except for $n = 22$ SNVs with FoldX scores greater than 12.0 in right panel).

The pattern across function scores is as expected, with LOF1/2 SNVs concordantly predicted by VEPs scoring lower than LOF1/2 SNVs with discordant VEP predictions. Among the 3 categories of neutral variants, there was no significant difference in score (1-way ANOVA, $P > 0.05$). EVE is an unsupervised deep learning model trained using species-level conservation. Interestingly, neutral variants discordantly classified by all 3 VEPs have significantly higher EVE scores than all other categories except LOF1/2 variants also predicted to be deleterious by all 3 VEPs. This trend is also reflected in nucleotide-level conservation, as vertebrate phyloP scores are not significantly different between LOF1/2 variants accurately predicted by all 3 VEPs and neutral variants discordantly classified by all 3 VEPs. In contrast, FoldX $\Delta\Delta G$ values are significantly higher for LOF1/2 variants predicted to be deleterious by all 3 VEPs or by 1-2 VEPs, compared to neutral variants predicted to be deleterious by all 3 VEPs (median 4.07 or 3.44 vs. 1.73, respectively; one-way ANOVA $P = 1.0 \times 10^{-7}$, $P = 0.01$).

In summary, the poor specificity of VEPs is, indeed, linked to high conservation. While this is a highly technical analysis, we do feel it is worth including this point. Therefore, we have added **Supplementary Fig. 8e** and reference it as follows:

Many missense SNVs scored neutrally by SGE are predicted to be deleterious by the models (**Supplementary Fig. 8c-e**). The absence of such discordantly scored variants from the gold-

standard ccRCC set indicates the computational models lack specificity for this phenotype, particularly for highly conserved variants. However, the pathogenicity of missense variants scoring lowly by SGE is generally well supported by computational prediction.

4. Calibration/clinical variant interpretation

We much appreciate the reviewer's feedback on this section of the manuscript and have made major changes accordingly. Central to our approach is a new classification system aiming to distinguish high-risk ccRCC alleles (LOF1) from other loss-of-function alleles (LOF2). This also increases the number of intermediate variants. We have also added many additional figure panels focused on missense and splice variants only. Lastly, we have validated the clinical utility of our function classes by analyzing phenotypic outcomes in a large cohort of patients with VHL disease. We think this section of the manuscript is now much stronger, better reflecting how functional impairment correlates to clinical risk across the full spectrum of SGE function scores.

- a. Fig 4 and Supp Fig 7 should be limited to missense/splice variants; otherwise the plots are somewhat dominated by syn and non/fs/indel which are peripheral to the interpretation challenge this paper addresses.

We fully agree missense and splice variants pose the biggest challenge for variant interpretation. We originally performed the "gold-standard" analysis of missense variants in **Fig. 4g** and the analysis of pathogenic missense variants only in **Supplementary Fig. 8a,b** in light of this. For reference, 60.7% of ClinVar-observed variants in Fig. 4a are either missense or splice region variants, including 39.6% of those with interpretations of pathogenicity (P/LP, B/LB).

Somewhat problematically, excluding synonymous SNVs removes nearly all ClinVar 'benign/likely benign' variants, making it harder to visualize the contrast between pathogenic and benign variants in the score distributions. As only 2 missense variants are deemed "likely benign" and none are deemed "benign" in ClinVar, this motivated our use of population data in defining "gold- standard" missense SNVs in Fig. 4g.

Therefore, we have expanded our analysis to show two versions of many plots and to emphasize

performance by consequence. We include several new figures limited to missense/splice region SNVs in **Supplementary Fig. 7**, but are amenable to moving some to the main figure if the reviewer feels this is essential upon seeing them.

First, we re-plotted the distribution of scores by ClinVar pathogenicity status (as in Fig. 4a) to only show missense and splice region variants, now included as **Supplementary Fig. 7a**:

Supplementary Fig 7a, The distribution of function scores for missense and splice region SNVs reported in ClinVar is shown ($n = 482$ SNVs, including 129 “pathogenic” and “likely pathogenic” SNVs and 15 “likely benign” SNVs).

As we had previously done for all Clinvar variants, we now generate ROC curves for only missense and splice region variants (**Supplementary Fig. 7c-f**). All four versions are shown here for comparison:

Classification performance drops only slightly when restricting analysis to missense and splice variants, yet a higher fraction of pathogenic variants are identified with 100% specificity. Importantly, as evidenced by the gold-standard analysis, none of the P/LP missense variants leading to reduced sensitivity here have been observed in a ccRCC sample in cBioPortal. While some are associated with VHL disease with other clinical manifestations, others may represent

classification errors, which are predicted to be more common when restricting to variants more difficult to interpret.

To explicitly show how our new functional classification system performs across different types of mutations, we have replaced old Supplementary Fig. 7c with a new figure that separates variants by consequence:

Supplementary Fig. 7g, Function classes, defined from SGE data, are illustrated to show performance at separating ClinVar variants by mutation consequence. Thresholds for distinguishing LOF1 (less than -1.26), LOF2 (less than -0.39), and neutral (greater than -0.23) classes are indicated. (Intermediately scored variants are not plotted.)

Regarding **Figs. 4b-d** on cBioPortal variants, the distributions look very similar if we restrict analysis to only missense and splice region variants. This analysis draws on variants discovered in an unbiased fashion across a wide range of tumors unselected for *VHL* mutation status. Therefore, missense variants are neither enriched or depleted. Here, we show the original **Fig. 4b** (left), and new **Supplementary Fig. 7b** with only missense / splice region SNVs (right):

Supplementary Fig. 7b,(right) Missense and splice region SNVs observed in cBioPortal are plotted by function score (inset shows variants present in at least one sample).

The correlation reported in **Fig. 4c** between all function scores and cBioPortal entries was $\rho =$
-

0.25 ($P = 9.7 \times 10^{-5}$). Restricting analysis to missense and splice variants, the correlation is slightly stronger: $\rho = -0.28$ ($P = 1.6 \times 10^{-4}$). Likewise, the correlation we reported between allele frequencies and function scores in **Fig. 4d** was $\rho = -0.22$ ($P = 3.7 \times 10^{-5}$); restricting to missense and splice, $\rho = -0.23$ ($P = 5.1 \times 10^{-4}$). The fact these correlations remain unchanged despite restricting analysis to missense and splice variants reflects the assay identifies driver mutations independent of variant type. In relation to the reviewer's comment, we have decided to keep **Fig. 4b-d** unchanged because biases in what is clinically interpretable are not influencing the distributions depicted, consistent with the unbiased nature of mining cBioPortal variants.

Lastly, in collaboration we have added a new analysis of ccRCC penetrance by analyzing patient outcomes in a VHL disease cohort according to our improved classification system. In **Fig. 5d**, we feature the analysis of this cohort restricted to patients with missense variants only.

Figure 5d, Patients in the Freiburg VHL Registry with missense variants assayed by SGE ($n = 321$) were grouped based on the function class of their germline variant. Due to the high prevalence of p.Y98H in this cohort, an additional LOF2 group excluding p.Y98H was analyzed, as well. A Kaplan-Meier estimator was used to assess age-related ccRCC penetrance (log-rank test for significance).

b. I wonder about the robustness of the thresholds in Figure 4f-h. It is great that 100% separation can be obtained between ccRCC and neutral variants after some curation, but I wonder how sensitive this may be to the curation criteria.

The reviewer once more hits on a critical question. We found it necessary to define gold-standard sets because mutation occurrence is inherently random in cancer, meaning inevitably some passenger mutations in *VHL* will occur in a small fraction of tumors. Relatedly, a small but appreciable fraction of pathogenic and benign germline variants in ClinVar are likely to be wrongly interpreted. Therefore, it is worth asking to what extent more stringent variant pathogenicity labels improve separation. If, for instance, discordant classifications are due to the assay not reflecting the biology of disease, we would not expect to see improved performance.

For these reasons, we required orthologous sources of evidence to define the gold standard set. However, the curation criteria we apply are still highly inclusive (i.e. requiring only 2 independent observations), leaving over 170 gold-standard missense variants for analysis. The key finding is that being more confident in what defines ccRCC-associated and neutral variants prevents what

may otherwise appear to be misclassification. In other words, there's no clear evidence the assay is missing causal variants for ccRCC because HAP1 is a poor model of VHL's ability to regulate HIF. We think this is an important point to show, though we fully agree with the next point about more carefully setting thresholds.

Also, in order to achieve this separation, the pathogenic/benign thresholds are very closely spaced, with a score difference of only 0.091 (-0.479 - (-0.388)), which appears (Fig 2A) to be considerably less than the average inter-replicate variation. I would encourage the authors to consider a less aggressive threshold selection (i.e., broadening the intermediate range or accepting some FP/FNs), and/or tempering the language about classification performance, lest marginal scores be given too much weight for novel variants.

Thanks for this suggestion. We have altered our classification system to include more intermediate variants. The assay reveals a continuous spectrum of functional impairment and *VHL* variants are associated with many clinical phenotypes, meaning no matter where we set thresholds there will inevitably be clinically relevant variants immediately on either side. For reference, variants scoring near the original thresholds (i.e., from -0.60 to -0.20) had a median inter-replicate function score difference of 0.190, reflecting relatively low technical noise. Yet, we fully agree that a difference between thresholds of 0.091 is too low.

In redefining our function classes, we found it important to distinguish between high-risk ccRCC variants and other variants with function scores inconsistent with germline benignity, but not typical of variants causing ccRCC. We think this is an important distinction to make in the context of VHL disease, and the threshold we use is well-supported by our analyses of clinical phenotypes. Therefore, we introduce the following 4-tier system:

1. LOF1 – Lowest scoring variants; in the range of gold-standard ccRCC-associated variants.
2. LOF2 – Variants with significant function scores that are below all gold-standard neutral variants yet not as low as LOF1 variants.
3. Intermediate – Variants with uncertain functional effects in the assay.
4. Neutral – Variants with non-significant function scores, in the range of gold-

standard neutral variants.

Previously only 35 variants scored intermediately. Our new intermediate class contains 173 variants. To define these groups we used both function score distributions and q-values, the latter of which reflect the degree of noise, thereby allowing us to ensure only variants with $q < 0.01$ are deemed LOF1 or LOF2. This classification system is now introduced in **Results**:

Together, these analyses indicate function scores can be used as strong evidence to support variant classification. Towards this end, we defined four function classes reflective of each variant's score in relation to gold-standard distributions and statistical confidence (see **Methods**). In summary, "LOF1" and "LOF2" variants both have significantly low function scores, though only "LOF1" variants score comparably to gold-standard ccRCC variants. In contrast, "neutral" variants scored similarly to gold-standard benign variants, whereas "intermediate" variants were scored ambiguously by SGE.

And detailed in **Methods** as follows:

Defining function classes

We derived four function classes for variants scored by SGE as follows:

1. Variants with q-values greater than 0.10 and function scores greater than -0.2188 (i.e. greater than the 5th percentile of gold-standard neutral variants) were deemed "Neutral".
2. Variants with function scores lower than -1.26 (i.e. lower than the 95th percentile of gold-standard ccRCC variants) were deemed "LOF1". (For all such variants, $q < 0.01$.)
3. Variants with q-values less than 0.01 and function scores less than -0.3875 (the lowest scoring gold-standard neutral variant) were deemed "LOF2".
4. Remaining variants were deemed "Intermediate". (q-value less than 0.10 and/or function score less than -0.2188, while not meeting criteria for LOF1 or LOF2).

This approach is nicely validated by the predominance of ClinVar-pathogenic variants in both LOF1 and LOF2 classes and nearly all ClinVar-benign variants being in the neutral class:

Response Fig. 9, Performance of SGE-based function classes across gold-standard ccRCC-associated variants, ClinVar variants, and variants yet to be reported in ClinVar.

LOF1 variants, explicitly defined by ccRCC association, include nearly all well-studied type 1 and type 2B VHL disease variants (i.e., types of VHL disease characterized by high ccRCC risk). On the other hand, many well-described type 2A and type 2C VHL disease variants score as LOF2. **Supplementary Table 2** has been added to show scores and classes for variants with well-curated phenotype data.

Some intermediate variants likely score as such because they truly have subtle effects on VHL function. For instance, 20 intermediate variants have q -values less than 0.01 (function score range: -0.387 to -0.286). While there are slightly more ClinVar-pathogenic variants ($n = 12$) than ClinVar-benign variants ($n = 10$) in the “intermediate” category, we would consider SGE-based evidence for these variants insufficient to support pathogenicity without more careful evaluation and explicit consideration of the specific clinical phenotype. We hope molecular pathologists and geneticists with expertise in different VHL-associated phenotypes will be able to use this data in the future to determine predictive thresholds for each VHL-associated phenotype, using this improved classification system as a foundation.

c. Fig 4h / lines 249-253. The suggestion here is that the function score may quantitatively measure the (i) degree of LOF, and (ii) risk conferred, within the low-scoring range. This is an interesting point and gets to a broader question of MAVEs’ calibration. The comparison made here is specifically among SNVs (a) absent from ClinVar and (b) with function score < -0.47 , between those seen in ccRCC vs not seen in ccRCC. There does seem to be a convincing difference (though it seems weaker when condition (a) is dropped; not obvious why this

would be?)

What it is notable then that very few of the seen-in-ccRCC variants fall between -0.479 and -2. This would seem to argue against the pathogenicity of the novel variants falling in to this range, but it also seems at odds with these variants' apparent depletion in population datasets (4e).

We agree this is a key point. Hopefully our new LOF1 and LOF2 classes make the distinction between the lowest scoring variants and other low-scoring variants more clear. In short, both classes are highly predictive of germline pathogenicity status for syndromic VHL disease, but only LOF1 correlates with high ccRCC risk.

First, in relation to the reviewer's point about condition (a) regarding ClinVar absence, our original finding was reported as:

Among SNVs absent from ClinVar that scored below -0.478, those observed in ccRCC tended to score more lowly (median score = -2.51 for ccRCC SNVs vs. median score = -1.21 for SNVs not seen in ccRCC; Wilcoxon rank-sum test $P = 7.2 \times 10^{-10}$).

If we include SNVs in ClinVar, too, the finding is similar:

Among SNVs that scored below -0.478, those observed in ccRCC tended to score more lowly (median score = -2.36 for ccRCC SNVs vs. median score = -1.28 for SNVs not seen in ccRCC; Wilcoxon rank-sum test $P = 1.2 \times 10^{-12}$).

As there is still a strong difference between ccRCC-SNVs and non-ccRCC SNVs, we can only speculate as to why dropping condition (a) makes the difference slightly weaker. We think this may relate to whether a variant arises through somatic mutation or is present in the germline. Perhaps in the context of germline disease, ccRCC predisposition is sufficiently conferred over the course of a lifetime by variants spanning a wider functional range, whereas somatic *VHL* mutations driving sporadic cases of ccRCC may be more strongly selected to be completely null. We do not feel we have sufficient data for such a claim, though.

Regarding the reviewer's observation that relatively few ccRCC variants score between -0.479 and -2.0, we agree that these variants seem less likely to cause ccRCC, yet tend to be pathogenic in ClinVar and depleted from population cohorts. This is because there is a wider range of functional impairment compatible with germline disease (including type 2 VHL disease, which lacks the markedly elevated ccRCC risk of type 1 disease) than there is for ccRCC development alone. Indeed, the reviewer's observation has motivated us to define the LOF1 class using the 95th percentile of gold-standard ccRCC variants as an upper bound.

We asked whether our new function classes accurately reflect differing cancer risk between LOF1 and LOF2 variants. Indeed, among $n = 225$ LOF1 SNVs, variants were seen in 1.30 ccRCC samples on average, compared to only 0.11 ccRCC samples for LOF2 SNVs ($n = 102$). In comparison, neutral variants averaged only 0.006 ccRCC samples per SNV and no intermediate variant was observed in ccRCC (new **Supplementary Fig. 7g**):

Supplementary Fig. 7h, The number of ccRCC entries in cBioPortal per variant is plotted by function class.

Whereas if we take both LOF1 and LOF2 variants together, we see a correlation between function score and ccRCC count (Spearman's $\rho = -0.40$, $P = 3.4 \times 10^{-14}$), within the LOF1 category only we see no correlation (Spearman's $\rho = 0.05$, $P = 0.44$). This justifies distinguishing between LOF1 and LOF2 variants without splitting further. As the reported statistic above was confusing, we have removed it in revision and instead reference this figure to make the point that LOF1 SNVs are seen in far more ccRCCs than LOF2 SNVs:

Among SNVs depleted in SGE, LOF1 variants were seen far more frequently in ccRCC samples compared to LOF2 variants ($n = 225$ LOF1 SNVs seen in 1.30 ccRCC samples on average versus $n = 102$ LOF2 SNVs seen in 0.11 samples on average; **Supplementary Fig. 7h**), indicating the

degree of functional impairment is strongly linked to the likelihood a ccRCC will develop once a mutation arises somatically.

Regarding the germline pathogenicity of LOF2 variants, 30 are “pathogenic” or “likely pathogenic” in ClinVar, and only 3 are “likely benign”. The “likely benign” annotations are potentially incorrect, as all come from single submissions of SNVs absent from reference databases and lacking functional evidence:

- 1.) c.357C>T (p.F119F) is predicted to potentially impact splicing (SpliceAI score = 0.41), and has an RNA score of -3.30 in SGE, which is below the dosage threshold set above.
- 2.) c.341-11T>G is an intronic variant with a high spliceAI score of 0.70 and a function score of -1.25.
- 3.) c.387G>A (p.L129L) scores near the intermediate range (-0.395), has a mildly low RNA score (-0.90) and a mildly high SpliceAI score (0.12).

In summary, we think the overall evidence is strongly in favor of LOF2 variants causing germline disease. Furthermore, many well-characterized type 2 VHL disease variants score in this range, including Y98H, Y112H, A149T, and T157I, and as detailed in **Supplementary Table 2**. Overall, the data strongly support the LOF2 function class as being indicative of germline pathogenicity for VHL disease, despite associating with lower ccRCC risk.

Importantly, we have now also analyzed our classifications using clinical data from a cohort of patients diagnosed with VHL disease:

Figure 5d, Patients in the Freiburg VHL Registry with missense variants assayed by SGE ($n = 321$) were grouped based on the function class of their germline variant. Due to the high prevalence of p.Y98H in this cohort, an additional LOF2 group excluding p.Y98H was analyzed, as well. A Kaplan-Meier estimator was used to assess age-related ccRCC penetrance (log-rank test for significance).

Supplementary Figure 8. Stratification of patients with VHL disease by SGE function class. **a**, Patients in the Freiburg VHL Registry were grouped according to whether their germline *VHL* variant was functionally classified as LOF1 or LOF2 by SGE, and a Kaplan-Meier estimator was used to assess differences in age-related ccRCC penetrance with log-rank test for significance (additional details in **Methods**). **b**, The same analysis was repeated including only patients with variants not reported to be pathogenic in ClinVar at time of analysis (i.e., absent, VUS, or conflicting interpretations).

This new analysis robustly confirms the difference in germline ccRCC risk between LOF1 and LOF2 and shows the applicability to variants lacking definitive ClinVar interpretations. This also suggests the data may one day help stratify patients with VHL disease for specific screening regimens and potentially therapies, especially in light of the availability of belzutifan.

5. Discussion

- As the authors acknowledge, a key limitation with MAVE assays is understanding how the cellular phenotypes map to the effects in vivo. In this case, the direction of effect appears to be opposite: VHL loss results in loss of fitness in culture but is tumorigenic in vivo (though the scores, with the sign flipped, are nevertheless highly predictive). It might be instructive for the authors to speculate (or cite any relevant literature) as to why constitutive HIF activity is toxic in culture.

We agree it's worth discussing more why the assay works despite not being performed in a ccRCC-related cell line, and mechanistically, why the effect may differ in specific cells of the kidney that give rise to cancer. We now address this in **Discussion**.

We show that reduced growth upon VHL loss is common to nearly all cell lines assayed in DepMap (**Fig. 1a**), and therefore not a unique feature of HAP1. We also show the effects we observe are consequential to HIF1A upregulation (**Fig. 5a**), though we have not examined which specific HIF1A targets are responsible. VHL's ability to regulate HIF is well-known to be critical to its tumor suppressor function, but how different HIF proteins play different roles in promoting or repressing cancers across different tissues has been a source of debate. It is also an open question as to how cells that become cancerous initially survive loss of VHL.

In addressing this point, we explain the mechanistic link between our assay and cancer and to summarize recent work on the contrasting roles of HIF1 and HIF2 in ccRCC, highlighting the role lineage-specific transcription factors and the hypoxic environment of the kidney:

Collectively, this study and other recent implementations of SGE^{2,23,24} show how relatively simple assays reflective of cell-intrinsic effects can identify variants driving human disease with accuracies close to 100%. In the case of *VHL*, the predictive power of SGE stems from accurate detection of LoF variants leading to reduced growth via upregulation of HIF1A. VHL's ability to regulate HIF is crucial to its tumor suppressor function across tissues²⁵, making the phenotype measured in SGE relevant to cancer despite the fact VHL loss leads to reduced growth in HAP1 and not excess proliferation. It is notable that HAP1 cells do not express *EPAS1*²⁶, which encodes HIF2A. The roles of HIF1 and HIF2 are thought to be opposing in ccRCC development, with the former acting as a tumor suppressor and the latter promoting growth *in vivo*^{27,28}. HIF1, specifically, has been recently shown to suppress cell proliferation by inhibiting amino acid biosynthesis²⁹, potentially explaining growth reduction in culture. While our study does not address why *VHL* loss and subsequent HIF upregulation promote growth specifically in cells that give rise to ccRCC, the hypoxic environment of the kidney and renal lineage-specific transcription factors are thought to play important roles^{30,31}. To date, SGE has only been performed in a limited number of cell systems. A key challenge going forward will be to extend such approaches to more cell types and

assays.

Minor

- Fig 1d. A scale bar in base pairs would be helpful. Also I presume this is so but the counts should be only variants for which the assay is informative, ie SNVs

We have added a scale bar to the revised figure.

For creating this schematic, the variant distribution graphic was obtained from gnomAD, which unfortunately doesn't allow restriction to SNVs only. Therefore, a small fraction of indels are included, too. To address this, we now explicitly state in the figure legend the number of SNVs reported in ClinVar that are included in SGE library design:

A total of 480 SNVs in ClinVar are in SGE regions, of which 269 are VUS.

For reference, we provide quality-controlled data for 459 of the 480 ClinVar SNVs included in library design, despite the challenge of high-GC content in exon 1.

- Fig 4c,d – shouldn't correlation stats be negative?

Yes, it is a negative correlation between function score and each metric. We have fixed this, thanks. (The previous correlation was calculated using negative function scores, so the statistics do not change other than in sign.)

- Fig 4e is overplotted and two of the status (conflicting and pathogenic) are shaded in very similar colors. Maybe separate by status and plot separately in a supplementary figure?

Thanks for this suggestion. We have made this figure more clear by splitting variants by ClinVar status and adding lines to separate variants not seen in population sequencing, from those seen once and those seen more than once. This is the new version of Fig. 4e:

Fig. 4e, A combined population allele count for each SNV assayed was determined by summing independent observations from gnomAD¹⁵, UK Biobank¹⁶, and TOPMed¹⁷. Of $n = 233$ variants observed more than once in population sequencing, 96.6% were not significantly depleted.

- Fig 7c – is the x label “Not pathogenic” accurate (as some have quite negative function scores)?
Or are these merely not seen in VHLdb?

(We assume this is in reference to Fig. 5c.) Thanks for the feedback. We see how this label is clearly confusing. Many of these SNVs are likely to be pathogenic based on our functional data, but the label reflects only the current status in ClinVar (i.e., “not pathogenic” includes variants that are VUS, (likely) benign, or absent from ClinVar).

We have changed the category label on the plot to “Other variants” to make this more clear, and explicitly defined what’s included in this category in the legend as follows:

The boxplot shows function scores for SNVs in each of these categories, as well as for all other SNVs assayed, including $n = 2,033$ variants either not deemed pathogenic in ClinVar or absent from the database.

- L290 “SNVs not deemed pathogenic” if this primarily refers to SNVs deemed absent from ClinVar I would rephrase as such

Thanks, we agree this is confusing. We have rephrased as suggested:

“Pathogenic variants not classifiable in this manner spanned a range of scores, as did SNVs absent from ClinVar.”

References

1. Richards, S. *et al.* Standards and guidelines for the interpretation of sequence variants: a joint consensus recommendation of the American College of Medical Genetics and Genomics and the Association for Molecular Pathology. *Genet. Med.* **17**, 405–424 (2015).
2. Findlay, G. M. *et al.* Accurate classification of BRCA1 variants with saturation genome editing. *Nature* **562**, 217–222 (2018).
3. Olbrich, T. *et al.* A Chemical Screen Identifies Compounds Capable of Selecting for Haploidy in Mammalian Cells. *Cell Rep.* **28**, 597–604.e4 (2019).
4. Beigl, T. B., Kjosås, I., Seljeseth, E., Glomnes, N. & Aksnes, H. Efficient and crucial quality control of HAP1 cell ploidy status. *Biol. Open* **9**, (2020).
5. Schymkowitz, J. *et al.* The FoldX web server: an online force field. *Nucleic Acids Res.* **33**, W382–8 (2005).
6. Jaganathan, K. *et al.* Predicting Splicing from Primary Sequence with Deep Learning. *Cell* **176**, 535–548.e24 (2019).
7. Flores, S. K. *et al.* Synonymous but Not Silent: A Synonymous VHL Variant in Exon 2 Confers Susceptibility to Familial Pheochromocytoma and von Hippel-Lindau Disease. *J. Clin. Endocrinol. Metab.* **104**, 3826–3834 (2019).
8. Toledano, I., Supek, F. & Lehner, B. Genome-scale quantification and prediction of pathogenic stop codon readthrough by small molecules. *bioRxiv* 2023.08.07.552350

(2023) doi:10.1101/2023.08.07.552350.

9. Zhang, M. *et al.* Von Hippel-Lindau disease type 2 in a Chinese family with a VHL p.W88X truncation. *Endocrine* **48**, 83–88 (2015).
10. Stebbins, C. E., Kaelin, W. G., Jr & Pavletich, N. P. Structure of the VHL-ElonginC-ElonginB complex: implications for VHL tumor suppressor function. *Science* **284**, 455–461 (1999).
11. Perrotta, S. *et al.* Effects of Germline VHL Deficiency on Growth, Metabolism, and Mitochondria. *N. Engl. J. Med.* **382**, 835–844 (2020).
12. Liu, F. *et al.* Case report: a synonymous VHL mutation (c.414A > G, p.Pro138Pro) causes pathogenic familial hemangioblastoma through dysregulated splicing. *BMC Med. Genet.* **21**, 42 (2020).
13. Gossage, L. *et al.* An integrated computational approach can classify VHL missense mutations according to risk of clear cell renal carcinoma. *Hum. Mol. Genet.* **23**, 5976–5988 (2014).
14. Giacomelli, A. O. *et al.* Mutational processes shape the landscape of TP53 mutations in human cancer. *Nat. Genet.* **50**, 1381–1387 (2018).
15. Karczewski, K. J. *et al.* The mutational constraint spectrum quantified from variation in 141,456 humans. *Nature* **581**, 434–443 (2020).
16. Karczewski, K. J. *et al.* Systematic single-variant and gene-based association testing of thousands of phenotypes in 394,841 UK Biobank exomes. *Cell Genom* **2**, 100168 (2022).
17. Taliun, D. *et al.* Sequencing of 53,831 diverse genomes from the NHLBI TOPMed Program. *Nature* **590**, 290–299 (2021).
18. Smith, C. & Kitzman, J. O. Benchmarking splice variant prediction algorithms

using massively parallel splicing assays. *bioRxiv* (2023)

doi:10.1101/2023.05.04.539398.

19. Lenglet, M. *et al.* Identification of a new VHL exon and complex splicing alterations in familial erythrocytosis or von Hippel-Lindau disease. *Blood* **132**, 469–483 (2018).
20. Buffet, A. *et al.* Germline mutations in the new E1' cryptic exon of the VHL gene in patients with tumours of von Hippel-Lindau disease spectrum or with paraganglioma. *J. Med. Genet.* **57**, 752–759 (2020).
21. Herman, J. G. *et al.* Silencing of the VHL tumor-suppressor gene by DNA methylation in renal carcinoma. *Proc. Natl. Acad. Sci. U. S. A.* **91**, 9700–9704 (1994).
22. Gerasimavicius, L., Livesey, B. J. & Marsh, J. A. Correspondence between functional scores from deep mutational scans and predicted effects on protein stability. *Protein Sci.* **32**, e4688 (2023).
23. Erwood, S. *et al.* Saturation variant interpretation using CRISPR prime editing. *Nat. Biotechnol.* **40**, 885–895 (2022).
24. Radford, E. J. *et al.* Saturation genome editing of DDX3X clarifies pathogenicity of germline and somatic variation. *bioRxiv* (2022) doi:10.1101/2022.06.10.22276179.
25. Ohh, M., Taber, C. C., Ferens, F. G. & Tarade, D. Hypoxia-inducible factor underlies von Hippel-Lindau disease stigmata. *Elife* **11**, (2022).
26. DepMap: The Cancer Dependency Map Project at Broad Institute. <https://depmap.org/portal/depmap/>.
27. Raval, R. R. *et al.* Contrasting properties of hypoxia-inducible factor 1 (HIF-1) and HIF-2 in von Hippel-Lindau-associated renal cell carcinoma. *Mol. Cell. Biol.* **25**, 5675–5686 (2005).

28. Shen, C. *et al.* Genetic and functional studies implicate HIF1 α as a 14q kidney cancer suppressor gene. *Cancer Discov.* **1**, 222–235 (2011).
29. Meléndez-Rodríguez, F. *et al.* HIF1 α Suppresses Tumor Cell Proliferation through Inhibition of Aspartate Biosynthesis. *Cell Rep.* **26**, 2257–2265.e4 (2019).
30. Kaelin, W. G., Jr. Von Hippel-Lindau disease: insights into oxygen sensing, protein degradation, and cancer. *J. Clin. Invest.* **132**, (2022).
31. Patel, S. A. *et al.* The renal lineage factor PAX8 controls oncogenic signalling in kidney cancer. *Nature* **606**, 999–1006 (2022).

Decision Letter, first revision:

1st Mar 2024

Hi Greg,

hope all is good.

Thank you for submitting your revised manuscript "Saturation Genome Editing Resolves the Functional Spectrum of Pathogenic *VHL* Alleles" (NG-A62550R). It has now been seen by the original referees and their comments are below. The reviewers find that the paper has improved in revision, and therefore we'll be happy in principle to publish it in Nature Genetics, pending minor revisions to satisfy the referees' final requests and to comply with our editorial and formatting guidelines.

Thank you again for your interest in Nature Genetics and please do not hesitate to contact me if you have any questions.

Congratulations!

My best wishes,
Chiara

Chiara Anania, PhD
Associate Editor

Nature Genetics
<https://orcid.org/0000-0003-1549-4157>

Reviewer #1 (Remarks to the Author):

The authors have made substantial revisions to respond the the prior review. No further revisions are necessary.

Reviewer #2 (Remarks to the Author):

Thanks to the authors for the thorough and thoughtful responses. The manuscript is greatly improved in revised form, in particular by the addition of the tiered classification.

I have only one comment on the added experiments, specifically on the clonal lines in Supp Fig 6. Isn't it surprising that c.222c>a, a splicing defect which seemed to reduce VHL levels more profoundly than all but the stop gain p.W88*, and has an RNA score (-4.45) under the threshold (Supp Fig 4c), yet has only a marginal function score (-0.40) and doesn't show increased HIF1A expression? I wonder if this might reflect variability in clonally isolated lines? In any event it would be worth commenting on.

Author Rebuttal, first revision:

Response to Referees of revisions: Saturation Genome Editing Resolves the Functional Spectrum of Pathogenic *VHL* Alleles

We thoroughly thank the reviewers for their prompt reviews of our substantially revised manuscript. Reviewer comments are reproduced below with our responses in blue.

Reviewer #1 Remarks to the Author:

The authors have made substantial revisions to respond the the prior review. No further revisions are necessary.

We thank the reviewer for their positive feedback.

Reviewer #2 Remarks to the Author:

Thanks to the authors for the thorough and thoughtful responses. The manuscript is greatly improved in revised form, in particular by the addition of the tiered classification.

We thank the reviewer for their kind feedback.

I have only one comment on the added experiments, specifically on the clonal lines in

Supp Fig 6. Isn't it surprising that c.222c>a, a splicing defect which seemed to reduce VHL levels more profoundly than all but the stop gain p.W88*, and has an RNA score (- 4.45) under the threshold (Supp Fig 4c), yet has only a marginal function score (- 0.40) and doesn't show increased HIF1A expression? I wonder if this might reflect variability in clonally isolated lines? In any event it would be worth commenting on.

We appreciate the attention to detail and fully agree regarding the caveats of clonally isolated lines. We have added the following comment to the figure legend to clarify:

“Clonal variability may account for subtle differences between results from the SGE assay and the degree of HIF1A upregulation observed by western blot.”

In the case of this specific variant, the strong decrease in VHL protein level confirms the variant's low RNA expression manifests in reduced protein expression. As c.222C>A is a synonymous SNV, any remaining protein is expected to be fully active. This is not necessarily the case for missense variants with low function scores (for instance, I180N exemplifies a missense variant with little impact on VHL level but a strong impact on HIF1A level). The c.222C>A clone suggests a relatively normal regulation of HIF1A can be maintained with substantially reduced VHL protein expression, provided it has full activity. This substantial dosage buffer may explain, for instance, why stop-codon readthrough can be sufficient to restore partial HIF1A regulation, thus altering clinical phenotypes.

We also note c.222C>A's function score of -0.40 is classified as “intermediate”. Function scores are more tightly linked to HIF1A levels than RNA scores are, so in this regard the absence of a strong HIF1A band is not fully contradictory. However, clearly the correlation between function score and observed HIF1A expression in clonally derived lines is imperfect, justifying the caveat we have added.

Final Decision Letter:

13th May 2024

Dear Dr. Findlay,

I am delighted to say that your manuscript "Saturation Genome Editing Maps the Functional Spectrum of Pathogenic *VHL* Alleles" has been accepted for publication in an upcoming issue of Nature Genetics.

Your paper will be published online after we receive your corrections and will appear in print in the next available issue. You can find out your date of online publication by contacting the Nature Press Office (press@nature.com) after sending your e-proof corrections.

Please note that *Nature Genetics* is a Transformative Journal (TJ). Authors may publish their research with us through the traditional subscription access route or make their paper immediately open access through payment of an article-processing charge (APC). Authors will not be required to make a final decision about access to their article until it has been accepted. Find out more about Transformative Journals

Authors may need to take specific actions to achieve compliance with funder and institutional open access mandates. If your research is supported by a funder that requires immediate open access (e.g. according to Plan S principles) then you should select the gold OA route, and we will direct you to the compliant route where possible. For authors selecting the subscription publication route, the journal's standard licensing terms will need to be accepted, including [a href="https://www.nature.com/nature-portfolio/editorial-policies/self-archiving-and-license-to-publish"](https://www.nature.com/nature-portfolio/editorial-policies/self-archiving-and-license-to-publish). Those licensing terms will supersede any other terms that the author or any third party may assert apply to any version of the manuscript.

If you have any questions about our publishing options, costs, Open Access requirements, or our legal

forms, please contact ASJournals@springernature.com

If you have not already done so, we invite you to upload the step-by-step protocols used in this manuscript to the Protocols Exchange, part of our on-line web resource, natureprotocols.com. If you complete the upload by the time you receive your manuscript proofs, we can insert links in your article that lead directly to the protocol details. Your protocol will be made freely available upon publication of your paper. By participating in natureprotocols.com, you are enabling researchers to more readily reproduce or adapt the methodology you use. [Natureprotocols.com](http://natureprotocols.com) is fully searchable, providing your protocols and paper with increased utility and visibility. Please submit your protocol to <https://protocolexchange.researchsquare.com/>. After entering your nature.com username and password you will need to enter your manuscript number (NG-A62550R1). Further information can be found at <https://www.nature.com/nature-portfolio/editorial-policies/reporting-standards#protocols>

Sincerely,
Chiara

Chiara Anania, PhD
Associate Editor
Nature Genetics
<https://orcid.org/0000-0003-1549-4157>